# LEARNING-GUIDED ROLLING HORIZON OPTIMIZATION FOR LONG-HORIZON FLEXIBLE JOB-SHOP SCHEDULING

**Sirui Li, Wenbin Ouyang, Yining Ma & Cathy Wu**
MIT
{siruil,oywenbin,yiningma,cathywu}@mit.edu

## ABSTRACT

Long-horizon combinatorial optimization problems (COPs), such as the Flexible Job-Shop Scheduling Problem (FJSP), often involve complex, interdependent decisions over extended time frames, posing significant challenges for existing solvers. While Rolling Horizon Optimization (RHO) addresses this by decomposing problems into overlapping shorter-horizon subproblems, such overlap often involves redundant computations. In this paper, we present L-RHO, the first learning-guided RHO framework for COPs. L-RHO employs a neural network to intelligently fix variables that in hindsight did not need to be re-optimized, resulting in smaller and thus easier-to-solve subproblems. For FJSP, this means identifying operations with unchanged machine assignments between consecutive subproblems. Applied to FJSP, L-RHO accelerates RHO by up to 54% while significantly improving solution quality, outperforming other heuristic and learning-based baselines. We also provide in-depth discussions and verify the desirable adaptability and generalization of L-RHO across numerous FJSP variates, distributions, online scenarios and benchmark instances. Moreover, we provide a theoretical analysis to elucidate the conditions under which learning is beneficial.

## 1 INTRODUCTION

Enhancing the efficiency and scalability of solving Combinatorial Optimization Problems (COPs) has been a central focus of both the Operations Research (OR) (Schrijver et al., 2003) and the emerging Neural Combinatorial Optimization (NCO) communities (Bengio et al., 2021). Numerous methods have been proposed to decompose large-scale problems into smaller, more tractable subproblems, with the majority focusing on spatial or structural decomposition of decision variables. Despite their success, these methods often face limitations in addressing the unique temporal challenges of *long-horizon* COPs - frequently encountered by industrial practitioners - which involve optimizing complex decisions over extended time horizons. Such unique challenges, coupled with the inherent NP-hardness and large-scale nature of the problems, call for advanced *temporal decomposition* strategies (Du and Pardalos, 1998; Hentenryck and Bent, 2006; Yang et al., 2013).

Building on successes in control for complex dynamical systems (Welikala and Cassandras, 2021; Franze and Lucia, 2015; Castaman et al., 2021), Rolling Horizon Optimization (RHO) has emerged as a natural temporal decomposition technique for long-horizon COPs (Glomb et al., 2022). RHO breaks the problem into overlapping subproblems with shorter planning horizons rolling forward over time, allowing for much better scalability. The key to RHO is its temporal overlap, which enhances decision-making at the boundaries of consecutive subproblems, mitigating myopic decisions and facilitating modelling interdependencies across the temporal dimension (Sethi and Sorger, 1991; Mattingley et al., 2011). However, such overlaps often lead to redundant computations that reduce the efficiency especially when only a small subset of variables needs re-optimization. This presents an opportunity to accelerate RHO by identifying such redundancies, an approach that, to our knowledge, has not yet been explored in the combinatorial optimization context.

To this end, this work introduces a novel learning-based RHO framework, termed L-RHO, designed to accelerate RHO for long-horizon COPs by identifying overlapping decision variables that do

not need to be re-optimized between consecutive iterations. We demonstrate the effectiveness of our L-RHO on the long-horizon Flexible Job-Shop Scheduling Problem (FJSP), a proof-of-concept example with numerous applications in manufacturing (Liu et al., 2017), healthcare (Addis et al., 2016), transportation (Zhang et al., 2012), and logistics (Xie et al., 2019). Long-horizon FJSPs represent a complex class of COPs involving interdependent assignment and scheduling decisions over extended time horizons, significantly increasing the difficulty compared to the basic JSP variant. The core of L-RHO is to intelligently fix operations assigned to the same machine across successive RHO iterations. By employing a customized network with tailored contextual embeddings, L-RHO learns to imitate a look-ahead oracle, effectively reducing the size of subproblems, significantly improving solve times, and delivering notable improvements in solution quality.

Moreover, to gain deeper insights into the research question - *"To what extent can machine learning accelerate RHO in the context of long-horizon COPs such as FJSP?"* - we validate and discuss our framework, L-RHO, through both computational experiments and theoretical analysis. Empirically, we show that L-RHO reduces RHO solve times by up to 54% and improves solution quality by up to 21% compared to various heuristic and learning-based baselines (with and without decomposition), across a range of FJSP settings and distributions. Moreover, we further discuss the unique potential of our L-RHO in online FJSP settings, where FJSPs operate with limited initial information and require ongoing but more efficient re-optimization as new orders arrive with dynamic environments. Theoretically, we formalize analytical criteria under which L-RHO outperforms the heuristic baselines; the analysis incorporates factors including the data distribution and the False Positive (FP) and False Negative (FN) Rates of the learning method, closely aligning with the empirical results.

Our contributions are fourfold: (1) We introduce L-RHO, the first learning-guided rolling-horizon method for COPs; by learning which overlapping solutions in consecutive RHO iterations do not require re-optimization, it significantly reduces RHO subproblem sizes and solve times. (2) Validated on long-horizon FJSP, L-RHO effectively accelerates RHO by up to 54% while largely improving the solution quality. (3) Extensive experiments show that L-RHO consistently outperforms various state-of-the-art heuristic and learning-based baselines (w/ or w/o decomposition), and demonstrates robust effectiveness across diverse FJSP settings, distributions, and online scenarios. (4) We further provide a principled probabilistic analysis to identify conditions where L-RHO surpasses heuristic RHO baselines, offering theoretical justifications that are aligned with the empirical observations.

## 2 RELATED WORKS

**Decomposition for COPs.** Despite advances in traditional solvers (Pisinger and Ropke, 2019; Vidal, 2022; Xiong et al., 2022), learning-guided solvers (Labassi et al., 2022; Scavuzzo et al., 2022; Wang et al., 2023b; Li et al., 2024) and neural solvers (Kool et al., 2018; Ma et al., 2024; Zhang et al., 2024c; Ye et al., 2024a) for COPs, scalability and adaptability to real-world complexities remain challenging. Various decomposition strategies have been explored by OR community, such as variable partitioning (Helsgaun, 2017), adaptive randomized decomposition (ARD) (Pacino and Van Hentenryck, 2011), and sub-problem constraint relaxation (Pisinger and Sigurd, 2007). More recently, machine learning has been exploited to guide the decomposition by selecting subproblems (Li et al., 2021; Zong et al., 2022; Huang et al., 2022) or to auto-regressively solve decomposed subproblems (Wang et al., 2021; Ye et al., 2024b; Luo et al., 2024), leading to notable improvements. However, they primarily emphasize *spatial* or *problem-structural* decomposition (e.g., for routing problems), which is not suitable for long-horizon time-structured COPs involving complex, interdependent decision variables and constraints spanning extended time horizons. This highlights the need for effective *temporal* decomposition. We note such temporal decomposition can be orthogonal to other existing ones, and future work could combine them to improve scalability and flexibility.

**RHO for Long-horizon COPs.** RHO is a temporal decomposition method originating from Model Predictive Control (MPC) (Garcia et al., 1989). It improves the scalability by dividing the time-structured problem into overlapping subproblems. While such overlap improves boundary decision-making, it can introduce redundant computations. Thus, many control and robotics studies leverage previous decisions to reduce the computations of the current subproblem. This is done, e.g., through hand-crafted methods, such as recording repeat computations (Hespanhol et al., 2019) or tightening primal and dual bounds (Marcucci and Tedrake, 2020), and learning-based models that predict active constraints (Bertsimas and Stellato, 2022) or solutions for discrete variables (Cauligi et al., 2021).

However, these methods are not tailored for long-horizon COPs. Recently, RHO has been initially adapted for long-horizon COPs, with wide applications such as scheduling (Bischi et al., 2019), lot-sizing problems (Glomb et al., 2022), railway platform scheduling (Lu et al., 2022), stochastic supply chain management (Fattahi and Govindan, 2022), and pickup and delivery with time windows (Kim et al., 2023). However, they often overlook the redundancies, and none of them have integrated machine learning to address this issue, leaving a gap in accelerating RHO for COPs.

**FJSPs.** FJSP is a complex class of COPs that involve interdependent assignment and scheduling decisions over extended time horizons, making it more challenging than the basic JSP, which only addresses scheduling (Dauzère-Pérès et al., 2023). Recently, learning-based solvers have shown power to outperform traditional methods such as Constraint Programming (Perron and Didier) and genetic algorithms (Li and Gao, 2016); for example, see Zhang et al. (2020; 2024b;a) for JSP and Wang et al. (2023c); Song et al. (2022) for FJSP. However, they are typically limited to small-scale instances (fewer than 200 operations) and struggle to scale to real-world, long-horizon scenarios. Although decomposition (e.g., the above ARD) has been explored for FJSP, their efficiency remains constrained. Moreover, these methods are offline, requiring full problem information, limiting their use in online setting. While some works have studied dynamic, online FJSP Luo et al. (2021); Lei et al. (2023), they are still restricted to small-scale experiments. Consequently, large research gaps remain in developing more scalable and flexible decomposition methods for long-horizon (and online) FJSP.

## 3 PRELIMINARIES

**FJSP Definition.** A FJSP instance consists of a set $\mathcal{T}$ of jobs, a set $\mathcal{M}$ of machines, and a set $\mathcal{O}$ of operations. Each job $j \in \mathcal{T}$ consists of a set of $n_j$ operations $\{O_{j,k}\}_{k=1}^{n_j} \subseteq \mathcal{O}$ required to be processed in a precedence order $O_{j,1} \rightarrow O_{j,2} \rightarrow ... \rightarrow O_{j,n_j}$. Each operation $O_{j,k}$ can be processed by any of the compatible machines $\mathcal{M}_{j,k} \subseteq \mathcal{M}$; the process duration of operation $O_{j,k}$ by machine $\bar{m} \in \mathcal{M}_{j,k}$ is denoted by $p_{j,k}^{\bar{m}}$. A solution to the FJSP, denoted as $\Pi = (m, \pi)$, consists of (i) an **assignment** $m : \mathcal{O} \rightarrow \mathcal{M}$ that represents the machine assignment of each operation, with $m(O_{j,k}) \in \mathcal{M}_{j,k}$ for all $j, k$, and (ii) a **schedule** $\pi : \{O_{j,k} \mid \forall j, k\} \rightarrow \mathbb{N}$ that represents the process start time of each operation $O_{j,k}$ as $\pi(O_{j,k})$; with the process duration as $p_{j,k}^{m(O_{j,k})}$, the process end time of the corresponding operation is $\pi_t(O_{j,k}) := \pi(O_{j,k}) + p_{j,k}^{m(O_{j,k})}$. Many FJSP objectives exist in the literature, and we consider testing our method on a variety of objectives including makespan (Xie et al., 2019), total start delays (Pham and Klinkert, 2008; Lee and Yih, 2014), and end delays (Zhou et al., 2009; Andrade-Pineda et al., 2020). Formally, the *makespan* objective can be expressed as $\max_{O_{j,k}} \pi_t(O_{j,k})$. For the delay-based objectives, let each operation $O_{j,k}$ be further associated with (i) a release time $s_{j,k}$ with a constraint $\pi(O_{j,k}) \geq s_{j,k}$ such that the operation can only be processed after the release time, and (ii) a target end time $t_{j,k}$; both respect the operations' precedence orders within the same job, that is, $\forall$ job $j$ and operations $k_1 < k_2$, we assume $s_{j,k_1} \leq s_{j,k_2}$ and $t_{j,k_1} \leq t_{j,k_2}$. The *total start delay objective* is expressed as $\sum_{O_{j,k}} \pi(O_{j,k}) - s_{j,k}$, and the *total end delay objective* as $\sum_{O_{j,k}} \max(\pi_t(O_{j,k}) - t_{j,k}, 0)$. Appendix A.1 provides a detailed list of notations.

**RHO for Long-Horizon FJSP.** The temporal structure of FJSP enables the use of RHO to decompose a long-horizon FJSP into a sequence of shorter-horizon FJSP subproblems.

Our RHO utilizes a planning window size $H$ but executes only a step size $S$ with $S \leq H$. Each subproblem is limited to solving for $T$ seconds. Given the long-horizon FJSP instance, we first sort the operations $\mathcal{O}$ into $\{O^{(1)}, ..., O^{(|\mathcal{O}|)}\}$ based on the precedence order within each job or the associated release time. These operations are then divided into overlapping subproblems. Specifically, the $r^{th}$ RHO iteration considers a FJSP subproblem $P_r$ given by a subset of operations $\mathcal{O}_{plan,r}$, consisting of the next $H$ non-executed operations according to the sorted RHO sequence order; we introduce additional constraints in $P_r$ to handle boundary conditions – for example, each operation in $\mathcal{O}_{plan,r}$ should start after all previously executed operations from the same job. After solving $P_r$ under the time limit $T$, we obtain the subproblem solution $\Pi_r = (m_r, \pi_r)$. We then execute the first $S$ operations with the earliest *process start time* $\pi_r(O)$ following the solution $\Pi_r$, while deferring the remaining $H - S$ operations for replanning in future iterations. The procedure is then repeated until we solve the full FJSP $P$, resulting in $|\mathcal{O}|/S$ subproblems. A detailed algorithm can be found in Appendix A.2. RHO is also well-suited for online settings with limited visibility, solving early operations while deferring the rest until future batches in the next iteration (see Sec. 5.2).

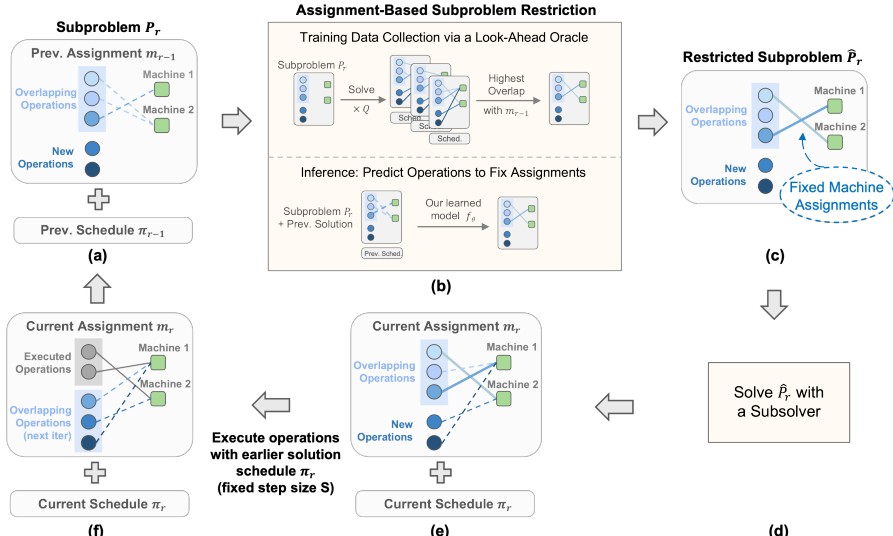

Figure 1: **One iteration of our L-RHO pipeline.** (a) Construct a subproblem $P_r$ with $H$ non-executed operations in $\mathcal{O}_{plan,r}$. This includes a set of overlapping operations $\mathcal{O}_{overlap,r} \subseteq \mathcal{O}_{plan,r}$, each associated with a solution from the previous iteration given by the assignment $m_{r-1}$ and the schedule $\pi_{r-1}$. (b) Identify operations $\mathcal{O}_{fix,r} \subseteq \mathcal{O}_{overlap,r}$ that in hindsight, their assignments did not need to be re-optimized; *during training*, a Look-Ahead Oracle determines $\mathcal{O}_{fix,r}$ by solving $P_r$ for $Q$ times; *During inference*, our learned neural network selects it. (c) Create a restricted subproblem $\hat{P}_r$ by fixing the machine assignments for $\mathcal{O}_{fix,r}$. (d) Feed $\hat{P}_r$ to a subsolver, solve for up to $T$ seconds, and (e) Obtain an updated solution $\Pi_r = (m_r, \pi_r)$. (f) Execute a subset of $S$ operations in $\mathcal{O}_{exec,r} \subseteq \mathcal{O}_{plan,r}$ based on the solution $\Pi_r$. We then repeat steps (a)-(f).

## 4 LEARNING-GUIDED ROLLING HORIZON OPTIMIZATION (L-RHO)

We now introduce our learning framework, L-RHO, that accelerates RHO for long-horizon FJSP. We refer to the original RHO, RHO for training data collection, and RHO for inference as $RHO_0$, $RHO_{data}$, and $RHO_{test}$. In the standard $RHO_0$, each consecutive iterations $r$-1 and $r$ share **overlapping operations** $\mathcal{O}_{overlap,r} = \mathcal{O}_{plan,r} \cap \mathcal{O}_{plan,r-1}$. While new operations $\mathcal{O}_{new,r} = \mathcal{O}_{plan,r} \setminus \mathcal{O}_{plan,r-1}$ in subproblem $P_r$ may change the solution of the overlap parts $\mathcal{O}_{overlap,r}$, we find **two key observations** that, for various FJSP distributions evaluated in Sec. 5:

(1) A significant subset of the overlapping operations retain the same machine assignment between the consecutive optimizations. Formally, the **shared operations** are given by $\mathcal{O}^*_{fix,r} = \{O \in \mathcal{O}_{overlap,r} \mid m_r(O) = m_{r-1}(O)\}$, where $\Pi_r = (m_r, \pi_r)$ and $\Pi_{r-1} = (m_{r-1}, \pi_{r-1})$ are the solutions of the unrestricted subproblem $P_r$ and $P_{r-1}$ in $RHO_0$.

(2) Let $\hat{P}^*_r$ be the restricted subproblem where we leverage the solution of the $r-1^{th}$ RHO iteration and fix the machine assignments of the shared operations $\mathcal{O}^*_{fix,r}$. The solve time of $\hat{P}^*_r$ is significantly reduced from that of the unrestricted subproblem $P_r$.

**Assignment-Based Subproblem Restriction.** Based on these observations, we propose restricting the subproblem by fixing machine assignments for a set of operations $\mathcal{O}_{fix,r} \subseteq \mathcal{O}_{overlap,r}$, selected through oracle, learning, or heuristic methods. This results in the restricted subproblem $\hat{P}_r$, with a formal formulation provided in Appendix A.5.2. Notably, this approach can be easily extended to other subproblem restriction methods, which we leave for future work.

**Our L-RHO Pipeline.** We design the L-RHO pipeline (which is shown in Fig. 1) as follows: First, we collect training data via a Look-Ahead Oracle by solving the unrestricted subproblem $P_r$ and identifying $\mathcal{O}^*_{fix,r}$ as the set of overlapping operations with the same assignments in $m_r$ and $m_{r-1}$. A neural network $f_\theta$ is then trained with such collected labels. During inference, $f_\theta$ predicts a subset $\hat{\mathcal{O}}_{fix,r}$, from which we form the assignment-based restricted subproblem $\hat{P}_r$. This largely speeds up the inference time optimization by replacing the expensive $P_r$ with the easier-to-solve $\hat{P}_r$.

**Training Data Collection (RHO$_{\text{data}}$).** At each iteration $r$, we solve the unrestricted sub-problem $P_r$ for $Q \geq 1$ times under a time limit, resulting in $Q$ possibly different solutions $\{\Pi_r^q = (m_r^q, \pi_r^q)\}_{q=1}^Q$. We then use the highest overlapping machine assignment as the training labels. That is, let $q^* = \arg\max_{q \in \{1,...,Q\}} \sum_{O' \in \mathcal{O}_{overlap,r}} y_{O'}^q$, with $y_{O'}^q = \mathbb{1}_{m_r^q(O')=m_{r-1}(O')}$. We set the training labels as $y_O^* = y_O^{q^*}$ for each operation $O \in \mathcal{O}_{overlap,r}$. Intuitively, the solve time of FJSP subproblem reduces when more assignemnts are fixed. Empirically, we find the labels $\{y_O^*\}_{O \in \mathcal{O}_{overlap,r}}$ not only lead to short solve times but also high solution quality. We then obtain the set $\mathcal{O}_{fix,r}^* = \{O \in \mathcal{O}_{overlap,r} \mid y_O^* = 1\}$ and construct the assignment-based restricted subproblem $\hat{P}_r^*$ accordingly. Fig. 1 (b) top illustrates the data collection procedure $RHO_{data}$. Comparing with $RHO_0$ that solves one unrestricted $P_r$ at each iteration, $RHO_{data}$ solves $P_r$ for $Q$ times and the restricted $\hat{P}_r$ one time, where $Q$ trades-off data collection efficiency and the training label quality.

**Input Features.** The state $\mathbf{s}_r$ at each RHO iteration $r$ consists of both the current FJSP subproblem $P_r$ and the previous solution $\Pi_{r-1} = (m_{r-1}, \pi_{r-1})$. We design a rich set of input features, capturing FJSP state details for overlapping operations $\mathcal{O}_{overlap,r}$, new operations $\mathcal{O}_{new,r}$, and machines $\mathcal{T}$. For $\mathcal{O}_{overlap,r}$ and $\mathcal{T}$, we further design features to capture the previous solution's information, including previous assignment, duration and end time (see Appendix A.3 for details).

**Neural Architecture.** Our neural network $f_\theta$ predicts the probability of whether each overlapping operation $O \in \mathcal{O}_{overlap,r}$ should be fixed at each state $\mathbf{s}_r$. It consists of three parts: (1) **Input Embedding**: generate embeddings for each operation $\mathcal{O}_{overlap,r} \cup \mathcal{O}_{new,r}$ and each machine $\mathcal{M}$, with separate MLPs. (2) **Machine/Operation Concatenation**: concatenate the hidden feature of each overlapping operation $O \in \mathcal{O}_{overlap,r}$ with both the hidden feature of the previously assigned machine $m_{r-1}(O) \in \mathcal{M}$, and a global feature obtained by a mean pooling. The concatenated features then pass through another MLP. (3) **Output**: each overlap-

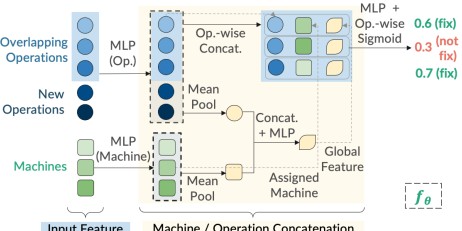

Figure 2: **Our neural architecture** $f_\theta$ to predict the probability of whether each overlapping operation should fix the assignment.

ping operation $\mathcal{O}_{overlap,r}$ is passed through a final MLP layer to output the probability prediction. Notably, the purpose of (2) is to augment each operation with both local information from the previously assigned machine, as well as global information from all entities, which include overlapping operations, new operations, and the machines. The illustration of $f_\theta$ is depicted in Fig. 2. The current simple architecture shows strong performance, and our preliminary comparisons with advanced architectures, such as Multi-head Attention, show no significant improvement (see Appendix A.6.3); we leave the exploration of more advanced designs in future work.

**Loss Function.** We use weighted binary cross-entropy loss where $w_{pos} > 0$ scales positive labels.

$$l(f_\theta(\mathbf{s}_r; O), y_O) = -[w_{pos}y_O \log f_\theta(\mathbf{s}_r; O) + (1 - y_O) \log(1 - f_\theta(\mathbf{s}_r; O))] \tag{1}$$

The weight $w_{pos}$ balances False Positive (FPR) and False Negative Rate (FNR). A $w_{pos} < 1$ emphasizes avoiding predicting False Positives, i.e., cases where the network incorrectly fixes changing operations, which can improve the objective but may increase solve time (see our analysis in Sec. 6).

**Inference (RHO$_{\text{test}}$).** The inference is illustrated in Fig. 1 (b) bottom, where we use the network $f_\theta$ to predict $\mathcal{O}_{fix,r}$: for each operation $O \in \mathcal{O}_{overlap,r}$, we predict $\tilde{y}_O = \mathbb{1}_{f_\theta(\mathbf{s}_r;O) \geq 0.5}$, which results in the fixed operation set $\mathcal{O}_{fix,r} = \{O \in \mathcal{O}_{overlap,r} \mid \tilde{y}_O = 1\}$ and the associated assignment-based restricted subproblem $\hat{P}_r$ solved at RHO iteration $r$. By replacing the unrestricted $P_r$ with the restricted $\hat{P}_r$, the computation time can be greatly reduced in comparison with $RHO_0$.

## 5 EXPERIMENT

In Sec. 5.1, we evaluate L-RHO in a standard offline setting, comparing it to various baselines for long-horizon FJSP. Next, in Sec. 5.2, we dive into detailed analysis of L-RHO under different FJSP variants including online settings. We provide additional experimental results in Appendix A.6, including a deep dive on the architecture design and evaluation on a real-world dataset. We aim to

answer: (1) To what extent does L-RHO improve upon competitive offline baseline methods and RHO variants? (2) Can L-RHO adapt to different FJSP distributions and objectives? Moreover, a unique benefit of RHO is its adaptability to online settings with limited visibility. As a step toward realistic online settings, we then study (3) Is L-RHO robust under noisy observation of process durations? (4) is L-RHO reliable under machine breakdowns?

**FJSP Data Distribution and Solver.** We generate synthetic large-scale FJSP instances following common procedure in (Behnke and Geiger, 2012), characterizes by $(|\mathcal{M}|, |\mathcal{T}|, n)$, where each job $j$ has $|\{\mathcal{O}_{j,k}\}_{k=1}^{n_j}| = n$ operations. The total number of operations $|\mathcal{O}|$ ranges from 600 to 2000. Notably, the problem horizon in this work (e.g., 600-2000) is **considerably larger** than those in previous studies (Wang et al., 2023a; Behnke and Geiger, 2012; Dauzère-Pérès and Paulli, 1997; Hurink et al., 1994; Brandimarte, 1993), which typically consider fewer than 500 operations. Following the conventions (Zhang et al., 2024a;b), we employ the competitive constraint programming (CP) solver OR-Tools CP-SAT (Perron and Didier) to solve FJSP, with its detailed CP formulation provided in Appendix A.5.2. We consider makespan objective in Sec. 5.1, and delay-based objectives in Sec. 5.2 (start and end delay) for comprehensive evaluation. More details are in Appendix A.5.1.

**Proposed Method (L-RHO) Setup.** We provide descriptions of the training set up in Appendix A.4, and the details on the choices of RHO parameters in Appendix A.5.4 and A.5.6.

**Baselines.** We include a range of traditional and learning-based baselines, with and without decomposition for comprehensive evaluation (see Appendix A.5.3 for more details about baselines):

(1) *Traditional solver w/o decomposition* - We include the widely used benchmark solver **CP-SAT** (Google, 2023) and **Genetic Algorithm (GA)** (Li and Gao, 2016).

(2) *Learning-based solver w/o decomposition* - We compare with the state-of-the-art DRL constructive method for FJSP (Wang et al., 2023a), which we denote as **DRL-20K** (using 20,000 training instances as in the original paper). We further compare with two recent learinng methods: **DRL-Echeverria** (Echeverria et al., 2023) and **DRL-Ho** (Ho et al., 2023).

(3) *Traditional solver w/ decomposition* - We include the time decomposition method **ARD-LNS (Time-based)** and machine decomposition method **ARD-LNS (Machine-based)** in (Pacino and Van Hentenryck, 2011), which decompose large neighborhood search (LNS) by randomly fixing temporal intervals or machine subsets as subproblems, respectively.

(4) *Learning-guided solver w/ decomposition* - While learning-guided decomposition exist, e.g, for routing problems (Li et al., 2021), they are not adaptable to long-horizon COPs like FJSP. We thus opt to include an **Oralcle-LNS (Time-based)** baseline to estimate the upper bound of enhancing ARD-LNS. It samples $K$ subproblems and selects the best one by looking ahead.

(5) *RHO decomposition* - We include **Default RHO**, where each iteration solves an unrestricted FJSP subproblem $P_r$ without fixing any variables, and **Warm-Start RHO**, where previous machine assignments of overlapping operations $\{m_{r-1}(O) \ \forall \ O \in \mathcal{O}_{overlap,r}\}$ are provided as hints (not fixed) to CP-SAT as a warm start, akin to (Hespanhol et al., 2019) in control systems.

**Evaluation Metric.** We compare the objective and solve time in Table 1. Objective improvement (OI%) and time improvement (TI%) over Default RHO are reported in Table 2. For a solver with an objective $obj$ and a solve time $t$, its OI and TI are calculated as $(obj_0 - obj)/obj_0 \times 100\%$ and $(t_0 - t)/t_0 \times 100\%$, respectively, where negative values indicate degradation.

## 5.1 CANONICAL OFFLINE FJSP UNDER MAKESPAN OBJECTIVE

We consider the canonical offline FJSP with makespan as the objective, with horizons up to 2000 operations (significantly longer than literature), following a similar synthetic data distribution as in Wang et al. (2023a). The number of jobs and machines are set to 20 and 10, respectively. For both L-RHO and DRL, we train separate models for 30, 40, and 60 operations per job, testing each in its respective setting and transferring the 60 operations-per-job model to a large-scale test with 100 operations per job. Results are gathered in Table 1. We highlight the following findings:

**Comparison with baseline solvers w/o decomposition.** L-RHO outperforms both traditional solvers (CP-SAT, GA) and the learning-based DRL solver in the standard (offline) FJSP setting. Specifically, (1) L-RHO achieves the best objective, outperforming DRL and all heuristic baselines in both in-domain and zero-shot generalization to larger scales; (2) We find the performance of DRL-Echeverria and DRL-Ho poor; (3) Comparing with DRL-20K (Wang et al., 2023a), we see that DRL Greedy is the fastest method but has a worse performance than Default RHO. (4) DRL

Table 1: Offline FJSP under Makespan Objective. Each column corresponds to a different FJSP size given by the format: total number of operations, followed by a tuple of (number of machines $|\mathcal{M}|$, jobs $|\mathcal{T}|$, and operations per job $n$). L-RHO outperforms all types of baselines.

| | **600** (10, 20, 30) | | **800** (10, 20, 40) | | **1200** (10, 20, 60) | | **2000** (10, 20, 100), **Transfer** | |
|---|---|---|---|---|---|---|---|---|
| | Time (s) ↓ | Makespan ↓ | Time (s) ↓ | Makespan ↓ | Time (s) ↓ | Makespan ↓ | Time (s) ↓ | Makespan ↓ |
| **CP-SAT (10 hours)** | 36000 | 1583 ± 65 | 36000 | 2128 ± 75 | 36000 | 3206 ± 87 | 36000 | 18821 ± 1986 |
| **CP-SAT** | 1800 | 2274 ± 147 | 1800 | 4017 ± 413 | 1800 | 10925 ± 1013 | 1800 | 39585 ± 2707 |
| **GA** | 1800 | 3659 ± 87 | 1800 | 5150 ± 92 | 1800 | 8086 ± 111 | 1800 | 10406 ± 640 |
| **ARD-LNS (Time-based)** | 300 | 2100 ± 269 | 400 | 3298 ± 365 | 600 | 5115 ± 558 | 1000 | 14258 ± 1522 |
| **ARD-LNS (Machine-based)** | 300 | 3974 ± 633 | 400 | 6594 ± 1227 | 600 | 12764 ± 4206 | 1000 | 52225 ± 1793 |
| **Oracle-LNS (Time-based)** | 300 | 1789 ± 104 | 400 | 2663 ± 120 | 600 | 4470 ± 151 | 1000 | 7275 ± 785 |
| **DRL-Echeverria, Greedy** | 84 ± 27 | 2347 ± 189 | 115 ± 32 | 3105 ± 161 | 105 ± 7 | 4570 ± 218 | 213 ± 10 | 7673 ± 356 |
| **DRL-Echeverria, Sampling** | 157 ± 8 | 2351 ± 139 | 217 ± 11 | 3131 ± 165 | 427 ± 20 | 4657 ± 192 | 603 ± 41 | 7774 ± 271 |
| **DRL-Ho, Greedy** | **4 ± 2** | 3030 ± 69 | **6 ± 4** | 4038 ± 87 | 11 ± 6 | 6045 ± 99 | 22 ± 3 | 10044 ± 115 |
| **DRL-Ho, Sampling** | 174 ± 1 | 2907 ± 46 | 168 ± 14 | 3907 ± 56 | 385 ± 12 | 5870 ± 71 | 517 ± 10 | 9848 ± 92 |
| **DRL-20K, Greedy** | 4 ± 0.02 | 1628 ± 72 | 6 ± 0.04 | 2128 ± 80 | 10 ± 0.1 | 3141 ± 97 | 16 ± 0.2 | 5184 ± 114 |
| **DRL-20K, Sample 100** | 29 ± 0.3 | 1551 ± 60 | 47 ± 1 | 2048 ± 69 | 110 ± 3 | 3063 ± 81 | 302 ± 10 | 5082 ± 98 |
| **DRL-20K, Sample 500** | 146 ± 5 | 1537 ± 61 | 261 ± 8 | 2031 ± 68 | 597 ± 16 | 3045 ± 79 | 1738 ± 16 | 5062 ± 98 |
| **Default RHO (Long)** | 599 ± 55 | 1529 ± 58 | 728 ± 98 | 2044 ± 75 | 1099 ± 108 | **3002 ± 87** | 2871 ± 244 | 4994 ± 114 |
| **Default RHO** | 244 ± 21 | 1558 ± 73 | 348 ± 26 | 2103 ± 78 | 545 ± 36 | 3136 ± 91 | 862 ± 42 | 5207 ± 114 |
| **Warm Start RHO** | 203 ± 23 | 1521 ± 67 | 278 ± 22 | 2055 ± 75 | 420 ± 33 | 3081 ± 96 | 716 ± 41 | 5057 ± 106 |
| **L-RHO (450)** | **126 ± 19** | **1513 ± 70** | **160 ± 23** | **2015 ± 86** | **259 ± 37** | 3011 ± 106 | **473 ± 52** | **4982 ± 132** |

Sampling improves its performance but significantly increases computational time, with DRL-20K (500 Samples) taking longer and performing worse than L-RHO. Additionally, the solve time of DRL heavily rely on batch-decoding and GPU availability, whereas our L-RHO has a competitive inference time even solely running on CPUs. (5) L-RHO requires significantly fewer training instances than DRL-20K, making it advantageous when obtaining FJSP instances is costly.

**Comparison with baseline solvers w/ decomposition.** We first observe that, (1) for ARD-LNS, time-based decomposition outperforms the machine-based variant but still results in a worse makespan and longer solve time compared to Default RHO. This suggests that effective temporal decomposition is crucial for time-structured, long-horizon COPs. In contrast, (2) L-RHO significantly outperforms both LNS methods in terms of solving time and objective. Furthermore, (3) while Oracle-LNS improves performance by simulating learning-based decomposition, it still underperforms compared to Default RHO. This suggests that RHO is better suited for long-horizon FJSP, partly due to complex temporal dependencies as follows: LNS begins with a complete solution and refines it iteratively. However, the makespan objective only improves when the last operation on the critical path is included. This temporal delay in propagating local improvements contrasts with prior learning-based LNS methods for Mixed Integer Programming (Huang et al., 2023) or Vehicle Routing (Li et al., 2021), where subproblem improvements directly impact the objective. In contrast, RHO builds the solution progressively, maintaining a tighter makespan by following the natural temporal order of operations. Lastly, we note that L-RHO's temporal decomposition is orthogonal to other strategies and could be combined in future work to enhance scalability and flexibility.

**Discussion - RHO for online settings.** Unlike offline baselines (CP-SAT, GA, ARD, DRL) that require complete information, RHO constructs solutions progressively using only near-future information, making it suitable for online settings. Next, we explore FJSP variants, including observation noise and machine breakdowns, to evaluate the applicability of L-RHO to batch-online FJSP.

## 5.2 Detailed Comparison with RHO Baselines Under Different FJSP Variants

We now benchmark L-RHO against diverse RHO baselines across FJSP variants. We evaluate objectives such as makespan, start delay, and end delay, while also testing its adaptability under higher system congestion, observation noise, and machine breakdowns to simulate real-world dynamics.

We include two more RHO baselines, following the $RHO_{test}$ procedure but obtaining the fixed operations $\mathcal{O}_{fix,r}$ with the following heuristics: **(a) Random** $\sigma_{\mathbf{R}} \in [\mathbf{0}, \mathbf{1}]$: $\mathcal{O}_{fix,r}$ is obtained by selecting each operation in $\mathcal{O}_{overlap,r}$ with a probability $\sigma_R$ uniformly at random, **(b) First** $\sigma_{\mathbf{F}} \in [\mathbf{0}, \mathbf{1}]$: $\mathcal{O}_{fix,r}$ is obtained as the first $\sigma_F$ fraction of operations in $\mathcal{O}_{overlap,r}$ with the earliest RHO sequence order based on the precedence order (for makespan) or release time (for delay-based objectives).

**New Objective: Start Delay.** In Table 2, under the new start delay objective, our L-RHO consistently outperforms all RHO variants in both solve time and objective across time horizons. Notably,

Table 2: Detailed Comparison of L-RHO with RHO variants under Start Time Delay Objective. We present the time and objective improvements (TI%, OI%) of RHO variants relative to Default RHO. The best TI% and OI% are bolded. L-RHO significantly outperforms all baseline RHO variants.

| | **600** (25, 25, 24) | | **1050** (35, 35, 30) | | **1600** (40, 40, 40) | |
|---|---|---|---|---|---|---|
| | **TI %↑** | **OI %↑** | **TI %↑** | **OI %↑** | **TI %↑** | **OI %↑** |
| **Default RHO** | 946.9s ± 31.6s (0.0% ± 0.0%) | 1790.5 ± 145.3 (0.0% ± 0.0%) | 1119.86s ± 35.5s (0.0% ± 0.0%) | 2960.4 ± 215.7 (0.0% ± 0.0%) | 2349.3s ± 111.7s (0.0% ± 0.0%) | 4148.6 ± 356.8 (0.0% ± 0.0%) |
| **Warm Start RHO** | 38.6% ± 1.0% | -7.5% ± 2.8% | -12.9% ± 1.6% | 11.2% ± 2.7% | 5.2% ± 2.4% | 6.8% ± 4.8% |
| **First 20%** | 26.1% ± 1.3% | 2.9% ± 2.5% | 10.2% ± 1.7% | -30.5% ± 10.7% | 31.0% ± 2.4% | -94.1% ± 24.3% |
| **First 30%** | 33.6% ± 1.3% | -7.0% ± 2.5% | 13.2% ± 1.7% | -48.9% ± 7.2% | 31.7% ± 2.1% | -129.4% ± 19.1% |
| **First 40%** | 34.8% ± 1.3% | -32.3% ± 3.7% | 14.1% ± 2.1% | -100.4% ± 8.7% | 34.7% ± 3.0% | -274.7% ± 28.1% |
| **Random 10%** | 23.0% ± 1.3% | -11.9% ± 2.5% | -15.5% ± 1.7% | -12.9% ± 3.4% | 10.0% ± 3.0% | -55.2% ± 9.6% |
| **Random 20%** | 23.8% ± 1.7% | -27.6% ± 3.4% | -2.0% ± 2.6% | -86.1% ± 7.9% | 10.2% ± 3.5% | -115.6% ± 14.5% |
| **L-RHO (Ours)** | **53.0% ± 1.0%** | **16.0% ± 1.9%** | **35.1% ± 1.3%** | **21.0% ± 2.4%** | **47.3% ± 2.2%** | **13.1% ± 4.4%** |

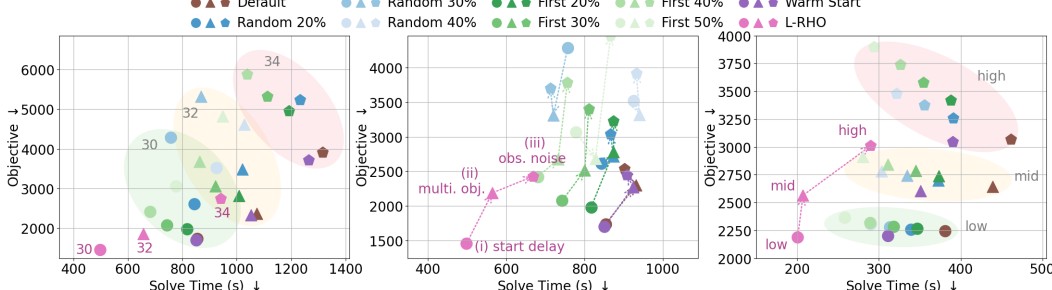

Figure 3: **L-RHO under different FJSP variants. Left: Increased Congestion level** with more jobs $|\mathcal{T}| \in \{30, 32, 34\}$. We circle the baselines for these three settings with green, yellow and red ellipsoids, using different markers to represent each setting. L-RHO is plotted in pink. **Middle: Multi-objective and Noisy Observation (online)**. We analyze FJSP $(30, 30, 24)$ with the (i) start delay objective (ii) start and end delay objective, and (iii) start and end delay objective plus observation noise. The arrows illustrate the performance changes of each method across (i)-(ii)-(iii). **Right: Machine Breakdown (online).** We simulate machine breakdowns during RHO's process, varying intensity (low, mid, high) by adjusting event frequency and machine availability.

it achieves a 35% - 53% speed up from Default RHO with a further 13%-21% improvement in the objective (total start delay), demonstrating the effectiveness of our learning method in accelerating RHO. Comparing with the baseline methods, we observe that: (1) Warm Start RHO does not significantly outperform Default RHO, indicating the inadequacy of simple warm start techniques for the complex long-horizon FJSP. (2) The assignment-based subproblem restriction heuristics, namely First and Random, improve the solve time from Default RHO, albeit at the expense of poorer objective. Moreover, First $\sigma_F$ emerges as a more effective heuristic than Random $\sigma_R$, with a better solve time and objective improvement when $\mathcal{O}_{fix,r}$ is of a comparable size ($\sigma_F = \sigma_R$).

**Data Distribution: Higher Congestion.** We raise the congestion level by increasing the number of jobs $|\mathcal{T}| \in \{30, 32, 34\}$ while keeping the number of machines $|\mathcal{M}| = 30$ and operations per machine to be 25 under the start delay objective. This results in a higher job-to-machine ratio, with congestion build-up over time. In Fig. 3 (left), we observe that L-RHO achieves solve time and objective improvements TI% $\in \{41\%, 39\%, 28\%\}$ and OI% $\in \{14\%, 20\%, 27\%\}$ across the different settings, demonstrating the robustness of the learning method to system congestion. In contrast, the baselines (First and Random) have limited solve time improvement and substantial objective degradation for all settings. This highlights L-RHO's ability to enhance system throughput by enabling the system to process more jobs than Default and baseline RHO methods.

**Multi-Objective: Start and End Delay.** While the total start delay objective emphasizes adhering to the release time, reducing the total end delay is crucial in many real world scenarios where following each operation's scheduled due time is important. Here, we use the sum of the start and end delay as the objective and generate random target end time for each operation (see Appendix A.5.1). In Fig. 3, middle (ii), we observe that L-RHO maintains solve time and objective improvements

with a TI% of $39\%$ and OI% of $4\%$ under this combined objective. Interestingly, Random and First also improve relative to Default RHO (better TI% and OI%, see Random $30\%$, for example). Empirically, we find more overlapping operations maintain the same assignment under this distribution, our analysis in Sec. 6 indicates the advantageous for Random and First under such a scenario.

**Online FJSP: Observation Noise.** In real-world FJSP deployments, practitioners often face incomplete and noisy operation data. To simulate this, we introduce observation noise into the RHO process, moving towards a more realistic online FJSP. Specifically, we assume that in each RHO subproblem, only the durations of operations with earlier release times are accurately observed. For other operations, we only have access to noisy estimates, and RHO optimizes subproblems based on this noisy information. Thus, the observed durations in $\hat{P}_r$ may differ from the true execution duration during execution (see Appendix A.5.1 for details). In Fig. 3, middle (iii), the Random and First strategies degrade significantly under observation noise, with a more pronounced effect on First. Our analysis in Sec. 6 captures such a behavior for First and Random, as the observation noise leads to (1) more overlapping operations changing assignments (2) a less informative distribution to determine which operation in the subproblem will change assignments. In contrast, L-RHO remains effective under observation noise, with a TI% of 25% and OI% of 4% over Default RHO.

**Online FJSP: Machine Breakdown.** In real-world FJSP scenarios, unexpected disruptions such as machine breakdowns can occur (Zhang and Wong, 2017). We simulate breakdowns, and combine the periodic RHO optimization with event-based rescheduling: when a breakdown or recovery occurs, we re-optimize the RHO subproblem with updated machine availability. This leads to two major changes in the RHO pipeline (1) RHO subproblems are solved more frequently due to the breakdown (or recovery) events, and (2) operations with all compatible machines down are deferred until the breakdown ends, thereby resulting in longer scheduling horizon (see Appendix A.5.1 for details). In Fig. 3 (Right), we test L-RHO under varying machine breakdown intensities by adjusting breakdown frequency and the proportion of downed machines, using makespan as the objective with 10 machines, 20 jobs, and 800 operations. Our L-RHO demonstrates significant time improvements TI% $\in \{47\%, 52\%, 37\%\}$ over Default RHO across all intensity levels, while also achieving a better makespan. This highlights L-RHO's reliability under unexpected disruptions. We also observe that, at high breakdown intensity, First loses its advantageous to Random. This can be similarly explained in our analysis next, as unexpected disruptions reduce the amount of information First can leverage.

## 6 THEORETICAL PROBABILISTIC ANALYSIS

In this section, we present a theoretical probabilistic analysis to identify when RHO can benefit from machine learning. Intuitively speaking, the more irregular the operations to be fixed ($\mathcal{O}_{fix}^*$), the more advantageous L-RHO can be, because of the greater potential gain from prediction. Also, L-RHO must balance both False Positive (FP) and False Negative (FN) errors: fixing something that should not have been (FP) harms the objective but helps the solve time, while failing to fix something that should have been (FN) harms the solve time and also indirectly harms the objective (under a fixed time limit). Thus, an ideal L-RHO method will balance the two errors (see Fig. 4).

**Notation.** We analyze a generic subproblem $\hat{P}_r := \hat{P}$ at each iteration $r$, and drop the $r$ subscript for ease of notation. We consider a fixed set of RHO parameters and denote $W = H - S = |\mathcal{O}_{overlap}|$. Given $\mathcal{O}_{fix}^*$ and $\mathcal{O}_{fix}$ from the Oracle and a method in {Random $\sigma_R$, First $\sigma_F$, L-RHO}, we denote $\mathbb{E}[n_{fix}^*] = \mathbb{E}[|\mathcal{O}_{fix}^*|]$, $\mathcal{O}_{fn} = \mathcal{O}_{fix}^* \setminus \mathcal{O}_{fix}$, and $\mathcal{O}_{fp} = \mathcal{O}_{fix} \setminus \mathcal{O}_{fix,r}^*$. We define the FN and FP *errors* $\mathbb{E}[n_{fn}] = \mathbb{E}[|\mathcal{O}_{fn}|]$ and $\mathbb{E}[n_{fp}] = \mathbb{E}[|\mathcal{O}_{fp}|]$. The (expected) FP and FN *rates* of each method are $\mathbb{E}[n_{fp}]/\mathbb{E}[n_{fix}^*]$, $\mathbb{E}[n_{fn}]/(W - \mathbb{E}[n_{fix}^*])$, with L-RHO's FPR and FNR denoted as $(\alpha, \beta)$.

We leverage an intuitive linear decay assumption, validated in Appendix A.7.2: in a given FJSP subproblem, the later an operation appears in the RHO sequence, the less likely its assignment should be fixed. This assumption allows us to formally relate the FP and FN errors of different methods (L-RHO versus Random versus First, see Prop. 1 and Fig. 4 (middle)).

**Assumption 1** (Linear Decay)**.** *Let the probability that each operation $O^{(i)} \in \mathcal{O}_{overlap}$ is included in $\mathcal{O}_{fix}^*$ be denoted as $p_{fix}^*(i) \in [0,1]$, where the index $i \in \{1,...,W\}$ follows the RHO sequence order. We consider linearly decreasing $p_{fix}^*(i) = b - m \cdot \frac{i}{W}$, where constants $0 \le b, m \le 1$ reflect average behavior across RHO iterations. We have $\mathbb{E}[n_{fix}^*] = \sum_i p_{fix}^*(i) = (b - \frac{m}{2})W - \frac{m}{2}$.*

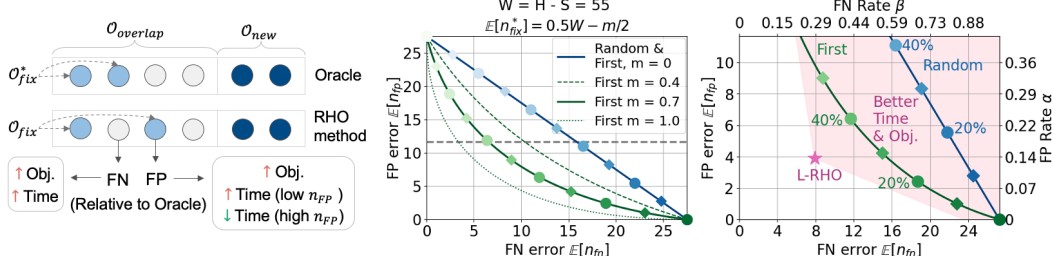

Figure 4: **Left:** We analyze FN and FP errors of each RHO method relative to Oracle and interpret their impact on objective and solve time. **Middle:** We show the FN and FP errors for Random and First with $\sigma_R, \sigma_F \in [0,1]$ under Assump. 1, fixing $b - \frac{m}{2} = 0.5$ while increasing the slope $m$. Higher $m$ enhances the performance of First relative to Random (with equal $\sigma$) by reducing both errors. We plot $\sigma_R, \sigma_F \in \{0, 0.1, ..., 1\}$ using circles and squares, with darker colors for smaller values. **Right:** For FJSP $(25, 25, 24)$ with the total start delay objective, we depict the FN and FP errors of {Random, First, and L-RHO} in the low FP region, using an empirically validated $p^*_{fix}(i)$ distribution. By reducing both $\mathbb{E}[n_{fp}]$ and $\mathbb{E}[n_{fn}]$, L-RHO empirically outperforms First $\sigma_F \leq 60\%$ in solve time and all baselines in objective (pink region). Transforming the coordinates (right and top axes) can further reveal the FP and FN rates L-RHO should achieve for learning to be effective.

**Proposition 1.** *Under Assump. 1, the FN and FP errors for each method is given in closed-form as*

- **Random:** $\mathbb{E}[n_{fn}^{Random}] = (1 - \sigma_R)\mathbb{E}[n_{fix}^*], \mathbb{E}[n_{fp}^{Random}] = \sigma_R(W - \mathbb{E}[n_{fix}^*])$ ;
- **First:** $\mathbb{E}[n_{fn}^{First}] = (1 - \sigma_F)(\mathbb{E}[n_{fix}^*] - \frac{m}{2}\sigma_F W), \mathbb{E}[n_{fp}^{First}] = \sigma_F(W - \mathbb{E}[n_{fix}^*] - \frac{m}{2}(1 - \sigma_F)W)$;
- **L-RHO:** $\mathbb{E}[n_{fn}^{L\text{-}RHO}] = \beta\mathbb{E}[n_{fix}^*], \mathbb{E}[n_{fp}^{L\text{-}RHO}] = \alpha(W - \mathbb{E}[n_{fix}^*])$.

*Furthermore, ignoring the $\frac{m}{2}$ term in $\mathbb{E}[n_{fix}^*]$ for ease of exposition, the FP and FN Rates of First $\sigma_F$ and Random $\sigma_R$ are $(\alpha_F, \beta_F) = (\frac{\sigma_F(1 - b + \frac{m}{2}\sigma_F)}{1 - b + \frac{m}{2}}, 1 - \frac{\sigma_F(b - \frac{m}{2}\sigma_F)}{b - \frac{m}{2}})$ and $(\alpha_R, \beta_R) = (\sigma_R, 1 - \sigma_R)$.*

**Interpretation.** The $p^*_{fix}(i)$ distribution determines the strength of the Random and First baselines and thus the potential for improvement by learning methods. (1) When $\mathbb{E}[n_{fix}^*]$ is close to 0 or 1, Random achieves low FP and FN errors by setting $\sigma_R$ close to $\mathbb{E}[n_{fix}^*]$. (2) Our analysis captures the empirical observation that First often outperforms Random: with a strong linear decay (a large slope $m$), First further reduces the FP and FN errors from Random for each $\sigma_F = \sigma_R$. (3) L-RHO is advantageous when its FP and FN rates $(\alpha, \beta)$ is lower than those for First and Random, as given in the Prop. This can provide practitioners a valuable way to assess the effectiveness of the learned model prior to deployment. We refer to Appendix A.7 for detailed proofs and interpretation.

## 7 CONCLUSION

We contribute the first learning-guided rolling-horizon method for COPs, which we call L-RHO. By learning which overlapping solutions in consecutive RHO iterations do not need to be re-optimized, our method substantially reduces RHO subproblem sizes and thus accelerates their solve time. L-RHO scales and accelerates RHO by up to 54%, improves solution quality, and outperforms a wide range of baselines, both with and without decomposition. We also analyze L-RHO's performance across various FJSP settings, distributions, and online scenarios, highlighting its flexibility and adaptability. We further provide a probability analysis to identify conditions where learning-guided RHO is most beneficial. One limitation is that L-RHO alone may not achieve state-of-the-art performance when extended to other long-horizon COPs. Future work could extend L-RHO to - 1) other long-horizon COPs, e.g., multi-item inventory management and vehicle routing with time windows, 2) training on diverse distributions for better generalization, 3) integration with other decomposition or subproblem restriction methods, and 4) addressing more complex real-world online scenarios. Our theoretical analysis may also guide the development of more effective RHO warm-start techniques with applications to control and robotics, e.g., accelerating Hybrid MPCs. We believe L-RHO offers a valuable tool for long-horizon COPs and will inspire future work in various domains. Our code is publicly available at https://github.com/mit-wu-lab/l-rho.

## 8 ACKNOWLEDGEMENT

The authors would like to thank Sean Lo for insightful discussions. This work was supported by the National Science Foundation (NSF) CAREER award (#2239566), MIT's Research Support Committee, an Amazon Robotics Ph.D. Fellowship, and a gift from Mathworks. The authors acknowledge the MIT SuperCloud and Lincoln Laboratory Supercomputing Center for providing HPC resources that have contributed to the research results reported within this paper.

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

# A SUPPLEMENTARY MATERIAL: *Learning-Guided Rolling Horizon Optimization for Long-Horizon Flexible Job-Shop Scheduling*

CONTENTS

## A.1 NOTATIONS

We provide detailed lists of notations for the Flexible Job-shop Scheduling Problem (FJSP) and Rolling Horizon Optimization (RHO) in Table 3 and Table 4.

Table 3: Notations for the Flexible Job-shop Scheduling Problem (FJSP).

| | | FJSP Related |
|---|---|---|
| **Problem P** | $\mathcal{M}$ | A set of machines. The machines are indexed by $m \in \mathcal{M}$. |
| | $\mathcal{T}$ | A set of job. The jobs are indexed by $j \in \mathcal{T}$. |
| | $\mathcal{O}$ | A set of operations. The operations are indexed by $O_{j,k} \in \mathcal{O}$. Each job $j \in \mathcal{T}$ consists of a set of $n_j$ operations $\{O_{j,k}\}_{k=1}^{n_j} \subseteq \mathcal{O}$. required to be processed in a predefined precedence order $O_{j,1} \to O_{j,2} \to ... \to O_{j,n_j}$. After sorting the operations by their respective release time (see below), each operation $O^{(i)} \in \mathcal{O} = \{O^{(1)}, ..., O^{(|\mathcal{O}|)}\}$ is indexed by $i$. |
| | Compatible Machines $\mathcal{M}_{j,k}$ | Each operation $O_{j,k}$ can be processed on any machine among a subset $\mathcal{M}_{j,k} \subseteq \mathcal{M}$ of compatible machines. |
| | Process Duration $p_{j,k}^m$ | $p_{j,k}^m$ denotes the process duration of operation $O_{j,k}$ on machine $m \in \mathcal{M}_{j,k}$. |
| | Release Time $s_{j,k}$ (Target Start Time) | Under delay-based objectives, each operation $O_{j,k}$ has a release time $s_{j,k}$, which respects the precedence orders of operations within the same job: for operations $O_{j,k_1}$ and $O_{j,k_2}$ with $\forall k_1 < k_2$, we assume $s_{j,k_1} \leq s_{j,k_2}$. Each operation can only be processed after its release time. |
| | Target End Time $t_{j,k}$ | Under the total end delay objective, each operation $O_{j,k}$ is associated with a target end time $t_{j,k}$, which represents a target time to finish the operation and similarly respects the operation precedence orders. |
| | Operations' Ordering $\{\mathcal{O}^{(1)}, ..., \mathcal{O}^{(|O|)}\}$ | Ordering of the operations $\mathcal{O} = \{O_{j,k}\}_{j,k}$ used to divide the full operation set $\mathcal{O}$ into overlapping RHO subproblems. • For the *makespan* objective, the operations are ordered by their relative position in the jobs based on the precedence order: For job $j$ with $n_j$ operations in precedence order $O_{j,1} \to ... \to O_{j,n_j}$, we give operation $O_{j,k}$ a score of $k/n_j$, and choose each RHO subproblem as the first $H$ non executed operations with the lowest score. • For *delay-based* objectives, the operations are ordered by their release time $s^{(1)} \leq ... \leq s^{(|\mathcal{O}|)}$. |
| **Solution** $\Pi = (\mathbf{m}, \pi)$ | Assignment $m$ | $m : \mathcal{O} \to \mathcal{M}$ represents the machine assignment of each operation, with $m(O_{j,k}) \in \mathcal{M}_{j,k}$ for all $j, k$. |
| | Schedule $\pi$ (Process Start Time) | $\pi : \{O_{j,k} \mid \forall j, k\} \to \mathbb{N}$, where the process start time of each operation $O_{j,k}$ satisfies $\pi(O_{j,k}) \geq s_{j,k}$. |
| | Process End Time $\pi_t$ | The process end time of operation $O_{j,k}$ is $\pi_t(O_{j,k}) := \pi(O_{j,k}) + p_{j,k}^{m(O_{j,k})}$, where $p_{j,k}^{m(O_{j,k})}$ is the process duration. |
| **Objective** | Makespan | $\max\limits_{O_{j,k} \in \mathcal{O}} \pi_t(O_{j,k})$, see Alg. 7. |
| | Total Start Delay | $\sum\limits_{O_{j,k} \in \mathcal{O}} \pi(O_{j,k}) - s_{j,k}$, see Alg. 7. |
| | Total Start and End Delay | $\sum\limits_{O_{j,k} \in \mathcal{O}} \pi(O_{j,k}) - s_{j,k} + \max(\pi_t(O_{j,k}) - t_{j,k}, 0)$, see Alg. 7. |

Table 4: Notations for the Rolling Horizon Optimization (RHO).

| | | **RHO Related: RHO Iteration $r$** |
|---|---|---|
| **Parameters** | $H$ | A planning window size ($H$ operations in the current planning window). |
| | $S$ | An execution step size ($S \leq H$ operations executed at each iteration). |
| | $T$ | A solve time limit of $T$ seconds per RHO subproblem. |
| | $T_{es}$ | We early terminate the solver if the value of the best feasible solution does not improve within $T_{es}$ seconds. (see Appendix. A.5.4) |
| **FJSP Optim. Subproblem** | Unrestricted $P_r$ | FJSP subproblem $P_r$ given by a subset of operations $\mathcal{O}_{plan,r}$, with additional constraints to ensure $\mathcal{O}_{plan,r}$ are processed after the previously executed operations within the same job or the same machine (see Appendix A.5.2 Alg. 8). |
| | Restricted $\hat{P}_r$ | Given a subset $O_{fix,r}$, we can construct a simpler, restricted FJSP subproblem $\hat{P}_r$ from $P_r$ by setting the compatible machines $M_{j,k} = \{m_{r-1}(O_{j,k})\}$ for $O_{j,k} \in \mathcal{O}_{fix,r}$. |
| **Relevant Operations Subset** | $\mathcal{O}_{plan,r}$ | Operations included in the current RHO subproblem $P_r$ or $\hat{P}_r$: The first $H$ non-executed operations with earliest RHO sequence order. See Appendix Sec. A.2 Alg. 2. |
| | $\mathcal{O}_{step,r}$ | Executed operations based on the solution of $P_r$ or $\hat{P}_r$: The first $S$ operations with the earliest process start time $Earliest\_S\_Operations(\{\pi_r(O)|O \in \mathcal{O}_{plan,r}\})$. See Alg. 3. |
| | $\mathcal{O}_{overlap,r}$ | Overlapping operations in consecutive RHO iterations ($r-1$ and $r$): $\mathcal{O}_{overlap,r} = \mathcal{O}_{plan,r} \cap \mathcal{O}_{plan,r-1}$ |
| | $\mathcal{O}_{new,r}$ | New operations in iteration $r$ but not in $r-1$: $\mathcal{O}_{new,r} = \mathcal{O}_{plan,r} \backslash \mathcal{O}_{plan,r-1}$ |
| | $\mathcal{O}^*_{fix,r}$ | The (look-ahead Oracle) subset of operations in $\mathcal{O}_{overlap,r}$ with the same machine assignment in the consecutive unrestricted RHO subproblems $P_r$ and $P_{r-1}$: $\mathcal{O}^*_{fix,r} = \{O \in \mathcal{O}_{overlap,r} \mid m_r(O) = m_{r-1}(O)\}$ That is, given the solution to the previous subproblem $P_{r-1}$, we lookahead by solving the current subproblem $P_r$ and identify operations with the same machine assignments. |
| | $\mathcal{O}_{fix,r}$ | The fixed operations subset given by a specific method. Given $\mathcal{O}_{fix,r} \subseteq \mathcal{O}_{overlap,r}$, we can construct $\hat{P}_r$ from $P_r$ by setting the compatible machines $M_{j,k} = \{m_{r-1}(O_{j,k})\}$ for operations $O_{j,k} \in \mathcal{O}_{fix,r}$. |
| **Subproblem Solution** $\Pi_r = (m_r, \pi_r)$ | Subproblem Assignment $m_r$ | $m_r : \mathcal{O}_{plan,r} \to \mathcal{M}$ represents the machine assignment of each operation in the subproblem, with $m(O_{j,k}) \in \mathcal{M}_{j,k}$ for all $O_{j,k} \in \mathcal{O}_{plan,r}$. |
| | Subproblem Schedule $\pi_r$ | $\pi_r : \mathcal{O}_{plan,r} \to \mathbb{N}$, where the (solution) start time of each operation in the subproblem $O_{j,k} \in \mathcal{O}_{plan,r}$ is $\pi(O_{j,k}) \geq s_{j,k}$. |
| **Methods** | Default (Cold Start) | Each RHO iteration solves an unrestricted FJSP subproblems $P_r$ from scratch (cold start, $\mathcal{O}_{fix,r} \in \emptyset$). |
| | Random $\sigma_R$ | Each RHO iteration solves a restricted FJSP subproblem $\hat{P}_r$ associated with the fixed operation set $\mathcal{O}_{fix,r}$, constructed by sampling each operation in $\mathcal{O}_{overlap,r}$ with a probability $\sigma_R$ uniformly at random. |
| | First $\sigma_F$ | Each RHO iteration solves a restricted FJSP subproblem $\hat{P}_r$ associated with the fixed operation set $\mathcal{O}_{fix,r}$ obtained as the first $\sigma_F$ fraction of operations in $\mathcal{O}_{overlap,r}$ with the earliest RHO sequence order (determined by operation precedence for makespan or release time for delay-based objectives). |
| | Warm Start | At each RHO iteration, we apply CP-SAT's internal warm-start techniques to solve the unrestricted FJSP subproblems $P_r$. Specifically, we provide the machine assignments of the overlapping operations in the previous iteration $\{m_{r-1}(O) \, \forall \, O \in \mathcal{O}_{overlap,r}\}$ as *hints* to the solver. See: "https://developers.google.com/optimization/reference/python/sat/python/cp_model#addhint". |
| | Oracle | Follow our data collection pipeline (Fig. 1 (b) top, omit the solve time for the unrestricted $P_r$), where each RHO iteration solves a restricted FJSP subproblem $\hat{P}_r$ associated with the *oracle* fixed operation set $\mathcal{O}^*_{fix,r}$. |
| | L-RHO (Ours) | Follow the inference procedure $RHO_{test}$ in Fig. 1 (b) bottom, where each RHO iteration solves a restricted FJSP subproblem $\hat{P}_r$ associated with the fixed operation set $\mathcal{O}_{fix,r}$ *predicted* by the learning model $f_\theta$. |

## A.2 ALGORITHM: LEARNING-GUIDED ROLLING HORIZON OPTIMIZATION

The inference procedure for the learning-guided rolling horizon algorithm (Fig. 1) is provided as follows, where at each RHO iteration, we use the neural network $f_\theta$ to identify a set of fixed operation $\mathcal{O}_{fix,r}$ (Line 10), resulting in the machine assignment-based restricted subproblem $\hat{P}_r$ (Line 15). We note that the heuristic RHO baselines considered in Sec. 5 and 6 (e.g. First $\sigma_F$ and Random $\sigma_R$), we follow the same algorithm but use the associated heuristics to find the set $\mathcal{O}_{fix,r}$ instead (Line 10).

---

**Algorithm 1:** Learning-Guided Rolling Horizon Optimization for FJSP

---

**Input:** FJSP problem instance $P = (\mathcal{M}, \mathcal{T}, \mathcal{O}, \{p_{j,k}^m\}, \{s_{j,k}\}, \{t_{j,k}\})$, RHO planning window size $H$ operations, execution step size $S$ operations, and solve time limit $T$ per subproblem.;

**Output:** Solution $\Pi = (m, \pi)$ of the FJSP problem instance $P$

1   Initialize $m \leftarrow \{\}, \pi \leftarrow \{\}$;

2   **for** *Iteration* $r = 1 : |\mathcal{O}|/S$ **do**

3     `// Obtain first H non-executed operations`

4     $\mathcal{O}_{plan,r} \leftarrow$ GetPlanOperations$(\mathcal{O}, H, r, m, \pi)$;

5     `// Generate FJSP subproblem (see Appendix A.5.2)`

6     $P_r \leftarrow$ FJSP_Subproblem$(P, \mathcal{O}_{plan,r}, m, \pi)$;

7     **if** *Iteration* $r \geq 2$ **then**

8       `// Learn to identify fixed operations in `$\mathcal{O}_{overlap,r}$

9       $\mathcal{O}_{fix,r} \leftarrow f_\theta(P_r, \mathcal{O}_{overlap,r}, P_{r-1}, m_{r-1}, \pi_{r-1})$;

10    **else**

11       $\mathcal{O}_{fix,r} \leftarrow \emptyset$;

12    **end if**

13    `// Obtain the assignment-based restricted FJSP subproblem`

14    $\hat{P}_r \leftarrow$ Restricted_FJSP_Subproblem$(P_r, \mathcal{O}_{fix,r})$;

15    `// Solve the restricted FJSP subproblem`

16    $(m_r, \pi_r) \leftarrow$ FJSP_Solver$(\hat{P}_r; T)$;

17    `// Execute first S operations in `$\mathcal{O}_{plan,r}$` with earliest start times`

18    $\mathcal{O}_{step,r} \leftarrow$ GetStepOperations$(\mathcal{O}_{plan,r}, S, r, m_r, \pi_r)$;

19    `// Fix final solution for executed operations`

20    **for** $O^{(i)} \in \mathcal{O}_{step,r}$ **do**

21       $m(O^{(i)}) \leftarrow m_r(O^{(i)})$;

22       $\pi(O^{(i)}) \leftarrow \pi_r(O^{(i)})$;

23    **end for**

24   **end for**

---

**Algorithm 2:** GetPlanOperations

---

**Input:** Operations
$\mathcal{O} = \{O^{(1)}, ..., O^{(|\mathcal{O}|)}\}$ *ordered based on the precedence or release time (Table 3)*,
RHO planning window size ($H$ operations), RHO current iteration $r$, executed solutions $(m, \pi)$

**Output:** Operations in the current RHO planning window $\mathcal{O}_{plan,r}$

1   $\mathcal{O}_{plan,r} \leftarrow []; i \leftarrow 0$

2   `// select the first H non-executed operations based on the sorted order`

3   **while** $|\mathcal{O}_{plan,r}| < H$ *and* $i < |\mathcal{O}|$ **do**

4     **if** $O^{(i)} \notin (m, \pi)$ **then**

5       $\mathcal{O}_{plan,r}.append(O^{(i)})$

6     **end if**

7     $i \leftarrow i + 1$

8   **end while**

---

---

**Algorithm 3:** GetStepOperations

---

**Input:** Operations in the current window $\mathcal{O}_{plan,r}$, RHO execution step size ($S$ operations),
    RHO current iteration $r$, solution $\Pi_r = (m_r, \pi_{s,r})$ of the $r^{th}$ FJSP subproblem
**Output:** Operations executed after the $r^{th}$ RHO iteration $\mathcal{O}_{step,r}$

1 // select the first $S$ non-executed operations with the *earliest*
   *process start time*

2 $\tilde{\mathcal{O}}_{plan,r} \leftarrow$ Sort $\mathcal{O}_{plan,r}$ by the process start time $\{\pi_{s,r}(O^{(i)})\}_{O^{(i)} \in \mathcal{O}_{plan,r}}$

3 $\mathcal{O}_{step,r} \leftarrow \{\tilde{\mathcal{O}}_{plan,r}^{(1)}, ..., \tilde{\mathcal{O}}_{plan,r}^{(S)}\}$

---

### A.3 INPUT FEATURES.

This section provides a detailed description of all input features, which can be categorized as the following three main parts:

(1) For each operation in $\mathcal{O}_{plan,r} = \mathcal{O}_{overlap,r} \cup \mathcal{O}_{new,r}$, we design input features to capture the FJSP instance information such as the earliest process start time of the operation (considering previously executed operations within the same job), the duration (mean / min / max / std) to complete the operation by the compatible machines, and the operation's release time (for delay-based objectives);

(2) For each operation in $\mathcal{O}_{overlap,r}$, we design additional input features to encode the solution information from the previous RHO iteration, including the duration, machine assignment of the operation and end time / delay.

(3) For each machine in $\mathcal{M}$, we further design input features that include the earliest process start time of the machine (considering previously executed operations by the machine), and the duration (mean / min / max / std) of the overlapping operations $\mathcal{O}_{overlap,r}$ previously assigned to the machine.

As we consider the following different FJSP scenarios in our experiments (Sec. 5), we adjust the input features accordingly as follows

- **Makespan objective without observation noise**: we use all input features as listed in Table 5.

- **Makespan objective under machine breakdowns**: we use all input features as listed in Table 5, and the additional input features as listed in Table 6 to incorporate the machine breakdown information.

- **Total Start Delay objective without observation noise**: we use all input features as listed in Table 7, where the delay input feature is the start time delay for each operation based on the previous RHO iteration's solution, i.e. $\pi_{r-1}(O_{j,k}) - s_{j,k}$.

- **Total Start and End Delay objective without observation noise**: we use all input features as listed in Table 7, where the delay input feature is the start and end time delay for each operation based on the previous RHO iteration's solution, i.e. $\pi_{r-1}(O_{j,k}) - s_{j,k} + \max(\pi_{t,r-1}(O_{j,k}) - t_{j,k}, 0)$ where $\pi_{r-1}(O_{j,k}) = \pi_{r-1}(O_{j,k}) + p_{j,k}^{m_{r-1}(O_{j,k})}$. We further consider the end time related input features as listed in Table 8.

- **Total Start and End Delay objective with observation noise**: we use all input features in Table 7 and 8. Additionally, as described in Appendix A.5, we obtain noisy observations of each operation's duration at each RHO iteration (the observation of the same operation may vary across different RHO iterations). That is, given the same machine assignment and start time $(\tilde{m}_{r-1}, \tilde{\pi}_{r-1})$ from the previous ($r-1^{th}$) iteration, the observed duration and end time for each overlapping operation can differ between the previous ($r - 1^{th}$) and current ($r^{th}$) iteration. As a result, we compute the delay and end time delay in Table 7 and 8 using the noisy observation from the previous RHO iteration, and we further re-evaluate the delay, duration, and end time for the overlapping operations based on the current RHO iteration's noisy observation, as detailed in Table 9.

We normalize all input features with mean and standard deviation computed on the training set $\mathcal{K}_{data}$.

Table 5: Input Features for the Operations and Machines for the makespan objective.

| Type | Feature | Description |
|---|---|---|
| **Operations** | job_start_time | The earliest start time of the job $j$ associated with the operation $O_{j,k}$, given by ... 
 ... the latest process end time of all executed operations for the same job |
| | avg_dur | Average duration of all compatible machine assignments for the operation |
| | std_dur | Duration standard deviation of all compatible machine assignments for the operation |
| | min_dur | Minimum duration of all compatible machine assignments for the operation |
| | max_dur | Maximum duration of all compatible machine assignments for the operation |
| | job_id | Job id for the operation ($j \in \{1, ...|\mathcal{T}|\}$ before normalization) |
| | ops_id | Operation id for the operation ($k \in \{1, ...n_j\}$ before normalization) |
| | in_overlap | The operation $O_{j,k} \in \mathcal{O}_{overlap,r}$ at the current RHO iteration $r$ |
| | prev_machine 
 (-1 if not in_overlap) | Machine assignment of the operation in the solution of... 
 ... the previous RHO iteration ($m(O_{j,k}) \in \{0, ...|\mathcal{M} - 1|\}$ before normalization) |
| | prev_duration 
 (-1 if not in_overlap) | Operation's process duration given by the previous machine assignment $p_{j,k}^{m_{r-1}(O_{j,k})}$ |
| | prev_end_time 
 (-1 if not in_overlap) | End time of the operation in the solution of ... 
 ... the previous RHO iteration ($\pi_{t,r-1}(O_{j,k})$) |
| | alt_avg_dur 
 (-1 if not in_overlap) | Average process duration of all other machines .. 
 ... (not prev_machine) to process the operation |
| | alt_std_dur 
 (-1 if not in_overlap) | Standard deviation of duration of all other machines ... 
 ... (not prev_machine) to process the operation |
| | alt_min_dur 
 (-1 if not in_overlap) | Minimum duration of all other machines ... 
 ... (not prev_machine) to process the operation |
| | alt_max_dur 
 (-1 if not in_overlap) | Maximum duration of all other machines ... 
 ... (not prev_machine) to process the operation |
| **Machines** | machine_start_time | The earliest start time of the machine $m$, given by the latest process ... 
 ... end time of all executed operations assigned to machine $m$ |
| | num_overlap | Number of overlapping operations assigned to the machine $m$ ... 
 ... in the solution of the previous RHO iteration $\sum_{O \in \mathcal{O}_{overlap,r}} \mathbb{1}\{m_{r-1}(O) = m\}$ |
| | avg_end_time 
 (-1 if num_overlap is 0) | Average end time of overlapping operations assigned to the machine ... 
 ... in the solution of the previous RHO iteration |
| | std_end_time 
 (-1 if num_overlap is 0) | Standard deviation of the end time of overlapping operations assigned to ... 
 ... the machine in the solution of the previous RHO iteration |
| | max_end_time 
 (-1 if num_overlap is 0) | Maximum end time of overlapping operations assigned to the machine ... 
 ... in the solution of the previous RHO iteration |
| | min_end_time 
 (-1 if num_overlap is 0) | Minimum end time of overlapping operations assigned to the machine ... 
 ... in the solution of the previous RHO iteration |
| | avg_duration 
 (-1 if num_overlap is 0) | Average duration of overlapping operations assigned to the machine ... 
 ... in the solution of the previous RHO iteration |
| | std_duration 
 (-1 if num_overlap is 0) | Standard deviation of the duration of overlapping operations assigned to ... 
 ... the machine in the solution of the previous RHO iteration |
| | max_duration 
 (-1 if num_overlap is 0) | Maximum duration of overlapping operations assigned to the machine ... 
 ... in the solution of the previous RHO iteration |
| | min_duration 
 (-1 if num_overlap is 0) | Minimum duration of overlapping operations assigned to the machine ... 
 ... in the solution of the previous RHO iteration |

Table 6: Additional Input Features for the Machine Breakdown.

| Type | Feature | Description |
|---|---|---|
| **Operations** | is_break_down | The current RHO subproblem has some machine breakdown |
| | is_ops_break_down | The machine assigned to the operation in the previous RHO subproblem ... 
 ... is broken down in the current RHO subproblem |
| | is_ops_recovered | The operation's machine assignment choices was restricted ... 
 ... in the previous RHO subproblem due to machine breakdown, ... 
 ... but the breakdown is recovered in the current subproblme |
| **Machines** | is_break_down | The current RHO subproblem has some machine breakdown |
| | is_machine_break_down | The machine itself is broken down in the current RHO subproblem |
| | prev_start 
 (-1 if not in_overlap) | Process start time of the operation in the solution of ... 
 ... the previous RHO iteration $\pi_{r-1}(O_{j,k})$ |
| | prev_end 
 (-1 if not in_overlap) | Process end time of the operation in the solution of ... 
 ... the previous RHO iteration $\pi_{t,r-1}(O_{j,k}) = \pi_{r-1}(O_{j,k}) + p_{j,k}^{m_{r-1}(O_{j,k})}$ |

Table 7: Input Features for the Operations and Machines for delay-based objectives.

| Type | Feature | Description |
|---|---|---|
| **Operations** | job_start_time | The earliest start time of the job $j$ associated with the operation $O_{j,k}$, given by ... ... the latest process end time of all executed operations for the same job |
| | ops_release_time | The release time $s_{j,k}$ of the operation $O_{j,k}$ |
| | avg_dur | Average duration of all compatible machine assignments for the operation |
| | std_dur | Duration standard deviation of all compatible machine assignments for the operation |
| | min_dur | Minimum duration of all compatible machine assignments for the operation |
| | max_dur | Maximum duration of all compatible machine assignments for the operation |
| | job_id | Job id for the operation ($j \in \{1,...|\mathcal{T}|\}$ before normalization) |
| | ops_id | Operation id for the operation ($k \in \{1,...n_j\}$ before normalization) |
| | in_overlap | The operation $O_{j,k} \in \mathcal{O}_{overlap,r}$ at the current RHO iteration $r$ |
| | prev_delay (-1 if not in_overlap) | Delay of the operation in the solution of ... ... the previous RHO iteration ($\pi_{r-1}(O_{j,k}) - s_{j,k}$) |
| | prev_machine (-1 if not in_overlap) | Machine assignment of the operation in the solution of... ... the previous RHO iteration ($m(O_{j,k}) \in \{0,...|\mathcal{M}-1|\}$ before normalization) |
| | prev_duration (-1 if not in_overlap) | Operation's process duration given by the previous machine assignment $p_{j,k}^{m_{r-1}(O_{j,k})}$ |
| | alt_avg_dur (-1 if not in_overlap) | Average duration of all other machines ... ... (not prev_machine) to process the operation |
| | alt_std_dur (-1 if not in_overlap) | Standard deviation of duration of all other machines ... ... (not prev_machine) to process the operation |
| | alt_min_dur (-1 if not in_overlap) | Minimum duration of all other machines ... ... (not prev_machine) to process the operation |
| | alt_max_dur (-1 if not in_overlap) | Maximum duration of all other machines ... ... (not prev_machine) to process the operation |
| **Machines** | machine_start_time | The earliest start time of the machine $m$, given by the latest process ... ... end time of all executed operations assigned to machine $m$ |
| | num_overlap | Number of overlapping operations assigned to the machine $m$ ... ... in the solution of the previous RHO iteration $\sum_{O\in\mathcal{O}_{overlap,r}} \mathbb{1}\{m_{r-1}(O)=m\}$ |
| | avg_delay (-1 if num_overlap is 0) | Average delay of overlapping operations assigned to the machine ... ... in the solution of the previous RHO iteration |
| | std_delay (-1 if num_overlap is 0) | Standard deviation of the delay of overlapping operations assigned to ... ... the machine in the solution of the previous RHO iteration |
| | max_delay (-1 if num_overlap is 0) | Maximum delay of overlapping operations assigned to the machine ... ... in the solution of the previous RHO iteration |
| | min_delay (-1 if num_overlap is 0) | Minimum delay of overlapping operations assigned to the machine ... ... in the solution of the previous RHO iteration |

Table 8: Additional Input Features for the Total Start and End Delay Objective.

| Type | Feature | Description |
|---|---|---|
| **Operations** | ops_target_due_time | The target due time $t_{j,k}$ of the operation $O_{j,k}$ |
| | prev_start (-1 if not in_overlap) | Process start time of the operation in the solution of ... ... the previous RHO iteration $\pi_{r-1}(O_{j,k})$ |
| | prev_end (-1 if not in_overlap) | Process end time of the operation in the solution of ... ... the previous RHO iteration $\pi_{t,r-1}(O_{j,k}) = \pi_{r-1}(O_{j,k}) + p_{j,k}^{m_{r-1}(O_{j,k})}$ |

Table 9: Additional Input Features for Observation Noise: given the solution from the previous RHO iteration, we *re-evaluate* the delay, duration, end time of the overlapping operation based on the noisy observation at the current RHO iteration.

| Type | Feature | Description |
|---|---|---|
| **Operations** | prev_duration_reeval (-1 if not in_overlap) | Currently observed process duration of the operation $O$ ... 
 ... (given the machine assignment from the previous RHO iteration) |
| | prev_end_reeval (-1 if not in_overlap) | Process end time of the operation (given the machine ... 
 ... assignment from the previous RHO iteration), 
 ... *reevaluated* based on the current observed process duration |
| | prev_delay_reeval (-1 if not in_overlap) | Delay of the operation (given the machine ... 
 ... assignment from the previous RHO iteration) ... 
 ... , *reevaluated* based on the current observed process duration |
| **Machines** | avg_delay_reeval (-1 if num_overlap is 0) | Average delay of overlapping operations assigned to the machine ... 
 ... in the solution of the previous RHO iteration, ... 
 ... , *reevaluated* based on the current observed process duration |
| | std_delay_reeval (-1 if num_overlap is 0) | Standard deviation of the delay of overlapping operations assigned to ... 
 ... the machine in the solution of the previous RHO iteration, ... 
 ... , *reevaluated* based on the current observed process duration |
| | max_delay_reeval (-1 if num_overlap is 0) | Maximum delay of overlapping operations assigned to the machine ... 
 ... in the solution of the previous RHO iteration, ... 
 ... , *reevaluated* based on the current observed process duration |
| | min_delay_reeval (-1 if num_overlap is 0) | Minimum delay of overlapping operations assigned to the machine ... 
 ... in the solution of the previous RHO iteration ... 
 ... , *reevaluated* based on the current observed process duration |
| | machine_end_time (-1 if num_overlap is 0) | Maximum process end time of overlapping operations ... 
 ... assigned to the machine in the solution of the previous RHO iteration, ... 
 ... , based on the previous iteration's observed process duration |
| | machine_end_time_reeval (-1 if num_overlap is 0) | Maximum process end time of overlapping operations ... 
 ... assigned to the machine in the solution of the previous RHO iteration, ... 
 ... , *reevaluated* based on the current observed process duration |

Table 10: **Architecture hyperparameters.** The input dimensions $(d_o, d_m) = (15, 11), (18, 13), (16, 6), (19, 6), (22, 12)$ for the five settings detailed in Table 5, 6, 8 and 9.

| Input Dimension | | MLP Layers | 2 |
|---|---|---|---|
| Operation $|\mathcal{O}_{plan,r}| \times d_o$ | $\mathbb{R}^{(H-S) \times d_o}$ | Activation | ReLU |
| Machine $|\mathcal{T}| \times d_m$ | $\mathbb{R}^{|\mathcal{T}| \times d_m}$ | | |
| Embedding dimension $d_{hidden}$ | 64 | Output Dimension | $\mathcal{R}^{(H-S) \times 1}$ |

Table 11: **Training hyperparameters.**

| Optimizer | Adam |
|---|---|
| Learning rate | $10^{-3}$ |
| Batch size | 64 |
| Positive Label Weight $w_{pos}$ | 0.5 |
| Num. of Gradient Steps | $5 \times 10^5$ |

## A.4 ARCHITECTURE, TRAIN AND EVALUATION SETUP.

We use the neural architecture $f_\theta$ in Fig. 2 to identify the fixed operation subset $\mathcal{O}_{fix,r}$. The network takes as input a set of input features associated with each operation and machine, with the aggregated size $\mathbb{R}^{|\mathcal{O}_{plan,r}| \times d_o}$ and $\mathbb{R}^{|\mathcal{T}| \times d_m}$, and outputs the predicted probability $f_\theta(\mathbf{s}_r) \in [0,1]^{|\mathcal{O}_{overlap,r}|}$ of whether each overlapping operation $O \in \mathcal{O}_{overlap,r}$ should be included in $\mathcal{O}_{fix,r}$. The architecture consists of the following four main components:

(1) **Input Embedding**: We first embed the input vector of the operations $\mathcal{O}_{plan,r} = \mathcal{O}_{overlap,r} \cup \mathcal{O}_{new,r}$ and machines $\mathcal{T}$, each with a input dimension of $\mathbb{R}^{d_o}$ and $\mathbb{R}^{d_m}$, into hidden representations, each with a hidden dimension of $\mathbb{R}^{d_{hidden}}$. We use two separate MLPs for the operations and machines.

(2) **Machine/Operation Concatenation**: We then concatenate the hidden feature of each operation $O \in \mathcal{O}_{plan,r}$ with the hidden feature of the previously assigned machine $m_{r-1}(O)$ (the local feature), and a global feature obtained by a mean pooling on the hidden features from all operations and machines. The concatenation results in a hidden dimension of $\mathbb{R}^{3 \cdot d_{hidden}}$ for each operation, and we apply another MLP to project the dimension down to $\mathbb{R}^{d_{hidden}}$.

(3) **Output**: Lastly, we pass the hidden feature of each overlapping operation $\mathcal{O}_{overlap,r}$ through another MLP, which outputs the predicted probability that each operation $O \in \mathcal{O}_{overlap,r}$ should be included in $\mathcal{O}_{fix,r}$.

We train the proposed architecture with Adam optimizer with a learning rate of $1e^{-3}$ and a batch size of 64 for 500 epochs with around $5 \times 10^5$ gradient steps. All hyperparameters are selected on the validation set and frozen before evaluating on the test set. Table 10 and 11 provide a list of hyperparameters.

**Training and Evaluation Setup.** For each FJSP setting, we collect a small set $\mathcal{K}_{ps}$ of 10 instances for parameter search on all RHO methods (default, baseline and learning), and a large training set $\mathcal{K}_{train}$ of 450 instances for neural network training. By default, we hold out a validation set $\mathcal{K}_{val}$ of 20 instances and a test set $\mathcal{K}_{test}$ of 100 instances each. We collect data, train, validate and test all methods on a distributed compute cluster using nodes equipped with 48 Intel AVX512 CPUs. A Nvidia Volta V100 GPU is used to train all neural networks. The model training time is within 12 hours for all FJSP settings. The parameter search for *all* RHO methods (Appendix A.5.4) requires approximately 48-96 hours for each FJSP setting in the experiment Sec. 5, based on the problem size. Furthermore, as the FJSP constraint programming solver CP-SAT allows multi-processing, we use 12 CPUs when solving each RHO subproblem ($P_r$ / $\hat{P}_r$) for all RHO methods (Default, Baselines and Learning) for both training and testing; we also use 12 CPUs to evaluate the Full FJSP baseline. When collecting training labels, we solve each unrestricted subproblem $P_r$ five times ($Q = 5$); the training data collection takes around 24-72 hours for each FJSP setting based on the problem size. In the experiment section, we report the solve time for different methods on the test set $\mathcal{K}_{test}$ (Table 1, Table 2, Fig. 3, and Appendix A.6).

## A.5 EXPERIMENTAL DETAILS

### A.5.1 FJSP INSTANCE DISTRIBUTION, MACHINE BREAKDOWN AND OBSERVATION NOISE.

**Makespan-based Data Distribution.**

We follow the a similar data distribution as in the previous learning literature (Wang et al., 2023a) but significantly increase the FJSP problem sizes from 200 operations (previously) to up to 2000 operations (this work). Each FJSP instance is generated by sampling the following attributes:

- $(n_{machines}, n_{jobs}, n_{operations\_per\_job}) = (|\mathcal{M}|, |\mathcal{T}|, |\mathcal{O}/\mathcal{T}|)$: we fix $|\mathcal{M}| = 10, |\mathcal{T}| = 20$ and consider a range of $n_{operations\_per\_job} \in \{30, 40, 60, 100\}$. That is, we have $(|\mathcal{M}|, |\mathcal{T}|, |\mathcal{O}/\mathcal{T}|) \in \{(10, 20, 30), (10, 20, 40), (10, 20, 60), (10, 20, 100)\}$, with the total number of operations ranging from 600 to 2000.
- Compatible Machines $\mathcal{M}_{j,k}$: For each operation $O_{j,k} \in \mathcal{O}$, we randomly select a subset of machines $\mathcal{M}_{j,k} \subseteq \mathcal{M}$ with size $|\mathcal{M}_{j,k}| \sim \text{Uniform}\{1, ..., |\mathcal{M}|\}$ as the compatible machines for the operation.
- Process Duration $p_{j,k}^m$: we sample the duration of operation $O_{j,k} \in \mathcal{O}$ processed by machine $m \in \mathcal{M}$ from a uniform distribution $U[1, 99]$.

**Machine Breakdown.** As a step towards online / dynamics FJSP, ours experiments in Fig. 3 (right) accounts for machine breakdowns during the RHO solution process. The intensity of machine breakdowns is determined by two factors: 1) the frequency of breakdowns 2) the proportion of unavailable machines during each breakdown. For each FJSP instance, we simulate a series of machine breakdown events represented by a set of time intervals $\{t_i^b\}_{i=1,2,...}$. For each breakdown event, we randomly sample a subset of machines $\mathcal{M}_i^b \subseteq \mathcal{M}$, which become unavailable during the event. Specifically, the first breakdown event starts at time $t_0^b \sim U[50, 150]$ and lasts for a duration of $dur$. Each subsequent breakdown occurs at time $t_i^b \sim t_{i-1}^b + dur + U[w_{lb}, w_{ub}]$ and similarly lasts for a duration of $dur$. We sample the subset of breakdown machines $\mathcal{M}_i^b$ by selecting each machine in $\mathcal{M}$ i.i.d. with a probability $p^b$. In our experiments, we explore three levels of breakdown intensities : 1) **Low**: we set $dur = 100, w_{lb} = 400, w_{ub} = 600, p^b = 0.2$, 2) **Mid**: we set $dur = 100, w_{lb} = 175, w_{ub} = 300, p^b = 0.35$, and 3) **High**: we set $dur = 50, w_{lb} = 100, w_{ub} = 200, p^b = 0.5$.

Following dynamic FJSP literature (Zhang and Wong, 2017), we combine periodic RHO subproblem optimization with *event-based rescheduling*: when a machine breakdown event starts or ends, we re-optimize the updated RHO subproblem using the updated machine availability. This event-based rescheduling affects (1) the RHO subproblem step size, which may be reduced due to breakdown-triggered re-optimizations. Specifically, for each RHO iteration, we execute the set of operations $\mathcal{O}_{exec,r}$ in order of their scheduled end times. If a breakdown occurs during an operation's execution window, we halt that operation and any subsequent ones, and reformulate the RHO subproblem with the updated set of the first $W$ unexecuted operations, incorporating the updated machine availability for re-optimization. (2) If all compatible machines for an operation are down, we defer that operation, along with any subsequent operations in the same job, until the breakdown ends.

Implementation-wise, (1) is formally described in Alg. 5, which replaces Alg. 3 for the RHO operation execution step; (2) is formally described in Alg. 4, which replaces Alg. 2 to construct each RHO subproblem.

**Delay-based Data Distribution.** Following common FJSP instance generation procedure in the literature (Behnke and Geiger, 2012), we generate each FJSP instance by sampling the following attributes:

- $(n_{machines}, n_{jobs}, n_{operations\_per\_job}) = (|\mathcal{M}|, |\mathcal{T}|, |\mathcal{O}/\mathcal{T}|)$: we consider a series of large scale settings $\{(25, 25, 24), (30, 30, 24), (35, 35, 30), (40, 40, 40)\}$, with the total number of operations ranging from 600 to 1600. We further set the compatible machines $\mathcal{M}_{j,k} = \mathcal{M}$ for all operations $O \in \mathcal{O}$.
- Process Duration $p_{j,k}^m$: we sample the duration of operation $O_{j,k} \in \mathcal{O}$ processed by machine $m \in \mathcal{M}$ from a uniform distribution $U[l_{low}, l_{high}]$, with $l_{low} \sim \text{Uniform}(\{3, 5, 7, 9\})$ and $l_{high} \sim l_{low} + \text{Uniform}(\{9, 12, 15, 18, 21\})$, resulting in the duration range $p_{j,k}^m \in [3, 30]$;

- Release time: we sample the release time for all operations $\{O_{j,k}\}_{k=1}^{n_j}$ within each job $j$ as $s_{j,1} \sim U([0,15])$ and $s_{j,k} = s_{j,k-1} + U([0,15])$ for all $k > 1$;
- Target end time: we sample the target end time for each operation $O_{j,k}$ as $t_{j,k} \sim s_{j,k} + U([0,30])$; this attribute is used when the objective contains total end delay (Sec. 5.2).

**Observation noise.** In the observation noise setting (Sec. 5.2), we have noisy observation of the process duration $p_{j,k}^m$. Specifically, we generate the clean (true) process duration $p_{j,k}^m$ for all operations and machines (which is only partially observed) following the procedure described above. Then, at each RHO iteration, we perturb process duration as follows: consider the operations $\mathcal{O}_{plan,r} = \{O^{(1)},...,O^{(H)}\}$ (ordered by their release times) in the current iteration's planning horizon. We assume perfect observation of the first $S \leq H$ operations with earlier release times $\mathcal{O}_{clean,r} = \{O^{(1)},...,O^{(S)}\}$, and we have noisy observation for a randomly selected subset $\mathcal{O}_{perturb,r}$ of the remaining operations with later release times, where we select each operation in $\mathcal{O}_{plan,r} \backslash \mathcal{O}_{clean,r}$ to be included in $\mathcal{O}_{perturb,r}$ uniformly at random with a probability of $\epsilon = 20\%$. We then set the noisy observation of the process duration for each operation $O_{j,k} \in \mathcal{O}_{plan,r}$ by machine $m \in \mathcal{M}_{j,k}$ at the current RHO iteration $r$ as:

$$\tilde{p}_{j,k}^{m,r} = \begin{cases} p_{j,k}^m & \text{if operation } O_{j,k} \in \mathcal{O}_{clean,r} \\ \text{clip}(p_{j,k}^m + U[-5,5], 3, 30) & \text{if operation } O_{j,k} \in \mathcal{O}_{perturb,r}. \end{cases}$$

Notably, we may have different noisy observations for the same overlapping operation in different RHO iterations. That is, we may have $\tilde{p}_{j,k}^{m,r-1} \neq \tilde{p}_{j,k}^{m,r}$ for the same operation $O_{j,k}$ and machine $m$.

**Operation execution under observation noise.** After obtaining the solutions $(\tilde{m}_r, \tilde{\pi}_r)$ to the noisy FJSP subproblem, we execute a subset of $S$ operation and update the final full FJSP solution $(m, \pi)$ as follows (replacing Line 17-21 in Alg. 1 with the procedure in Alg. 6): We first determine $\mathcal{O}_{step,r}$ as the $S$ operations with the earliest process start time according to the noisy solution $\tilde{\pi}_r$. Then, for each operation $O_{j,k} \in \mathcal{O}_{step,r}$ (ordered by the solution start time $\tilde{\pi}_r(O_{j,k})$), we assign the operation to the machine $m(O_{j,k}) := \tilde{m}_r(O_{j,k})$, which takes a *true duration* of $p_{j,k}^{\tilde{m}_r(O_{j,k})}$ process the operation. Due to discrepancy between observed duration at planning and the actual duration at execution, the operation may not be able to start at its solution start time $\tilde{\pi}_r(O_{j,k})$ given by the solution. We hence further adjust the actual start time $\pi(O_{j,k})$ at execution to maintain solution feasibility. The detailed execution procedure is provided in Alg. 6.

---

**Algorithm 4:** GetPlanOperations Under Machine Breakdown

**Input:** Operations $\mathcal{O} =$
    $\{O^{(1)},...,O^{(|\mathcal{O}|)}\}$ *ordered based on the precedence order or release time (Table 3)*,
    RHO planning window size ($H$ operations), RHO current iteration $r$, executed solutions
    $(m, \pi)$
**Output:** Operations in the current RHO planning window $\mathcal{O}_{plan,r}$

1 $\mathcal{O}_{plan,r} \leftarrow []; i \leftarrow 0$
2 $Ignored\_Jobs = Set()$
3 // select the first $H$ non-executed operations based on the
    sorted order
4 **while** $|\mathcal{O}_{plan,r}| < H$ *and* $i < |\mathcal{O}|$ **do**
5     **if** $O^{(i)} \notin (m, \pi)$ **then**
6         **if** *All compatible machines of* $O^{(i)}$ *are down* **then**
7             $Ignored\_Jobs.add(Job\_of\_O^{(i)}))$
8             continue
9         **end if**
10         **if** $Job\_of\_O^{(i)} \in Ignored\_Jobs$ **then**
11             continue
12         **end if**
13         $\mathcal{O}_{plan,r}.append(O^{(i)})$
14     **end if**
15     $i \leftarrow i+1$
16 **end while**

---

---

**Algorithm 5:** Execute Operations Under Machine Breakdown

---

**Input:** Operations in the current window $\mathcal{O}_{plan,r}$, RHO execution step size ($S$ operations), RHO current iteration $r$, solution $\Pi_r = (m_r, \pi_{s,r})$ of the $r^{th}$ FJSP subproblem

**Output:** Operations executed after the $r^{th}$ RHO iteration $\mathcal{O}_{step,r}$

1   $\tilde{\mathcal{O}}_{plan,r} \leftarrow$ Sort $\mathcal{O}_{plan,r}$ by the process start time $\{\pi_{s,r}(O^{(i)})\}_{O^{(i)} \in \mathcal{O}_{plan,r}}$

2   $\mathcal{O}_{step,r} \leftarrow []$

3   **for** $O_{j,k} \in \mathcal{O}_{step,r}$ *(sorted by the solution process start time)* **do**

4      **if** *machine breakdown starts or ends during the operation's execution window* $[\pi(O_{j,k}), \pi_t(O_{j,k})]$ **then**

5          `// End the current RHO subproblem.  Do not execute any more operations.`

6          break

7      **end if**

8      **else**

9          $\mathcal{O}_{step,r}.append(O_{j,k})$

10     **end if**

11 **end for**

---

**Algorithm 6:** Execute Operations Under Observation Noise

---

**Input:** Operations in the current window $\mathcal{O}_{plan,r}$; RHO execution step size ($S$ operations); RHO current iteration $r$; (partial) solution $\Pi = (m, \pi)$ of the full FJSP problem instance $P$ before the current iteration $r$; solution $\tilde{\Pi}_r = (\tilde{m}_r, \tilde{\pi}_r)$ of the $r^{th}$ FJSP subproblem under the noisy observation; clean process duration $\{p_{j,k}^m\}_{\substack{m \in \mathcal{M} \\ O_{j,k} \in \mathcal{O}_{overlap,r}}}$

**Output:** Updated (partial) solution $\Pi = (m, \pi)$ of the FJSP problem instance $P$ after the current RHO iteration $r$ based on the clean process duration

1   `// Find executed operations for the current RHO iteration, using` $(\tilde{m}_r, \tilde{\pi}_r)$

2   $\tilde{\mathcal{O}}_{plan,r} \leftarrow$ Sort $\mathcal{O}_{plan,r}$ by the process start time $\{\tilde{\pi}_r(O^{(i)})\}_{O^{(i)} \in \mathcal{O}_{plan,r}}$

3   $\mathcal{O}_{step,r} \leftarrow \{\tilde{\mathcal{O}}_{plan,r}^{(1)}, ..., \tilde{\mathcal{O}}_{plan,r}^{(S)}\}$

4   `// Auxiliary variables to track solution feasibility`

5   machine_end_time $\leftarrow \{alt: \max\limits_{\substack{O_{j,k} \in (m,\pi), \\ m(O_{j,k})=alt}} \pi_t(O_{j,k})$    for *each machine* $alt \in \mathcal{M}\}$

6   job_end_time $\leftarrow \{j: \max\limits_{O_{j,k} \in (m,\pi)} \pi_t(O_{j,k})$    for *each job* $j \in \mathcal{T}\}$

7   `// Fix the final solution for the executed operations based on the clean process duration`

8   **for** $O_{j,k} \in \mathcal{O}_{step,r}$ *(sorted by the solution process start time)* **do**

9      $m(O_{j,k}) \leftarrow \tilde{m}_r(O_{j,k})$

10     `// Update the process start time to the earliest feasible start time`

11     $\pi(O_{j,k}) \leftarrow \max(\tilde{\pi}_r(O_{j,k}), \text{machine\_end\_time}[m(O_{j,k})], \text{job\_end\_time}[j])$

12     `// Re-evaluate the process end time based on the clean process duration`

13     $\pi_t \leftarrow \pi(O_{j,k}) + p_{j,k}^{m(O_{j,k})}$

14     `// Update the auxiliary variables to track feasibility`

15     machine_end_time$[m(O_{j,k})] \leftarrow \max\big(\text{machine\_end\_time}[m(O_{j,k})], \pi_t\big)$

16     job_end_time$[j] \leftarrow \max\big(\text{job\_end\_time}[j], \pi_t\big)$

17 **end for**

---

A.5.2   FJSP FORMULATION

**Decision Variables.** Given a FJSP problem instance

$$P = (\mathcal{M}, \mathcal{T}, \mathcal{O}, \text{ process duration } \{p_{j,k}^m\}_{\substack{O_{j,k} \in \mathcal{O} \\ m \in \mathcal{M}_{j,k}}},$$

$$\text{(optional) release time } \{s_{j,k}\}_{O_{j,k} \in \mathcal{O}}, \tag{2}$$

$$\text{(optional) target end time } \{t_{j,k}\}_{O_{j,k} \in \mathcal{O}})$$

The constraint programming formulation for the FJSP is obtained by introducing additional boolean variables denoting whether an operation is assigned to a machine[1]. That is, we define the following set of decision variables

$$X = \Big( \{\pi(O_{j,k})\}_{O_{j,k} \in \mathcal{O}}, \{dur(O_{j,k})\}_{O_{j,k} \in \mathcal{O}}, \{\pi_t(O_{j,k})\}_{O_{j,k} \in \mathcal{O}}, \text{ // process start time, duration, and end time}$$

$$\{l_{alt}(O_{j,k})\}_{\substack{O_{j,k} \in \mathcal{O} \\ alt \in \mathcal{M}_{j,k}}}, \quad \text{// boolean variable: whether } O_{j,k} \text{ is assigned to machine } alt$$

$$\{\pi_{alt}(O_{j,k})\}_{\substack{O_{j,k} \in \mathcal{O} \\ alt \in \mathcal{M}_{j,k}}}, \quad \text{// process start time if } O_{j,k} \text{ is assigned to machine } alt$$

$$\{dur_{alt}(O_{j,k})\}_{\substack{O_{j,k} \in \mathcal{O} \\ alt \in \mathcal{M}_{j,k}}}, \quad \text{// process duration if } O_{j,k} \text{ is assigned to machine } alt$$

$$\{\pi_{t,alt}(O_{j,k})\}_{\substack{O_{j,k} \in \mathcal{O} \\ alt \in \mathcal{M}_{j,k}}} \Big) \quad \text{// process end time if } O_{j,k} \text{ is assigned to machine } alt$$

$$\tag{3}$$

where the target end time and the decision variables related to the process end time are only used when the objective contains total end delay. From the above decision variables, the machine assignment of operation $O_{j,k}$ can be found as $m(O_{j,k}) = alt \in \mathcal{M}_{j,k}$ wher $l_{alt}(O_{j,k}) = 1$, and the process start time of the operation can be found as $\pi(O_{j,k})$.

**FJSP Full and Subproblem Formuation.** Given the above decision variables, the Constraint Programming (CP) formulation for the full FJSP problem $P$ is given in Alg. 7.

- **Unrestricted Subproblem** $P_r$: Given the set of operations in the current planning horizon $\mathcal{O}_{plan,r}$, we can obtain the the subproblem instance $P_r$ and the set of decision variables $X_r$ associated with the restricted operation set $\mathcal{O}_{plan,r}$. Then we use the same CP formulation as in Alg. 7 on the decision variables $X_r$; we additionally introduce new constraints in Alg. 8, which requires operations in $\mathcal{O}_{plan,r}$ to be processed after all previously executed operations within the same job or the same machine.

- **Assignment-Based Restricted Subproblem** $\hat{P}_r$: Given a set of operations in the current planning horizon $\mathcal{O}_{plan,r}$, we follow the same CP formulation in Alg. 7 and 8 as for the unrestricted subproblem $P_r$. Given the fixed operations $\mathcal{O}_{fix,r} \subseteq \mathcal{O}_{overlap,r}$, we further restrict the compatible machines for each fixed operation as the single machine from the previous RHO iteration's solution, i.e. we restrict $\mathcal{M}_{j,k} = \{m_{r-1}(O_{j,k})\}$ for all operations $O_{j,k} \in \mathcal{O}_{fix,r}$.

---

[1]Our implementation is based on the CP-SAT's Official FJSP implementation `https://github.com/google/or-tools/blob/stable/examples/python/flexible_job_shop_sat.py`.

---

**Algorithm 7:** FJSP Constraint Programming Formulation

---

1 **Decision Variables:** $X$ (see Eq. equation 3)

2 **Objective:**

$$\text{Makespan:} \quad \min_X \max_{O_{j,k} \in \mathcal{O}} \pi_t(O_{j,k})$$

$$\text{Total start delay:} \quad \min_X \sum_{O_{j,k} \in \mathcal{O}} \pi(O_{j,k}) - s_{j,k} \quad (4)$$

$$\text{Total start and end delay:} \quad \min_X \sum_{O_{j,k} \in \mathcal{O}} \pi(O_{j,k}) - s_{j,k} + \max(\pi_t(O_{j,k}) - t_{j,k}, 0)$$

3 **Constraints:**

4 // Machine assignment definition for each operation

5 **for** *each operation $O_{j,k} \in \mathcal{O}$* **do**

6    // Operation end time definition

7

$$\pi_t(O_{j,k}) = \pi(O_{j,k}) + dur(O_{j,k}) \quad (5)$$

8    **for** *each possible machine assignment $alt \in \mathcal{M}_{j,k}$* **do**

9       // Operation duration definition for machine assignment

10

$$dur_{alt}(O_{j,k}) = p_{j,k}^{alt} \quad (6)$$

11       // Operation end time definition for alternative assignment

12

$$\pi_{t,alt}(O_{j,k}) = \pi_{alt}(O_{j,k}) + dur_{alt}(O_{j,k}) \quad (7)$$

13       // Enforce solution linking if *alt* is selected

14

$$(\pi(O_{j,k}) = \pi_{alt}(O_{j,k})).\text{EnforceIf}(l_{alt}(O_{j,k}) = 1) \quad (8)$$

15

$$(dur(O_{j,k}) = dur_{alt}(O_{j,k})).\text{EnforceIf}(l_{alt}(O_{j,k}) = 1) \quad (9)$$

16

$$(\pi_t(O_{j,k}) = \pi_{t,alt}(O_{j,k})).\text{EnforceIf}(l_{alt}(O_{j,k}) = 1) \quad (10)$$

17    **end for**

18    // Only one machine assignment alternative can be selected

19

$$ExactlyOne(\{l_{alt}(O_{j,k}) = 1\}_{alt \in \mathcal{M}_{j,k}}) \quad (11)$$

20 **end for**

21 // Operation precedence constraints for each job

22 **for** *each job $j \in \mathcal{T}$* **do**

23    **for** *each operation $O_{j,k}$ with $1 \le k \le n_j$* **do**

24

$$\pi_t(O_{j,k}) \le \pi(O_{j,k+1}) \quad (12)$$

25    **end for**

26 **end for**

27 // Non-overlapping Operation constraints on each machine

28 **for** *each machine $alt \in \mathcal{M}$* **do**

29

$$\text{NoOverlap}\Big(\{\text{Interval}[\pi_{alt}(O_{j,k}), \pi_{t,alt}(O_{j,k})]\}_{O_{j,k} \in \mathcal{O}}\Big) \quad (13)$$

30 **end for**

---

---

**Algorithm 8:** Additional Constraints for FJSP RHO Subproblems $P_r$

---

1 **Decision Variables:** $X_r$ for the operation set $\mathcal{O}_{overlap,r}$ in the planning horizon (see Eq. equation 3)

2 **Solutions of Previously Executed Operations:** $(m, \pi)$

3 **Additional Constraints:**

4 `// Compliance with previously executed operations (if exists)`

5 **if** $(m, \pi)$ *is not empty* **then**

6      prev_machine_end_time $\leftarrow \{$alt $: \max\limits_{\substack{O_{j,k} \in (m,\pi), \\ m(O_{j,k})=\text{alt}}} \pi_t(O_{j,k})\}$ for each machine alt $\in \mathcal{M}$

7      prev_job_end_time $\leftarrow \{j : \max\limits_{O_{j,k} \in (m,\pi)} \pi_t(O_{j,k})\}$ for each job $j \in \mathcal{T}$

8      **for** *each operation* $O_{j,k} \in \mathcal{O}$ **do**

9          `// Ensure processing after all executed operations within the same job`

10          $\pi(O_{j,k}) \geq$ prev_job_end_time$[j]$

11          `// Ensure processing after all executed operations on the same machine`

12          **for** *each possible machine assignment* alt $\in \mathcal{M}_{j,k}$ **do**

13              $\pi_{alt}(O_{j,k}) \geq$ prev_machine_end_time$[alt]$

14          **end for**

15      **end for**

16 **end if**

---

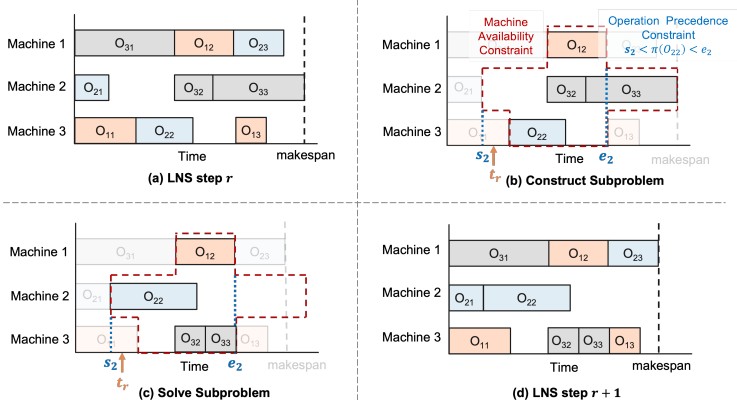

Figure 5: The Large Neighborhood Search (LNS) Pipeline from (Pacino and Van Hentenryck, 2011). (a) At the $r^{th}$ LNS iteration, we start with a complete solution $\Pi$ for the full FJSP. The solution at the first iteration is given by solving the full FSJP for a short duration (30s). (b) We construct a FJSP subproblem by selecting a subset of FJSP operations to update their solution. Two subproblem selection methods are considered based on time and machine based decomposition methods. When constructing the FJSP subproblem, we include additional constraints on the operation's start time and each machine's available time so that the subproblem's solution is compatible with the solution of the non-selected FJSP operations. (c) We feed the FJSP subproblem into a subsolver to get a new solution $\Pi_r$. The old solution of the subproblem is given to warm start the solve, as it empirically improves the performance. (d) We update the complete solution $\Pi$ with the new subproblem's solution $\Pi_r$, and repeat (a)-(d).

### A.5.3   BASELINE DETAILS

**Traditional solver w/o decomposition**   The CP-SAT and Genetic Algorithm (GA) results are, respectively, obtained by applying the competitive constraint programming solver OR-Tools CP-SAT (Perron and Didier), and the widely used hybrid genetic algorithm with tabu search meta-heuristics (Li and Gao, 2016), to solve the full FJSPs without decomposition. These comparisons emphasize the significance of decomposition in solving long-horizon FJSP.

**Learning-based solver w/o decomposition.**   For each setting, we train a DRL model (Wang et al., 2023c) on the same set of 450 training instances as L-RHO and also on a larger set of 20,000 instances, following the original paper[2]. At inference time, we decode the DRL model using Greedy and Sampling strategies (100 and 500 samples).

**Traditional solver w/ decomposition.**   Fig. 5 provides an illustration of the Large Neighborhood Search (LNS) algorithm that we compare as a baseline in the offline setting. In particular, (Pacino and Van Hentenryck, 2011) introduces Time Decomposition and Machine Decomposition to select different LNS neighborhoods for large-scale FJSP. The implementation details for each LNS iteration are as follows:

- The **ARD-LNS (Time-based)** leverages temporal locality to select a random start and end time of a contiguous time interval to define the LNS subproblem. It imposes additional machine availability constraints and operation precedence constraints in defining the subproblem. The former restricts the start and end time of each machine to be compatible of fixed operations, and the later restricts the start and end time of each selected operations to be compatible with the fixed operations within the same job. It further includes (1) adaptive LNS subproblem size: The contiguous time interval starts at a length of 20% of the horizon. If no improvement is found within 5 iterations, the size of time window increases by 5%, with the longest time window length capped at 50%. If improvement is found, the size returns to 20%. (2) New objective for each LNS subproblem: Instead of makespan, it instead maximizes the distance between

---

[2]We use the publicly available implementation from the authors at `https://github.com/wrqccc/FJSP-DRL`.

each operation's end time and its latest feasible completion time as the surrogate objective when solving each LNS subproblem.

- The **ARD-LNS (Machine-based)** in each LNS iteration selects a subset of machines and define the subproblem as the set of operations on those machines. As we select all operations on each machine, the subproblem only requires the operation precedence constraint. The subproblem is solved with Makespan as the objective.

In the original paper by (Pacino and Van Hentenryck, 2011), the LNS procedures are benchmarked on the dataset from (Hurink et al., 1994), where the largest instance consists of 225 operations (15 machines, 15 jobs, and 15 operations per job). We successfully reproduced their reported results on these instances; however, when applying their algorithm to the significantly larger-scale setting considered in Table 1, we observed a dramatic degradation in LNS performance. In contrast, the RHO-based algorithm maintains competitive performance on these long-horizon FJSP instances.

**Learning-guided solver w/ decomposition: Oracle-LNS (Time-based).** We use a LookAhead Oracle similar to the one in previous learning-guided LNS studies (Huang et al., 2022; Li et al., 2021). In each LNS iteration, we first select $K = 10$ Time Decomposition LNS subproblems as candidates. We then look ahead by solving these subproblems and choose the one that gives the biggest improvement in makespan. We randomly decide whether to evaluate the improvement based on the local makespan (for the current subproblem) or the global makespan (for the full FJSP problem), as this approach works better than focusing on just one objective. The local objective helps LNS prioritize subproblems needing the most fine-tuning, even though updating them doesn't immediately impact the final makespan. Meanwhile, the global objective encourages selecting subproblems toward the end of the time horizon to optimize the overall makespan. This differs from prior studies on Vehicle Routing (Li et al., 2021), Multi-Agent Path Finding (Huang et al., 2022) or Graph-Based Mixed Integer Linear Programming (Huang et al., 2023), where improving the local subproblem directly enhances the global objective.

**RHO decomposition.** Detailed descriptions of different RHO baselines (Default, Warm Start, Random, First) can be found in Table 4. In Table 1, for Default RHO, Warm Start RHO, and L-RHO, we set the RHO parameters $(H, S, T, T_{es}) = (80, 30, 60, 3)$. That is, each RHO subproblem involves $H = 80$ operations, and each execution steps move forward $S = 30$ operations. The time limit of solving each RHO subproblem is $T = 60s$, and we early terminate the CP-SAT (with a callback) if the best objective does not change after $T_{es} = 3s$. For Default RHO (Long), we use a more generous early stop time $T_{es} = 10s$, where we see the Default RHO's performance improves at the expense of substantially longer solve time. In Table 2, the RHO parameters are determined based on an extensive parameter grid search procedure, as described in Appendix A.5.4.

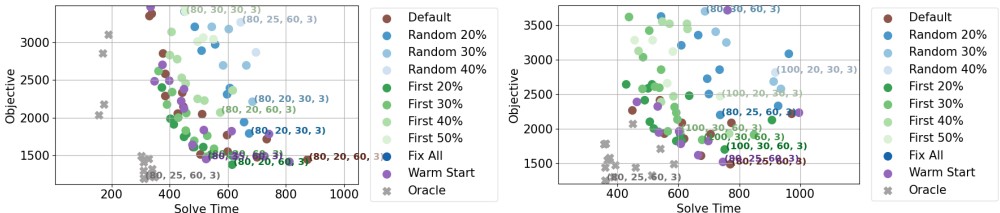

Figure 6: The objective and solve time for a variety of methods and RHO parameter settings, average across a set $\mathcal{K}_{ps}$ of FJSP instances with sizes $(25, 25, 24)$ on the left and $(30, 30, 24)$ on the right. Each dot represents a method and a RHO parameter setting $(H, S, T, T_{es})$. We label the selected best parameter setting for each method with text.

### A.5.4 ROLLING HORIZON PARAMETER SEARCH

We perform a grid search to find the best parameter setting for each RHO variant (default, baselines and our learning method). We consider the following four parameters as a joint parameter setting

$$\{ \text{RHO Planning Window Size } H, \quad \text{RHO Execution Step Size } S,$$
$$\text{Time Limit } T \text{ of each RHO FJSP Subproblem},$$
$$\text{Early Stopping Time } T_{es} \text{ for each RHO Subproblem}\}$$

In particular, the Early Stopping Time $T_{es}$ represents the time to early terminate the CP-SAT (with a callback) if the best objective does not change after $T_{es}$ seconds. We introduce this parameter due to the empirical observation that under a (relaxed) time limit, the objective of the best feasible solution obtained by CP-SAT may stuck at a given value for a long period of time.

We perform a grid search to evaluate the performances on a set $\mathcal{K}_{ps}$ of FJSP instances for each FJSP data distribution in the experiment Sec. 5, on the following parameter settings

$$\begin{aligned}
\mathcal{H} =& (\text{Window Size}, \text{Step Size}) \times (\text{Time Limit}, \text{Early Stop Time}) \\
=& \{(50, 15), (50, 20), (50, 25), (50, 30), \\
& (80, 20), (80, 25), (80, 30), (80, 35), (80, 40), \\
& (100, 20), (100, 30), (100, 40), (100, 50)\} \\
& \times \{(15s, 2s), (30s, 3s), (60s, 3s)\}
\end{aligned} \tag{14}$$

We visualize the objective and solve time for all methods and parameters in Fig. 6, where each dot represents the objective and solve time (averaged over $\mathcal{K}_{ps}$) of each method evaluated under a parameter setting. For each method, the dots of all parameters define a Pareto frontier for the objective and solve time trade-off. As we aim to reduce solve time with minimal impact on the objective, we use the following procedure to identify the individual best parameter setting of each method:

- Let $obj^*$ be the best objective across all methods and all parameter settings. We empirically find that $obj^*$ is typically achieved by our **Oracle** data collection method, which offer an objective improvement from default RHO.

- For each method $M$, we perform a line search with a step size $r_s = 0.1$ to find the best parameter setting: starting with iteration $i = 0$, let $\mathcal{H}_i^M \subseteq \mathcal{H}$ be the parameters whose objectives are within the $i^{th}$ bucket $[(1 + r_s \times i) \times obj^*, (1 + r_s \times (i+1)) \times obj^*]$. If $\mathcal{H}_i^M \neq \emptyset$, we then select the parameter setting $H^{M*} \in \mathcal{H}_i$ with the shortest solve time as the individual best parameter setting for the method $M$; if $\mathcal{H}_i^M = \emptyset$, we then move on to the next iteration $i + 1$ and repeat the procedure, until we find the individual best parameter setting.

The line search procedure ensures that if there are two parameter settings with similar objectives, but one results in a much shorter solve time than the other, we would select the parameter setting with a shorter solve time as the best parameter setting for the method. In Table 12, We provide a list of the best parameter setting for each method, identified through the line search procedure.

In Table 2, we apply the best parameter setting selected for each RHO variant to evaluate the method's performance. L-RHO is evaluated using the same parameters as Oracle. Since the best parameters for Default, Warm Start, and Oracle are consistent across different settings, we use a fixed set of RHO parameters $(H, S, T, T_{es}) = (80, 30, 60, 3)$ in Table 1 to avoid additional computation for parameter search.

Table 12: Best RHO parameter settings $(H, S, T, T_{es})$ of various RHO methods for the different FJSP settings presented in Table 2. Parameter settings highlighted in bold are the same as the Oracle's best parameter setting for each FJSP setting.

|  | 25, 25, 24 | 30, 30, 24 | 35, 35, 30 | 40, 40, 40 |
|---|---|---|---|---|
| **Default** | (80, 20, 60, 3) | **(80, 25, 60, 3)** | (80, 30, 60, 3) | **(80, 20, 60, 3)** |
| **Warm Start** | (80, 35, 60, 3) | **(80, 25, 60, 3)** | **(80, 25, 60, 3)** | **(80, 20, 60, 3)** |
| **Fix first 20%** | (80, 20, 60, 3) | (100, 30, 60, 3) | (100, 40, 60, 3) | (100, 40, 60, 3) |
| **Fix first 30%** | (80, 20, 60, 3) | (100, 30, 60, 3) | (100, 40, 60, 3) | (100, 40, 60, 3) |
| **Fix first 40%** | (80, 20, 60, 3) | (100, 30, 60, 3) | (100, 30, 30, 3) | (100, 30, 30, 3) |
| **Fix first 50%** | (80, 30, 30, 3) | (100, 20, 30, 3) | (100, 40, 30, 3) | (100, 40, 30, 3) |
| **Random 10%** | **(80, 25, 60, 3)** | **(80, 25, 60, 3)** | **(80, 25, 60, 3)** | (80, 25, 60, 3) |
| **Random 20%** | (80, 20, 30, 3) | **(80, 25, 60, 3)** | (80, 25, 30, 3) | (80, 25, 60, 3) |
| **Random 30%** | (80, 20, 30, 3) | (80, 30, 60, 3) | (100, 30, 30, 3) | (80, 25, 30, 3) |
| **Oracle** | **(80, 25, 60, 3)** | **(80, 25, 60, 3)** | **(80, 25, 60, 3)** | **(80, 20, 60, 3)** |

### A.5.5 Performance of RHO methods for FJSP $(30, 30, 24)$.

In Table 13, we provide the detailed results of all RHO baseline methods' performance under both the individual best RHO parameter setting (**Individual Best Param.**) and a fixed RHO parameter setting (the Oracle's best parameter, **Same Param.**) for FJSP $(30, 30, 24)$ with the total start delay objective and no observation noise, which is used as the base setting in Fig. 3 middle, (ii), of Sec. 5.2.

Table 13: Time and Objective improvements (TI%, OI%) of different RHO variants for **FJSP** $(\mathbf{30, 30, 24})$ under the start delay objective. We report the mean values (higher the better, negative indicates degradation) and two standard errors. The best TI% and OI% among the RHO methods (excluding Oracle) are bold-faced. L-RHO significantly outperforms all baseline methods in the objective with substantially reduced solve time. Additionally, L-RHO closely aligns with the performance of the Look-Ahead Oracle, demonstrating its learning effectiveness.

| 30, 30, 24 | Individual Best Param. | | Same Param. | |
|---|---|---|---|---|
|  | TI % | OI % | TI % | OI % |
| **Default** | $853.9s \pm 26.9s$ $(0.0\% \pm 0.0\%)$ | $1736.9 \pm 116.8$ $(0.0\% \pm 0.0\%)$ | $0.0\% \pm 0.0\%$ | $0.0\% \pm 0.0\%$ |
| **Warm Start All** | $-0.1\% \pm 1.3\%$ | $0.8\% \pm 2.2\%$ | $-0.1\% \pm 1.3\%$ | $0.8\% \pm 2.2\%$ |
| **Fix first 20%** | $3.6\% \pm 1.5\%$ | $-14.3\% \pm 2.7\%$ | $19.4\% \pm 1.5\%$ | $-23.1\% \pm 3.8\%$ |
| **Fix first 30%** | $12.4\% \pm 1.6\%$ | $-21.5\% \pm 3.8\%$ | $20.9\% \pm 1.7\%$ | $-57.9\% \pm 5.1\%$ |
| **Fix first 40%** | $19.5\% \pm 1.7\%$ | $-41.2\% \pm 4.4\%$ | $17.8\% \pm 2.0\%$ | $-131.4\% \pm 7.7\%$ |
| **Fix first 50%** | $7.9\% \pm 2.0\%$ | $-78.1\% \pm 5.5\%$ | $19.0\% \pm 2.1\%$ | $-238.5\% \pm 13.0\%$ |
| **Fix first 60%** | $33.6\% \pm 1.7\%$ | $-225.8\% \pm 11.8\%$ | $26.9\% \pm 2.3\%$ | $-416.0\% \pm 19.9\%$ |
| **Fix first 80%** | $\mathbf{51.0\% \pm 1.7\%}$ | $-727.1\% \pm 35.4\%$ | $50.2\% \pm 1.6\%$ | $-874.8\% \pm 46.3\%$ |
| **Random 10%** | $1.8\% \pm 1.7\%$ | $-20.2\% \pm 3.1\%$ | $1.80\% \pm 1.69\%$ | $-20.2\% \pm 3.1\%$ |
| **Random 20%** | $0.3\% \pm 2.0\%$ | $-54.7\% \pm 4.6\%$ | $1.8\% \pm 1.7\%$ | $-20.2\% \pm 3.1\%$ |
| **Random 30%** | $9.9\% \pm 2.3\%$ | $-153.6\% \pm 8.1\%$ | $-0.3\% \pm 2.4\%$ | $-103.8\% \pm 7.2\%$ |
| **Random 40%** | $-10.9\% \pm 3.3\%$ | $-105.6\% \pm 7.2\%$ | $2.0\% \pm 2.5\%$ | $-164.68\% \pm 9.43\%$ |
| **(Oracle)** | $(49.9\% \pm 1.1\%)$ | $(15.3\% \pm 2.22\%)$ | $(49.9\% \pm 1.1\%)$ | $(15.3\% \pm 2.2\%)$ |
| **L-RHO (Ours)** | $41.5\% \pm 6.4\%$ | $\mathbf{14.0\% \pm 10.2\%}$ | $41.5\% \pm 6.4\%$ | $14.0\% \pm 10.2\%$ |

### A.5.6 Performance of RHO methods under the same RHO parameter.

In Table 14, we present the performance of all RHO variants under the same RHO parameter setting (the Oracle's best parameter), for all three FJSP settings in Table 2. The TI% and OI% is relative to Default RHO under Default's best parameter (same as in Table 2). From Table 13 and 14, we

observe that although the First $\sigma_f$ baseline can substantially reduce the solve time when $\sigma_f$ is high, the associated objective drastically deteriorates at the same time. In contrast, our learning method L-RHO not only significantly significantly accelerates Default RHO, but it is also the only method that simultaneously achieves a notable improvement in the objective.

Table 14: Time and Objective improvements (TI%, OI%) of different RHO methods in comparison with Default RHO (with Default's best parameter) for three FJSP settings in Table 2; *all the methods are evaluated on the same RHO parameter setting (**Oracle's best RHO parameter**)*. We report the mean values (higher the better, negative indicates degradation) and two standard errors. The best TI% and OI% among the RHO methods (excluding Oracle) are bold-faced. L-RHO significantly outperforms all baseline methods in the objective with substantially reduced solve time.

| | 25, 25, 24 | | 35, 35, 30 | | 40, 40, 40 | |
|---|---|---|---|---|---|---|
| | TI % | OI % | TI % | OI % | TI % | OI % |
| **Default** | 17.6% ± 1.3% | -6.9% ± 4.3% | -16.0% ± 2.1% | 5.1% ± 3.8% | 0.0% ± 0.0% | 0.0% ± 0.0% |
| **Warm Start** | 20.0% ± 1.3% | -1.7% ± 2.5% | -12.9% ± 1.6% | 11.2% ± 2.7% | 5.2% ± 2.4% | 6.8% ± 4.8% |
| **Fix first 20%** | 36.2% ± 1.2% | -3.78% ± 2.7% | 0.3% ± 1.6% | -33.4% ± 4.0% | 4.6% ± 3.5% | -78.3% ± 9.0% |
| **Fix first 30%** | 41.6% ± 1.2% | -14.6% ± 3.0% | -0.9% ± 1.7% | -91.5% ± 5.4% | 1.5% ± 4.2% | -207.7% ± 17.3% |
| **Fix first 40%** | 42.1% ± 1.3% | -46.8% ± 4.2% | -5.1% ± 2.6% | -222.2% ± 11.1% | 7.7% ± 4.7% | -423.2% ± 30.6% |
| **Fix first 50%** | 43.8% ± 1.2% | -100.9% ± 6.8% | 3.2% ± 3.0% | -359.2% ± 16.6% | 24.5% ± 3.7% | -748.0% ± 46.2% |
| **Fix first 60%** | 46.9% ± 1.4% | -218.4% ± 12.1% | 17.2% ± 2.7% | -600.4% ± 29.1% | 40.7% ± 2.5% | -1118.0% ± 80.6% |
| **Fix first 80%** | **59.5% ± 1.4%** | -594.5% ± 32.5% | **49.6% ± 1.8%** | -1105.2% ± 55.3% | **73.2% ± 1.1%** | -1892.8% ± 138.9% |
| **Random 10%** | 23.0% ± 1.3% | -11.9% ± 2.5% | -15.5% ± 1.7% | -12.9% ± 3.4% | -0.3% ± 2.7% | -16.8% ± 5.8% |
| **Random 20%** | 24.2% ± 1.5% | -32.8% ± 3.6% | -18.8% ± 2.0% | -58.0% ± 5.7% | -1.6% ± 3.1% | -61.8% ± 10.0% |
| **Random 30%** | 23.6% ± 1.7% | -75.1% ± 5.9% | -17.4% ± 2.3% | -104.1% ± 7.3% | -0.8% ± 3.3% | -131.8% ± 12.8% |
| **Random 40%** | 27.4% ± 2.0% | -153.3% ± 9.1% | -14.4% ± 3.1% | -204.9% ± 11.4% | 1.4% ± 3.6% | -202.6% ± 20.9% |
| **(Oracle)** | (60.3% ± 1.0%) | (18.5% ± 1.7%) | (40.5% ± 1.0%) | (22.4% ± 2.3%) | (53.2% ± 1.3%) | (12.8% ± 4.4%) |
| **L-RHO (Ours)** | 53.0% ± 1.0% | **16.0% ± 1.9%** | 35.1% ± 1.3% | **21.0% ± 2.4%** | 47.3% ± 2.2% | **13.13% ± 4.4%** |

## A.6 Additional Experimental Results

### A.6.1 Performance on Real World Dataset (Dauzère-Pérès and Paulli (1997)

We find most of the traditional benchmarks on small scale ($< 500$ operations). We benchmark the performance on Dauzère-Pérès and Paulli (1997), 1997, which is the largest traditional benchmarks available, containing 16 instances evenly divided to three different sizes (respectively, 196, 293, and 387 operations). We exclude 4 instances from the dataset, which is simple and take $< 2s$ for RHO methods to solve. To mimic large scale settings with more operations, We further replicate all operations for each job $k$ times, with $k = 3, 6, 9$ to mimic long horizon scale settings with more operations. We provide the performance comparison in Table 15. For DRL and L-RHO, we both directly transfer the model learned on 600 operations (10, 20, 30) setting to the real world instances. The *best known solution* for the Dauzere original setting has an average makespan of *2120* with standard deviation 78.

We find that the direct transfer setting, our L-RHO outperforms DRL in the real world dataset, achieving a close gap with respect to best known solution in the literature for the smallest non-augmented setting, and the performance advantage further holds for longer-horizon augmented settings. Furthermore, L-RHO significantly speeds up Default RHO in all settings, demonstrating the benefit of introducing learning to accelerate RHO. We further note there exists slight mismatch in the training distribution and the real world dataset, and it is an interesting future work is to train the model on a diverse set of distribution to further improve L-RHO's transfer learning performance.

Table 15: Performance Comparison on the Dauzère-Pérès and Paulli (1997) dataset. The best known solution of the original dataset (the first result column) has a makespan of $2120 \pm 78$. In the last three result column, we randomly sample (with replacement) $\times k$ operations for each job to mimic the long horizon settings.

|  | Original | | Augment 3 | | Augment 6 | | Augment 9 | |
|---|---|---|---|---|---|---|---|---|
|  | **Time** | **Makespan** | **Time** | **Makespan** | **Time** | **Makespan** | **Time** | **Makespan** |
| DRL-20K (Wang) - Greedy | $2 \pm 0.1$ | $2493 \pm 44$ | $6 \pm 0.5$ | $7279 \pm 69$ | $12 \pm 1.0$ | $14337 \pm 218$ | $22 \pm 2.6$ | $21218 \pm 250$ |
| DRL-20K (Wang) - Sample 100 | $8 \pm 1$ | $2279 \pm 19$ | $57 \pm 9.6$ | $6946 \pm 83$ | $235 \pm 42$ | $13935 \pm 164$ | $534 \pm 95.2$ | $20784 \pm 226$ |
| DRL-20K (Wang) - Sample 500 | $35 \pm 6$ | $2253 \pm 21$ | $321 \pm 58$ | $6887 \pm 79$ | $1341 \pm 231$ | $13884 \pm 162$ | $3014 \pm 516$ | $20722 \pm 216$ |
| DRL-20K (Wang) - Sample 1000 | $72 \pm 13$ | $2247 \pm 20$ | $670 \pm 116$ | $6858 \pm 83$ | $2679 \pm 458$ | $13859 \pm 156$ | $6068 \pm 1038$ | $20706 \pm 217$ |
| Default RHO | $83 \pm 33$ | $2174 \pm 68$ | $290 \pm 84$ | $6505 \pm 232$ | $509 \pm 150$ | $13085 \pm 428$ | $794 \pm 241$ | $19689 \pm 592$ |
| Warm Start RHO | $75 \pm 25$ | $2167 \pm 68$ | $241 \pm 67$ | $6490 \pm 237$ | $450 \pm 141$ | $13034 \pm 432$ | $684 \pm 210$ | $19589 \pm 609$ |
| **L-RHO Transfer** | $65 \pm 23$ | $2185 \pm 66$ | $199 \pm 70$ | $6499 \pm 254$ | $345 \pm 134$ | $13015 \pm 504$ | $492 \pm 200$ | $19593 \pm 725$ |

### A.6.2 Accuracy, TPR, TNR, Precision, and Recall of the Learning Method.

We provide the accuracy, true positive rate (TPR), true negative rate (TNR) precision, and recall of the learning architecture (Fig. 2) in Table 16.

### A.6.3 Ablation Study: A Detailed Comparison with Attention-Based Architecture.

We compare the architecture in Fig. 2 with a more advanced attention based architecture shown in Fig. 7. We make the following two modifications to obtain the new architecture:

- First, at the machine/operation concatenation block, for each new operation $O \in \mathcal{O}_{new,r}$ that does not have a machine assignment in $m_{r-1}$, we concatenate it a dummy machine embedding (all-zero feature) and the global embedding from mean pooling on all entities (operations and machines).

- Then, after the machine/operation concatenation block of the original architecture, we perform a Multi-Head Attention between the overlapping operations $\mathcal{O}_{overlap,r}$ and the new operations $\mathcal{O}_{new,r}$ (with 4 attention heads), and we also include a residual connection for the overlapping operations.

For the Multi-Head Attention, we take the overlapping operations' embeddings as queries ($Q$), and consider **the following three setup for the attention keys ($K$) and values ($V$)**: (1) New operations only, (2) Overlapping operations only, and (3) Both overlapping and new operations.

Table 16: Accuracy, True Positive Rate (TPR), True Negative Rate (TNR), Precision, and Recall of the learning architecture in Fig. 2 on the hold out validation set for each setting in Table 1 and 2.

|                  | Accuracy | True Positive Rate | True Negative Rate | Precision | Recall |
|------------------|----------|--------------------|--------------------|-----------|--------|
| Table 1          |          |                    |                    |           |        |
| 600 (10, 20, 30) | 0.77     | 0.81               | 0.67               | 0.86      | 0.81   |
| 800 (10, 20, 40) | 0.76     | 0.79               | 0.67               | 0.86      | 0.79   |
| 1200 (10, 20, 60)| 0.76     | 0.77               | 0.73               | 0.87      | 0.77   |
| Table 2          |          |                    |                    |           |        |
| 625 (25, 25, 25) | 0.78     | 0.71               | 0.85               | 0.83      | 0.71   |
| 1225 (35, 35, 35)| 0.81     | 0.72               | 0.87               | 0.79      | 0.72   |
| 1600 (40, 40, 40)| 0.81     | 0.69               | 0.90               | 0.83      | 0.69   |

In Table 17, we find that for option (1) and (2) the introduction of the attention architecture leads to a slightly higher True Negative Rate (TNR) but lower True Positive Rate (TPR) and a slight reduction accuracy, when compared with the architecture without attention. Furthermore, when evaluating the performance on 600 operations FJSP (10, 20, 30) in Table 1, we see that option (1) and (2) , results in a longer solve time but an improved makespan from the architecture without attention. We also note that option (3) is strictly dominated by the performance of the architecture without attention.

We note that the TNR-TPR tradeoff on the performance and solve time aligns with our theoretical analysis, as fixing something that should not have been (low TNR) harms the objective but helps the solve time, while failing to fix something that should have been (low TPR) harms the solve time and also indirectly harms the objective (under a fixed time limit).

Due to the time benefit of the architecture without attention and the relatively competitive objective, we believe it makes sense to keep the simpler architecture without attention in the main paper.

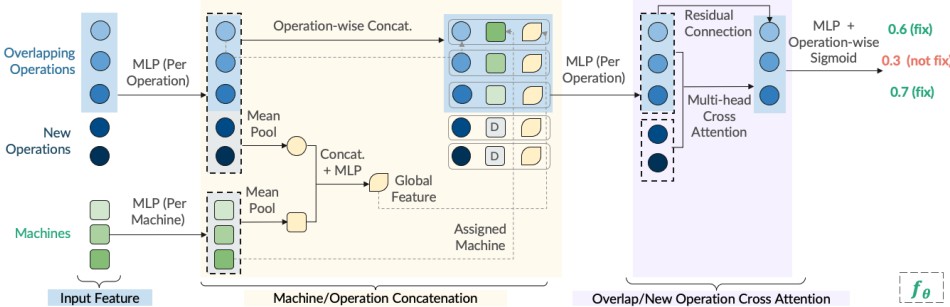

Figure 7: **Ablation neural architecture: Attention among the overlapping and new operations.** The architecture follows Fig. 2, but introduces an additional cross attention among the overlapping and new operations before output the predicted probability for each overlapping operation.

Table 17: Comparison of the learning architecture in Fig. 2 and the attention-based learning architecture in Fig. 7 on the hold out validation set for the 600 operations setting (10, 20, 30) in Table 1.

|                                              | Accuracy | TPR  | TNR  | Precision | Recall | Time (s)      | Makespan       |
|----------------------------------------------|----------|------|------|-----------|--------|---------------|----------------|
| No Attention                                 | 0.77     | 0.81 | 0.67 | 0.86      | 0.81   | **126 ± 19**  | 1513 ± 70      |
| Attention ($K\&V$: New Ops.)                 | 0.76     | 0.77 | 0.74 | 0.88      | 0.77   | 170 ± 18      | **1504 ± 68**  |
| Attention ($K\&V$: Overlapping Ops.)         | 0.75     | 0.75 | 0.75 | 0.88      | 0.75   | 179 ± 21      | 1507 ± 57      |
| Attention ($K\&V$: Overlapping and New Ops.) | 0.76     | 0.76 | 0.74 | 0.88      | 0.76   | 133 ± 14      | 1527 ± 68      |

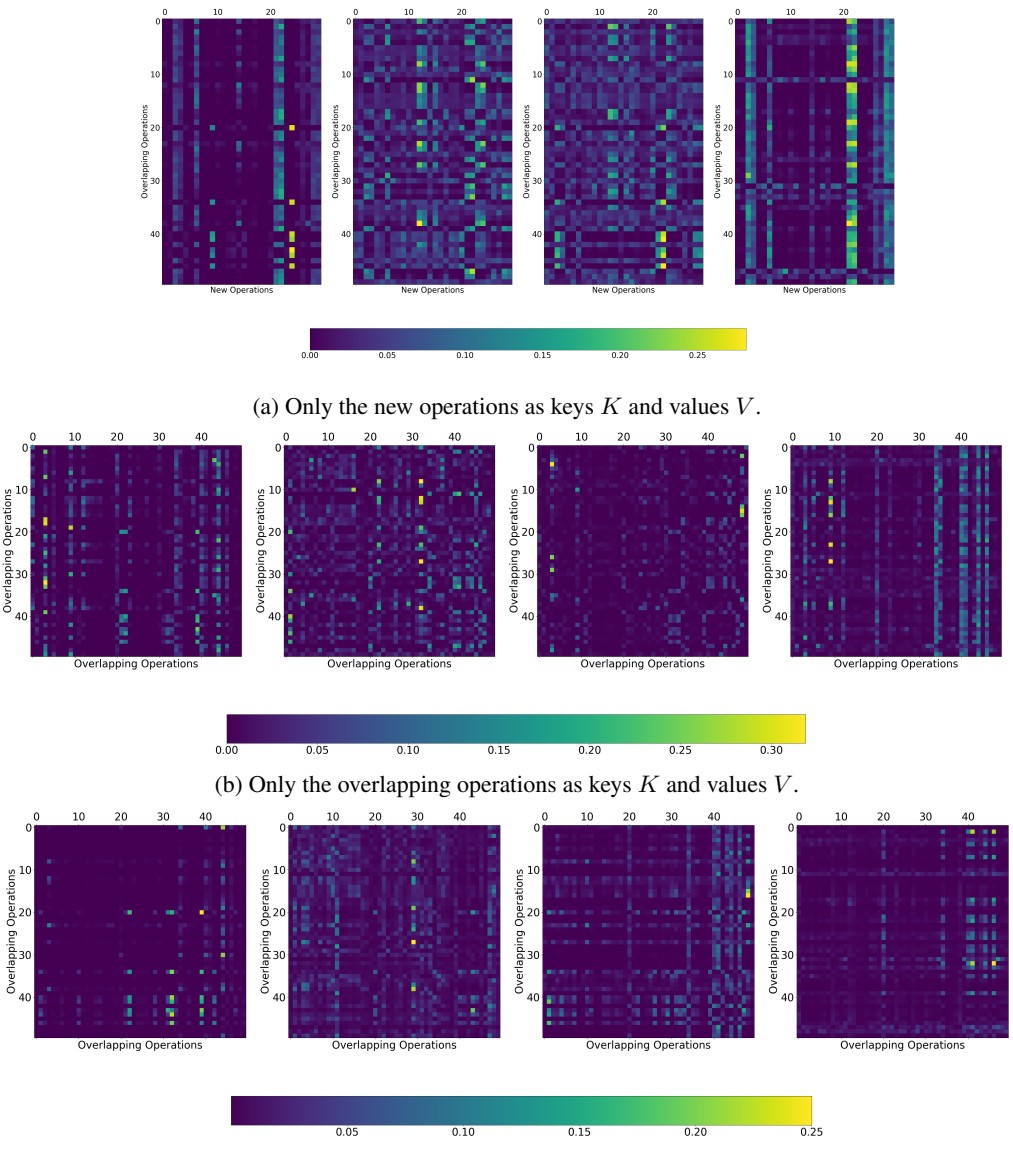

(a) Only the new operations as keys $K$ and values $V$.

(b) Only the overlapping operations as keys $K$ and values $V$.

(c) Both overlapping and new operations as keys $K$ and values $V$.

Figure 8: **Comparison of Learned Attention Maps.** We set the overlapping operations as the query $Q$ (the rows) and consider three different types of keys $K$ and values $V$ (the columns).

### A.6.4 PERFORMANCE COMPARISON WITH DRL UNDER THE SAME NUMBER OF FJSP TRAINING INSTANCE.

In Table 18, we compare L-RHO with DRL (Wang et al., 2023a) when training on the same set of 450 FJSP instances. We observe that DRL's performance degrades with 450 training instances, especially at larger scales, highlighting L-RHO's unique advantage with a lighter training set. This can be beneficial under situations where FJSP training instances are harder to acquire.

Table 18: Compare L-RHO with DLR Wang et al. (2023a) under the same number of FJSP training instances (DRL-450).

| | **600** (10, 20, 30) | | **800** (10, 20, 40) | | **1200** (10, 20, 60) | | **2000** (10, 20, 100), **Transfer** | |
|---|---|---|---|---|---|---|---|---|
| | Time (s) ↓ | Makespan ↓ | Time (s) ↓ | Makespan ↓ | Time (s) ↓ | Makespan ↓ | Time (s) ↓ | Makespan ↓ |
| **CP-SAT** | 1800 | $2274 \pm 147$ | 1800 | $4017 \pm 413$ | 1800 | $10925 \pm 1013$ | 1800 | $39585 \pm 2707$ |
| **CP-SAT (10 hours)** | 36000 | $1583 \pm 65$ | 1800 | $2128 \pm 75$ | 1800 | $3206 \pm 87$ | 1800 | $18821 \pm 1986$ |
| **DRL-450, Greedy** | $4 \pm 0.04$ | $1635 \pm 75$ | $6 \pm 0.04$ | $2178 \pm 85$ | $11 \pm 0.2$ | $3307 \pm 115$ | $17 \pm 0.5$ | $5415 \pm 134$ |
| **DRL-450, Sample 100** | $30 \pm 1$ | $1563 \pm 62$ | $48 \pm 1$ | $2093 \pm 70$ | $111 \pm 3$ | $3237 \pm 91$ | $301 \pm 8$ | $5341 \pm 108$ |
| **DRL-450, Sample 500** | $143 \pm 4$ | $1546 \pm 50$ | $261 \pm 8$ | $2070 \pm 68$ | $610 \pm 25$ | $3210 \pm 94$ | $1751 \pm 30$ | $5311 \pm 111$ |
| **DRL-20K, Greedy** | $4 \pm 0.02$ | $1628 \pm 72$ | $6 \pm 0.04$ | $2128 \pm 80$ | $10 \pm 0.1$ | $3141 \pm 97$ | $16 \pm 0.2$ | $5184 \pm 114$ |
| **DRL-20K, Sample 100** | $29 \pm 0.3$ | $1551 \pm 60$ | $47 \pm 1$ | $2048 \pm 69$ | $110 \pm 3$ | $3063 \pm 81$ | $302 \pm 10$ | $5082 \pm 98$ |
| **DRL-20K, Sample 500** | $146 \pm 5$ | $1537 \pm 61$ | $261 \pm 8$ | $2031 \pm 68$ | $597 \pm 16$ | $3045 \pm 79$ | $1738 \pm 16$ | $5062 \pm 98$ |
| **Default RHO** | $244 \pm 21$ | $1558 \pm 73$ | $348 \pm 26$ | $2103 \pm 78$ | $545 \pm 36$ | $3136 \pm 91$ | $862 \pm 42$ | $5207 \pm 114$ |
| **Warm Start RHO** | $203 \pm 23$ | $1521 \pm 67$ | $278 \pm 22$ | $2055 \pm 75$ | $420 \pm 33$ | $3081 \pm 96$ | $716 \pm 41$ | $5057 \pm 106$ |
| **L-RHO (450)** | $\mathbf{126 \pm 19}$ | $\mathbf{1513 \pm 70}$ | $\mathbf{160 \pm 23}$ | $\mathbf{2015 \pm 86}$ | $\mathbf{259 \pm 37}$ | $\mathbf{3011 \pm 106}$ | $\mathbf{473 \pm 52}$ | $\mathbf{4982 \pm 132}$ |

### A.6.5 PERFORMANCE OF DEFAULT RHO WITH LONGER TIME LIMITS

As described in Appendix A.5.4, two parameters control the time limit per RHO subproblem: 1) total time limit, where solving terminates if the time exceeds $T$ seconds; 2) early stop time, where solving terminates early if the objective does not improve within $T_{es}$ seconds. In Table **??**, we evaluate the default RHO with increased $T$ and $T_{es}$. The results show that, as the time limit increases by over 10x, the default RHO achieves better objectives than our L-RHO, albeit with prohibitively long runtime. Our L-RHO delivers superior performance under practical time constraints, offering benefits especially in online cases.

Table 19: Comparison of Default RHO's performance across different time limit parameters $T$ and $T_{es}$ for RHO.

| | **600 (10, 20, 30)** | | **800 (10, 20, 40)** | | **1200 (10, 20, 60)** | |
|---|---|---|---|---|---|---|
| **T, T$_{es}$** | **Time (s)** | **Makespan** | **Time (s)** | **Makespan** | **Time (s)** | **Makespan** |
| 60, 3 (reported) | $244 \pm 21$ | $1558 \pm 73$ | $348 \pm 26$ | $2103 \pm 78$ | $545 \pm 36$ | $3136 \pm 91$ |
| 60, 10 | $599 \pm 55$ | $1529 \pm 58$ | $728 \pm 98$ | $2044 \pm 75$ | $1109 \pm 108$ | $3002 \pm 87$ |
| 60, 30 | $923 \pm 71$ | $1502 \pm 66$ | $1099 \pm 130$ | $2028 \pm 72$ | $1697 \pm 175$ | $2963 \pm 78$ |
| 60, 60 | $819 \pm 117$ | $1494 \pm 67$ | $1096 \pm 156$ | $2018 \pm 71$ | $1714 \pm 154$ | $2968 \pm 89$ |
| 120, 60 | $1476 \pm 312$ | $1497 \pm 70$ | $1811 \pm 241$ | $1997 \pm 70$ | $2809 \pm 308$ | $2932 \pm 75$ |
| 180, 120 | $2142 \pm 461$ | $1490 \pm 70$ | $2509 \pm 346$ | $1994 \pm 76$ | $3932 \pm 590$ | $2924 \pm 83$ |
| 360, 180 | $3631 \pm 816$ | $1476 \pm 64$ | $3673 \pm 797$ | $1987 \pm 68$ | $5663 \pm 900$ | $2924 \pm 86$ |
| **L-RHO** | $126 \pm 19$ | $1513 \pm 70$ | $160 \pm 23$ | $2015 \pm 86$ | $259 \pm 37$ | $3011 \pm 106$ |

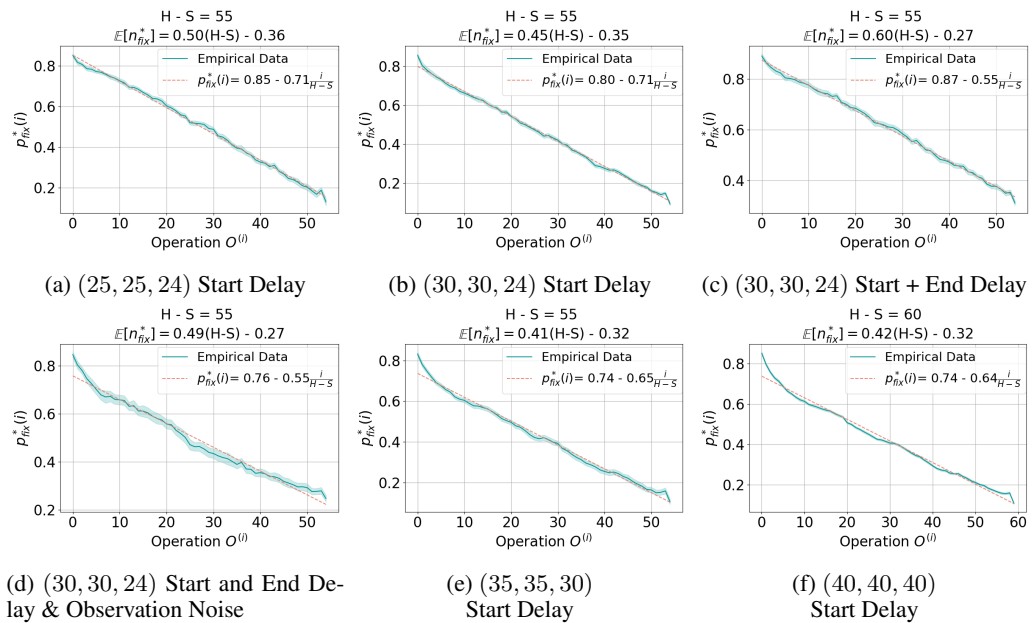

Figure 9: **The empirical $p^*_{fix}(i)$ distribution across a variety of FJSP settings.** We plot the empirical distribution calculated by Eq. equation 15 with the solid green curve, and the fitted linear approximation $p^*_{fix}(i) = b - m\frac{i}{H-S}$ with the dashed red line. We also plot the empirical standard error (the shaded green area) computed across all RHO iterations, with $N_{iter}$ ranging from 20 to 75 (based on the FJSP problem size). We observe a consistent linear decreasing trend across all FJSP distributions, and the low standard errors further indicate consistency across RHO iterations.

### A.7 DETAILED THEORETICAL PROBABILISTIC ANALYSIS

Following the main paper, we analyze a generic subproblem $\hat{P}_r := \hat{P}$ at each RHO iteration $r$. For ease of notation, we omit the $r$ subscript for the rest of this section when it is clear from the context.

#### A.7.1 DETAILS OF ALL CONSIDERED RHO METHODS

We consider a set of assignment-based RHO warm start procedure, where we follow the $RHO_{test}$ procedure in Fig. 1 (b), bottom but obtain $\mathcal{O}_{fix}$ at each RHO iteration with different methods described as follows. For tractability analysis, we consider the same RHO window size $H$ and step size $S$, which results in $H - S$ overlapping operations $|\mathcal{O}_{overlap}| = H - S$ in consecutive RHO iterations.

- **Random $\sigma_{\mathbf{R}}$**: $\mathcal{O}_{fix}$ is obtained by selecting each operation in $\mathcal{O}_{overlap}$ with a probability $\sigma_R$ uniformly at random.

- **First $\sigma_{\mathbf{F}}$**: $\mathcal{O}_{fix}$ is obtained as the first $\sigma_F$ fraction of operations in $\mathcal{O}_{overlap}$ with the earliest RHO sequence order (determined by release time for delay-based objectives or operation precedence for makespan). Notably, Random/First $\sigma = 0$ exactly matches the default RHO procedure.

- **L-RHO (Ours)**: $\mathcal{O}_{fix}$ is *predicted* by the learning model $f_\theta$ as described in Sec. 4 (Fig. 1, (b) bottom).

- **Oracle**: the look-ahead Oracle with $\mathcal{O}^*_{fix}$, which is empirically obtained by applying the same procedure as in Fig. 1 (b) top but only includes the solve time of the restricted subproblem $\hat{P}^*$. As illustrated in Fig. 4 (left), we analyze the performance of each method (Random, First, L-RHO) based on the closeness of $\mathcal{O}_{fix}$ to the Oracle $\mathcal{O}^*_{fix}$.

A.7.2  EMPIRICAL VALIDATION OF THE LINEAR DECAY ASSUMPTION FOR $p^*_{fix}(i)$

We compute the distribution of $p^*_{fix}(i)$ using the training set $\mathcal{K}_{train}$. Specifically, for each FJSP training instance $P$ and RHO iteration $r$, we have a training label with the form $y^{P,r*} = \{y^{P,r*}_{O^{(i)}}\}_{O^{(i)} \in \mathcal{O}_{overlap,r}}$, where the index $i \in \{1, ..., H - S\}$ of each operations $O^{(i)}$ follows the RHO sequence order, and the associated $\mathcal{O}^*_{fix,r} = \{O^{(i)} \in \mathcal{O}_{overlap,r} : y_{O^{(i)}} = 1\}$. For each RHO iteration $r$, we calculate the empirical

$$p^{*emp}_{fix,r}(i) = \frac{1}{N_P} \sum_{\text{FJSP instance } P} y^{P,r*}_{O^{(i)}},$$

where $N_P$ is the number of training instances, from which we obtain the empirical distribution averaged across all RHO iterations and the associated standard error

$$p^{*emp}_{fix}(i) = \frac{1}{N_{iter}} \sum_{\text{RHO iteration } r} p^{*emp}_{fix,r}(i),$$

$$s.e.(p^{*emp}_{fix}(i)) = s.t.d(\{p^{*emp}_{fix,r}(i)\}_{\text{RHO iteration } r})/\sqrt{N_{iter}},$$

(15)

where $N_{iter}$ is the number of RHO iterations. In Fig 9, we visualize the empirical $p^{*emp}_{fix}(i)$ distributions (and the associated two standard errors) across a variety of FJSP settings, under the Oracle's best RHO parameter setting from grid search ($H = 80, S = 25$ for most settings). We also plot the linear approximation $p^*_{fix}(i) = b - m\frac{i}{H-S}$ through fitting the empirical distribution, resulting in $\mathbb{E}[n^*_{fix}] = \sum_{i=1}^{H-S} p^*_{fix}(i) = b - m\sum_{i=1}^{H-S} \frac{i}{H-S} = (b - \frac{m}{2})(H - S) - \frac{m}{2}$.

**Observations from the $p^*_{fix}$ distribution in Fig. 9.** We observe a consistent linear decreasing trend across all distributions, with varying slopes and intercepts. The empirical distribution closely aligns with our fitted linear approximation, hence validating our Assump. 1; the low standard error further reflects the consistency across RHO iterations. Intuitively, this is because the new operations $\mathcal{O}_{new}$ in the current RHO subproblem $\hat{P}$ are closer in the precedence order or release time with the later overlapping operations in $\mathcal{O}_{overlap}$, making them more likely to be scheduled around similar time and thus have more effect on each other's solution.

Moreover, $\mathbb{E}[n^*_{fix}]$ is around $0.5(H - S)$ for all settings, with a higher $\mathbb{E}[n^*_{fix}]$ in (c) for $(30, 30, 24)$ with Start and End Delay, but lower $\mathbb{E}[n^*_{fix}]$ for larger FJSP sizes $(35, 35, 30)$ and $(40, 40, 40)$ in (e) and (f). We have the following noteworthy comparisons:

1.  Comparing FJSP $(30, 30, 24)$ (c) Start and End Delay and (b) Start Delay, our analysis indicates that the larger number of Oracle fixed operations (high $\mathbb{E}[n^*_{fix}]$) in (c) enhances the performance of the Random and First heuristics (reducing performance gap relative to Default RHO) compared with (b). This aligns with our empirically observation when comparing (i) and (ii) of Fig 3 (Right).

2.  Comparing FJSP $(30, 30, 24)$ (d) Start and End Delay & Observation Noise with (c), our analysis indicates a performance decline of Random and First under observation noise, as (d) has a significantly lower $\mathbb{E}[n^*_{fix,r}]$ than (c). This is consistent with our empirical observation when comparing (iii) and (ii) in Fig 3 (Right). Moreover, comparing (d) with (b), we see that (d) has a similar $\mathbb{E}[n^*_{fix,r}]$ but a notably lower slope $m$ than (b). Our analysis indicates that this reduces the performance advantage of First over Random, which is reflected by our empirical comparison of (iii) and (i) in Fig. 3 (Right).

A.7.3  DERIVATIONS OF $\mathbb{E}[n_{fp}]$ AND $\mathbb{E}[n_{fn}]$ FOR EACH METHOD

**Proposition 1.** *The closed-form of $\mathbb{E}[n_{fp}]$ and $\mathbb{E}[n_{fn}]$ for Random $\sigma_R$, First $\sigma_F$ and L-RHO can be found in Table 20 for a general $p^*_{fix}(i)$ and a linearly decreasing $p^*_{fix}(i)$ distribution (Assump. 1). Furthermore, ignoring an $\frac{m}{2}$ term in $\mathbb{E}[n^*_{fix}]$ for ease of exposition, The FPR and FNR of First $\sigma_F$ and Random $\sigma_R$ are $(\alpha_F, \beta_F) = (\frac{\sigma_F(1-b+\frac{m}{2}\sigma_F)}{1-b+\frac{m}{2}}, 1 - \frac{\sigma_F(b-\frac{m}{2}\sigma_F)}{b-\frac{m}{2}})$ and $(\alpha_R, \beta_R) = (\sigma_R, 1 - \sigma_R)$.*

Table 20: Closed-form expressions of $\mathbb{E}[n_{fp}]$ and $\mathbb{E}[n_{fn}]$ for Default, Random, First, and the learning method L-RHO with False Positive and Negative Rates $(\alpha, \beta)$.

| | Random $\sigma_R$ | First $\sigma_F$ | L-RHO $(\alpha, \beta)$ |
|---|---|---|---|
| | General $p^*_{fix}(i)$ | | |
| $\mathbb{E}[n_{fp}]$ | $\sigma_R \sum\limits_{1 \le i \le H-S}(1 - p^*_{fix}(i))$ | $\sum\limits_{1 \le i \le \sigma_F(H-S)}(1 - p^*_{fix}(i))$ | $\alpha \sum\limits_{1 \le i \le H-S}(1 - p^*_{fix}(i))$ |
| $\mathbb{E}[n_{fn}]$ | $(1 - \sigma_R) \sum\limits_{1 \le i \le H-S} p^*_{fix}(i)$ | $\sum\limits_{\sigma_F(H-S)+1 \le i \le (H-S)} p^*_{fix}(i)$ | $\beta \sum\limits_{1 \le i \le H-S} p^*_{fix}(i)$ |
| | Linear Decreasing $p^*_{fix}(i) = b - m\frac{i}{H-S}$ | | |
| $\mathbb{E}[n_{fp}]$ | $\sigma_R[(1 - b + \frac{m}{2})(H-S) + \frac{m}{2}]$ | $\sigma_F[(1 - b + \frac{m}{2}\sigma_F)(H-S) + \frac{m}{2}]$ | $\alpha[(1 - b + \frac{m}{2})(H-S) + \frac{m}{2}]$ |
| $\mathbb{E}[n_{fn}]$ | $(1 - \sigma_R)[(b - \frac{m}{2})(H-S) - \frac{m}{2}]$ | $(1 - \sigma_F)[(b - \frac{m}{2} - \frac{m}{2}\sigma_F)(H-S) - \frac{m}{2}]$ | $\beta[(b - \frac{m}{2})(H-S) - \frac{m}{2}]$ |
| | Equivalently, Given $\mathbb{E}[n^*_{fix}] = \sum_i p^*_{fix}(i) = (b - \frac{m}{2})(H-S) - \frac{m}{2}$ | | |
| $\mathbb{E}[n_{fp}]$ | $\sigma_R\left(H - S - \mathbb{E}[n^*_{fix}]\right)$ | $\sigma_F\left(H - S - \mathbb{E}[n^*_{fix}] - \frac{m}{2}(1 - \sigma_F)(H-S)\right)$ | $\alpha\left(H - S - \mathbb{E}[n^*_{fix}]\right)$ |
| $\mathbb{E}[n_{fn}]$ | $(1 - \sigma_R)\mathbb{E}[n^*_{fix}]$ | $(1 - \sigma_F)\left(\mathbb{E}[n^*_{fix}] - \frac{m}{2}\sigma_R(H-S)\right)$ | $\beta\mathbb{E}[n^*_{fix}]$ |

*Proof.* See Appendix A.7.5 for the detailed derivation of Table 20. Since we have $\mathbb{E}[n^*_{fix}] = \sum_i p^*_{fix}(i) = (b - \frac{m}{2})(H - S) - \frac{m}{2}$, we can substitute the corresponding terms in the table with $\mathbb{E}[n^*_{fix}]$ and arrive at the last equivalent closed-forms as in the main paper Prop. 1.

Ignoring the $\frac{m}{2} \in [0, 1]$ term in $\mathbb{E}[n^*_{fix}]$, the coordinate transformation from the FP and FN errors $(\mathbb{E}[n_{fp}], \mathbb{E}[n_{fn}])$ to the FP and FN rates $(\alpha, \beta)$ involves scaling the axes by $H - S - \mathbb{E}[n^*_{fix}] = (1 - b + \frac{m}{2})(H - S)$ and $\mathbb{E}[n^*_{fix}] = (b - \frac{m}{2})(H - S)$, respectively, from which we obtain the FPR and FNR for First $\sigma_F$ and Random $\sigma_R$. For example,

$$\beta_F = \frac{\mathbb{E}[n_{fn}]}{\mathbb{E}[n^*_{fix}]} = \frac{(1 - \sigma_F)(b - \frac{m}{2} - \frac{m}{2}\sigma_F)(H - S)}{(b - \frac{m}{2})(H - S)} = \frac{b - \frac{m}{2} - \frac{m}{2}\sigma_F - b\sigma_F + \frac{m}{s}\sigma_F + \frac{m}{2}\sigma_F^2}{b - \frac{m}{2}}$$

$$= \frac{b - \frac{m}{2} - b\sigma_F + \frac{m}{2}\sigma_F^2}{b - \frac{m}{2}} = 1 - \frac{\sigma_F(b - \frac{m}{2}\sigma_F)}{b - \frac{m}{2}}.$$

$\square$

### A.7.4 QUALITATIVE RELATIONSHIP BETWEEN ($\mathbb{E}[n_{fp}]$ AND $\mathbb{E}[n_{fn}]$) AND (OBJECTIVE, SOLVE TIME)

Based on empirical observation and analytical insights, we interpret the effects of $\mathbb{E}[n_{fp}]$ and $\mathbb{E}[n_{fn}]$ on the objective and solve time in the table below (and also in Fig. 4 left), where $+$ and $-$ reflects positive and negative correlations, respectively.

| | Objective | Solve Time |
|---|---|---|
| $\mathbb{E}[n_{fp}]$ | $+$ | $+$ (small $\mathbb{E}[n_{fp}]$) 
 or 
 $-$ (large $\mathbb{E}[n_{fp}]$) |
| $\mathbb{E}[n_{fn}]$ | $+$ | $+$ |

Specifically,

- The objective is significantly affected by $\mathbb{E}[n_{fp}]$, the number of operations with incorrectly fixed machine assignments. A higher $\mathbb{E}[n_{fp}]$ indicates fixing more sub-optimal decision variables in the restricted subproblem $\hat{P}_r$, and hence worsens the objective value (often exponentially, as observed in Appendix A.6 when $\sigma$ increases for Random / First). *Therefore, reducing $\mathbb{E}[n_{fp}]$ improves the objective.*

- The solve time is heavily influenced by $\mathbb{E}[n_{fn}]$, the number of operations that have the same machine assignments but are failed to be identified. A higher $\mathbb{E}[n_{fn}]$ indicates a larger search space of the optimization subproblem $\hat{P}_r$, hence increasing the solve time (often exponentially, as observed when $\sigma$ decreases for Random / First). *Hence, a lower $\mathbb{E}[n_{fn}]$ reduces the solve time.*

- $\mathbb{E}[n_{fp}]$ has a mixed contribution to the solve time. On one hand, an increasing $\mathbb{E}[n_{fp}]$ reduces the size of the subproblem $\hat{P}_r$, which could reduce the solve time; on the other hand, an increasing $\mathbb{E}[n_{fp}]$ leads to more sub-optimal assignment fixing in $\hat{P}_r$, which potentially complicates the optimization landscapes and hence can increase the solve time, especially when $\mathbb{E}[n_{fp}]$ is small. *Generally speaking, we observe that a higher $\mathbb{E}[n_{fp}]$ slightly increases the solve time in the low $\mathbb{E}[n_{fp}]$ regime (positive correlation) but reduces the solve time when $\mathbb{E}[n_{fp}]$ is close to $1$ (negative correlation).* Specifically, in Appendix A.6, we see that when $\mathbb{E}[n_{fp}]$ is low, L-RHO typically solves faster than First $\sigma_f$ baselines with a similar $\mathbb{E}[n_{fn}]$ but a higher $\mathbb{E}[n_{fp}]$; likewise, Oracle generally has a similar solve time as First $80\%$, whose $\mathbb{E}[n_{fp}]$ is much higher than Oracle.

- *Reducing $\mathbb{E}[n_{fn}]$ can improve the objective under a given time limit*, as the correctly restricted subproblem $\hat{P}_r$ from a low $\mathbb{E}[n_{fn}]$ can simplify the optimization landscape. This effect is especially evident in the low $\mathbb{E}[n_{fp}]$ regime, where the baseline RHO methods often reach the time limit with a suboptimal solution, whereas our L-RHO with a lower $\mathbb{E}[n_{fn}]$ can often solve the subproblem $\hat{P}_r$ optimally with a shorter solve time.

*Therefore, to achieve objectives comparable to Default RHO while reducing the solve time, an effective RHO method should first **reduce** $\mathbb{E}[\mathbf{n_{fp}}]$ to mitigate objective degradation, and subsequently **reduce** $\mathbb{E}[\mathbf{n_{fn}}]$, which can further reduces both solve time and objective in the small $\mathbb{E}[n_{fp}]$ regime.*

**More Visualizations.** We provide additional visualizations of different FJSP variants in Fig. 10, using the same plotting procedure as in Fig. 4 (Right). The $(\alpha, \beta)$ of L-RHO is selected as the False Positive and Negative Rates on the validation set $\mathcal{K}_{val}$. Empirically, our L-RHO outperforms all baseline methods (Random, First, Default RHO) with a lower objective and has a similar solve time as First $60\%$; the pink regions in Fig. 10 illustrates the empirical dominance relationship of L-RHO over the respective First and Random baselines. Based on the above analysis, the effectiveness of our learning method L-RHO is due to its ability to simultaneously achieve a small $\mathbb{E}[n_{fp}]$ (which ensures a low objective) and a small $\mathbb{E}[n_{fn}]$ (which improves both the objective and solve time). In contrast, the heuristic baselines Random or First trades-off $\mathbb{E}[n_{fn}]$ and $\mathbb{E}[n_{fp}]$ by adjusting $\sigma$, but cannot obtain a low $\mathbb{E}[n_{fn}]$ and a $\mathbb{E}[n_{fp}]$ at the same time.

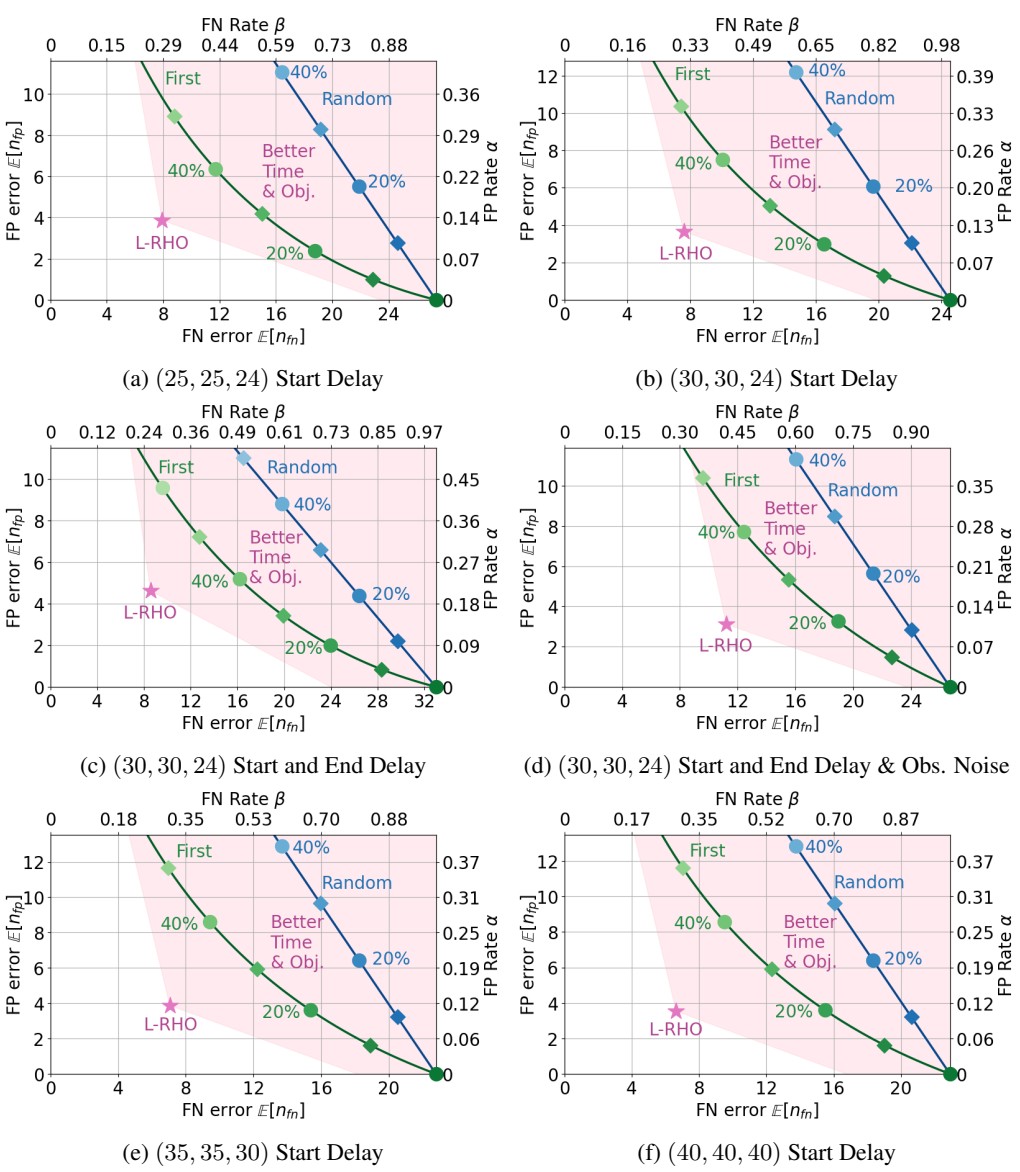

Figure 10: For a variety of FJSP distributions, we depict $\mathbb{E}[n_{fn}]$ and $\mathbb{E}[n_{fp}]$ of Random, First, and L-RHO in the low $\mathbb{E}[n_{fp}]$ region, using empirically validated $p^*_{fix}(i)$ distribution from Fig. 9 and L-RHO FPR-FNR $(\alpha, \beta)$ on the validation set. The pink region illustrates the empirical dominating region where our L-RHO has a better solve time and objective than all the baseline methods in the region; in particular, L-RHO has a better objective than all baseline methods, and a better solve time for a large range of $\sigma$ values (up to First $50\% - 60\%$), as it can achieve low $\mathbb{E}[n_{fp}]$ and $\mathbb{E}[n_{fn}]$ simultaneously. Additionally, we can transform the coordinates to the FP and FN rates $(\alpha, \beta)$, as depicted by the right and top axes, to provide insights on what FP and FN rates should L-RHO achieve for learning to be effective.

### A.7.5 DETAILED DERIVATION OF TABLE 20.

*Proof.* **(1) General $p^*_{fix}(i)$ distribution.**

- **Random $\sigma_R$:** by independence from the i.i.d. random sampling procedure, we have

$$\mathbb{E}[n^{\text{Random }\sigma_R}_{fp,r}] = \sum_{1 \leq i \leq H-S} Pr(\text{operation } O^i \in \mathcal{O}_{fix,r} \text{ and } i \notin \mathcal{O}^*_{fix,r}),$$

$$= \sum_{1 \leq i \leq H-S} Pr(\text{operation } O^{(i)} \in \mathcal{O}_{fix,r})(1 - Pr(O^{(i)} \in \mathcal{O}^*_{fix,r}))$$

$$= \sum_{1 \leq i \leq H-S} \sigma_R(1 - p^*_{fix}(i)),$$

$$\mathbb{E}[n^{\text{Random }\sigma_R}_{fn,r}] = \sum_{1 \leq i \leq H-S} Pr(\text{operation } O^{(i)} \notin \mathcal{O}_{fix,r} \text{ and } O^{(i)} \in \mathcal{O}^*_{fix,r})$$

$$= \sum_{1 \leq i \leq H-S} Pr(\text{operation } O^{(i)} \notin \mathcal{O}_{fix,r}) Pr(O^{(i)} \in \mathcal{O}^*_{fix,r})$$

$$= \sum_{1 \leq i \leq H-S} (1 - \sigma_R) p^*_{fix}(i),$$

where the sum is over all overlapping operations in $\mathcal{O}_{overlap,r}$.

- **First $\sigma_F$:** as it always selects the first $\sigma_F(H-S)$ operations, we have

$$\mathbb{E}[n^{\text{First }\sigma_F}_{fp,r}] = \sum_{1 \leq i \leq \sigma_F(H-S)} Pr(\text{operation } O^{(i)} \notin \mathcal{O}^*_{fix,r}) = \sum_{1 \leq i \leq \sigma_F(H-S)} (1 - p^*_{fix}(i)),$$

$$\mathbb{E}[n^{\text{First }\sigma_F}_{fn,r}] = \sum_{\sigma_F(H-S) \leq i \leq (H-S)} Pr(\text{operation } O^{(i)} \in \mathcal{O}^*_{fix,r}) = \sum_{\sigma_F(H-S) \leq i \leq (H-S)} p^*_{fix}(i),$$

where the sum is over the first $k(H-S)$ overlapping operations in $\mathcal{O}_{overlap,r}$.

- **L-RHO $(\alpha, \beta)$:** Given the the relationship between FP, FN rates and FP, FN errors, we have

$$\mathbb{E}[n^{\text{L-RHO}}_{fp,r}] = \alpha(H - S - \mathbb{E}[n^*_{fix,r}]) = \sum_{1 \leq i \leq H-S} \alpha(1 - p^*_{fix}(i)),$$

$$\mathbb{E}[n^{\text{L-RHO}}_{fn,r}] = \beta \mathbb{E}[n^*_{fix,r}] = \sum_{1 \leq i \leq H-S} \beta \, p^*_{fix}(i).$$

**(2) Linearly decreasing $p^*_{fix}(i) = b - m \cdot \frac{i}{H-S}$.**

We have $1 - p^*_{fix}(i) = (1 - b) + m \cdot \frac{i}{H-S}$, and we first compute the following quantities

$$\sum_{1 \leq i \leq H-S} \frac{i}{H-S} = \frac{1+H-S}{2}; \qquad \sum_{1 \leq i \leq \sigma_F(H-S)} \frac{i}{H-S} = \frac{\sigma_F + \sigma_F^2(H-S)}{2}$$

$$\sum_{\sigma_F(H-S) \leq i \leq (H-S)} \frac{i}{H-S} = \sum_{1 \leq i \leq H-S} \frac{i}{H-S} - \sum_{1 \leq i \leq \sigma_F(H-S)} \frac{i}{H-S} = \frac{(1 - \sigma_F) + (1 - \sigma_F^2)(H-S)}{2}$$

$$\sum_{1 \leq i \leq H-S} p^*_{fix}(i) = \sum_{1 \leq i \leq H-S} b - m \cdot \frac{i}{H-S} = (b - \frac{m}{2})(H-S) - \frac{m}{2}$$

$$\sum_{1 \leq i \leq \sigma_F(H-S)} p^*_{fix}(i) = \sum_{1 \leq i \leq \sigma_F(H-S)} b - m \cdot \frac{i}{H-S} = \sigma_F(b - \frac{m}{2}\sigma_F)(H-S) - \sigma_F\frac{m}{2}$$

$$\sum_{\sigma_F(H-S)+1 \leq i \leq (H-S)} p^*_{fix}(i) = (1 - \sigma_F)(b - \frac{m}{2} - \frac{m}{2}\sigma_F)(H-S) - (1 - \sigma_F)\frac{m}{2}$$

$$\sum_{1 \leq i \leq H-S} (1 - p^*_{fix}(i)) = \sum_{1 \leq i \leq (H-S)} (1 - b) + m \cdot \frac{i}{H-S} = (1 - b + \frac{m}{2})(H-S) + \frac{m}{2}$$

$$\sum_{1 \leq i \leq \sigma_F(H-S)} (1 - p^*_{fix}(i)) = \sum_{1 \leq i \leq \sigma_F(H-S)} (1 - b) + m \cdot \frac{i}{H-S} = \sigma_F(1 - b + \frac{m}{2}\sigma_F)(H-S) + \sigma_F\frac{m}{2}$$

We hence have

- **Random $\sigma_R$:**

$$\mathbb{E}[n_{fp}] = \sum_{1 \leq i \leq H-S} \sigma_R(1 - p^*_{fix}(i)) = \sigma_R(1 - b + \frac{m}{2})(H - S) + \sigma_R\frac{m}{2},$$

$$\mathbb{E}[n_{fn}] = \sum_{1 \leq i \leq H-S} (1 - \sigma_R)p^*_{fix}(i) = (1 - \sigma_R)(b - \frac{m}{2})(H - S) - (1 - \sigma_R)\frac{m}{2}.$$

- **First $\sigma_F$:**

$$\mathbb{E}[n_{fp,r}^{\text{First } \sigma_F}] = \sum_{1 \leq i \leq \sigma_F(H-S)} (1 - p^*_{fix}(i)) = \sigma_F(1 - b + \frac{m}{2}\sigma_F)(H - S) + \sigma_F\frac{m}{2},$$

$$\mathbb{E}[n_{fn,r}^{\text{First } \sigma_F}] = \sum_{\sigma_F(H-S)+1 \leq i \leq (H-S)} p^*_{fix}(i) = (1 - \sigma_F)(b - \frac{m}{2} - \frac{m}{2}\sigma_F)(H - S) - (1 - \sigma_F)\frac{m}{2}.$$

- **L-RHO $(\alpha, \beta)$:**

$$\mathbb{E}[n_{fp,r}^{\text{L-RHO}}] = \sum_{1 \leq i \leq H-S} \alpha(1 - p^*_{fix}(i)) = \alpha(1 - b + \frac{m}{2})(H - S) + \alpha\frac{m}{2},$$

$$\mathbb{E}[n_{fn,r}^{\text{L-RHO}}] = \sum_{1 \leq i \leq H-S} \beta\, p^*_{fix}(i) = \beta(b - \frac{m}{2})(H - S) - \beta\frac{m}{2}.$$

$\square$

### A.7.6 Interpretation of Prop. 1

We obtain the following insights by interpreting Prop. 1 .

1. The $p^*_{fix}(i)$ distribution determines the strength of the Random and First baselines and thus the potential for improvement by learning methods. Specifically, $\mathbb{E}[n^*_{fix}]$ governs the performance of Random: when $\mathbb{E}[n^*_{fix}] \approx 0.5$, Random suffers from either high FP error or high FN error across all $\sigma_R$, but when $\mathbb{E}[n^*_{fix}] = 0$ (or 1), then Random $\sigma_R = 0$ (or 100%) can simultaneously achieve zero FP and FN errors. Moreover, the magnitude of the slope $m$ indicates how much First can improve over Random, with a larger $m$ leading to greater improvement (Fig. 4 (middle)).

2. The qualitative relationship between the FP & FN errors and objective & solve time is illustrated in Fig. 4 (left) and further elaborated in Appendix A.7.4. Specifically, the FP error represents sub-optimal assignments incorrectly fixed in $\hat{P}$, which harms the objective, whereas the FN error corresponds to (near)-optimal assignments failed to be fixed in $\hat{P}$, which negatively impacts both metrics (solve time and objective) when a time limit is imposed. Thus, effective RHO methods should maintain low a FP error for a competitive objective and a low FN error to further improve both metrics; this is indeed achieved by our L-RHO, as evident in Fig. 4 (right).

3. We can further convert the FP and FN errors ($\mathbb{E}[n_{fp}], \mathbb{E}[n_{fn}]$) into the FP and FN rates ($\alpha, \beta$), as shown by the right and top axes in Fig. 4 (right). In this case, $\alpha_F$ and $\beta_F$ in Prop. 1 can be interpreted as the FPR and FNR that L-RHO's learned model should achieve to match the performance of First $\sigma_F$. This provides a valuable estimate to identify what $(\alpha, \beta)$ should we aim for when training L-RHO and to assess the effectiveness of the learned model before deployment.

