# OpenReview forum: "Learning-Guided Rolling Horizon Optimization for Long-Horizon Flexible Job-Shop Scheduling"
_ICLR.cc/2025/Conference — ICLR 2025 Poster_

### Official Review · Reviewer_PSPu · 2024-11-01

**Soundness:** 2
**Presentation:** 2
**Contribution:** 2
**Rating:** 8
**Confidence:** 4

**Summary:**

This paper presents L-RHO, the first learning-guided Rolling Horizon Optimization (RHO) framework for long-horizon FJSP.  Empirically, this paper shows that L-RHO accelerates RHO by up to 54% while showing significantly improved solution quality, enabling it to outperform other heuristic and learning-based baselines.

**Strengths:**

1. This paper claims to be the first learning-based framework to guide RHO for long-horizon Combinatorial Optimization Problems (COP).
2. In their experiments, L-RHO reduces RHO solve times by up to 54% and improves solution quality by up to 21% compared to various heuristic and learning-based baselines, across a range of FJSP settings and distributions.
3. Test on large instances, which operations are over 600, much larger than previous benchmarks.

**Weaknesses:**

**The novelty for learning-based RHO is limited.** L-RHO is very similar to RHO except for the customized attention based model. When compared to RHO, the experiment results (in Table 1) show only a slight improvement (less than 3% in Table 1 and also for the issue of large operations mentioned in the next item).

**The experiments are unconvincing.**
1. First, more comparisons to others are expected. For example, [1] and [2].
2. Second, although it seems impressive to perform well for “long-horizon” FJSP (e.g., 2000 operations in Table 1), unfortunately it is commonly agreed in this area that the more operations, the better performance. For example, in [3], the performance is nearly the optimal for large operations.

[1] Echeverria, I., Murua, M., & Santana, R. (2023). Solving large flexible job shop scheduling instances by generating a diverse set of scheduling policies with deep reinforcement learning. arXiv preprint arXiv:2310.15706.

[2] Han, B., & Yang, J. (2021). A deep reinforcement learning based solution for flexible job shop scheduling problem. International Journal of Simulation Modelling, 20(2), 375-386.

[3] Ho, K. H., Cheng, J. Y., Wu, J. H., Chiang, F., Chen, Y. C., Wu, Y. Y., & Wu, I. C. (2024). Residual scheduling: A new reinforcement learning approach to solving job shop scheduling problem. IEEE Access.

3. Third, from the above argument, it is also interesting and important to see the performances for some traditional benchmarks, such as those mentioned in line 289 (Wang et al., 2023a; Behnke and Geiger, 2012; Dauzere-P ` er´ es and Paulli ` , 1997; Hurink et al., 1994; Brandimarte, 1993).
4. Fourth, for CP-SAT or GA, as the problem size increases, the required time for CP-SAT and GA apparently grows a lot. It seems meaningless to compare these methods with 1800 second computation.

**Unclear theoretical analysis**
In the section of theoretical analysis, the authors present notations and formulas of “Assump 1.” and “Prop.1” without interpretation. Some are not defined, such as $n_{fn}$, $n_{pn}$, $n_{fix}$, etc. Note that they leave partial interpretation to Appendix A.5, which readers are not obligated to read. In the main text, it still needs to be self-contained.

**Minor comments.**
1. Symbol is inconsistent: \Tau in Section 3 (e.g.: line 130) and 5 (e.g.: line 267) means jobs set, but in Section 4 (e.g.: line 235), it means machines.
2. Figure 4 should be presented in the main text.
3. About "Machine/Operation Concatenation" (In Section LEARNING-GUIDED ROLLING HORIZON OPTIMIZATION), it needs to add " For each new operation that does not have a machine assignment, the concatenated machine feature is a dummy all-zero feature ", as mentioned in Appendix.

**Questions:**

1. Look-ahead oracle is unclear. An introduction and framework should be added, probably in appendix.
2. What is the value of the weight (W_pos) of loss function?

---

> ### Author Response · Authors · 2024-11-22
> **Response to Reviewer PSPu (1/4)**
>
> We thank the reviewer for recognizing our L-RHO as the first learning-based framework for long-horizon COP with significantly more operations than previous benchmarks. We understand that the main concerns are regarding additional experiments and the significance of the performance. We hope our response below will clarify any misunderstandings and concerns about our work.
>
> ----------
>
> ### **[Novelty and Significance of L-RHO]**
> We would like to first clarify that our main contribution is not about the neural network design, but **the idea that by learning which overlapping solutions in consecutive RHO iterations do not need to be re-optimized, our method substantially reduces RHO subproblem sizes and thus accelerates their solve time**. That is, we are the first to introduce learning to RHO, and we find the combined algorithm highly effective for solving long-horizon COPs such as the exemplar FJSP. Your comment prompted us to revise and clarify our contributions (General Response 1/4), so we are grateful for that.
>
> We kindly refer the reviewer to our responses **[Restating our contributions and impact]** **[Significance of L-RHO Results]** in the general response above for further discussion. Regarding the significance of the performance gains, we highlight the following points:
>
>
> 1. **L-RHO significantly accelerate the default RHO by 35-54% while enhancing its performance by 3-21%.**
> 2. **L-RHO consistently delivers state-of-the-art results across diverse scheduling variants.**
> 3. **L-RHO outperforms DRL baselines with less information and broader applicability.**
> 4. **L-RHO excels in its simplicity and effectiveness**, outperforming numerous complex baselines across a range of practical FJSP variations.
> 5. **L-RHO can have real-world impact.** As discussed in the general response, L-RHO achieves a 2.7% gap lower than the DRL baseline on real-world instances. While the number may look small, performance improvements that approach optimality are indeed challenging. As such, a 2.7% performance improvement is significant, and is typical for novel research in this area. In addition, such 2.7% improvement can translate in practice to a 2.7% revenue increase—an enormous benefit for companies managing billion-dollar operations.

---

> ### Author Response · Authors · 2024-11-22
> **Response to Reviewer PSPu (2/4)**
>
> ### **[More Comparisons to Other Baselines]**
> Thank you for suggesting other DRL baselines to compare! Following your suggestions, we compare with the two suggested baselines [1] and [3] since we are not able to find the source code for [2]. **We both directly load [1] and [3]’s checkpoint to evaluate the performance and retrain the model with the provided source code (see the table below)*. Unfortunately, we find [1] and [3]’s performance significantly worse than the DRL method that we benchmark in the original paper. We note that [1]’s greedy solve time is not as competitive as other DRL methods as it trains a suite of ensemble models and take the best one as the final result.**
>
> \* *We note that, when we tried to retrain the model in [1] and [3], we find that the training is very inefficient . For example, Echeverria et al.’s code uses Bayesian Optimization to train multiple models and select a subset of models for testing based on performance dominance; [3] uses a heuristic dispatch rule as the RL baseline, and the time for computing the RL baseline substantially increases for longer-horizon instances. In the paper of [1] and [3], they only learn on much smaller FJSP instances (up to 200 operations) with a large number of episodes (>100K). **However, we find that, when training on our FJSP distributions, training for 800-1000 episodes (the retrained setting we use) already takes 1.5-2 days. This makes the implementation of [1]-[3] not suitable for learning on longer-horizon FJSP instances.***
>
> [1] Solving large flexible job shop scheduling instances by generating a diverse set of scheduling policies with deep reinforcement learning (arXiv 2023)
>
> [2] A deep reinforcement learning based solution for flexible job shop scheduling problem. International Journal of Simulation Modelling (IJSIMM, 2021)
>
> [3] Residual scheduling: A new reinforcement learning approach to solving job shop scheduling problem (IEEE Access 2024)
>
> ----------
>
> ### **[CP-SAT with Prolonged Runtime]**
> Following your suggestion, we have further updated the result table to include CP-SAT results with a prolonged runtime of 36,000 seconds (10 hours). Even under this generous time limit, CP-SAT performs worse than RHO-based methods. We would also like to emphasize that the comparison with CP-SAT and GA highlights the importance of decomposition-based methods in solving large-scale, long-horizon FJSPs. **This further underscores the challenges of addressing long-horizon FJSPs with existing approaches and demonstrates the significant impact of our proposed L-RHO.**
>
>
> | | 600 (10, 20, 30) ||800 (10, 20, 40)||1200 (10, 20, 60)||2000 (10, 20, 100)|Transfer|
> |-|-|-|-|-|-|-|-|-|
> ||Time (s)|Makespan|Time (s)|Makespan|Time (s)|Makespan|Time (s)|Makespan|
> |CP-SAT (10 hours)|36000|1583 ± 65|36000|2128 ± 75|36000|3206 ± 87|36000|18821 ± 1986|
> |CP-SAT|1800| 2274 ± 147|1800| 4017 ± 413|1800| 10925 ± 1013|1800|39585 ± 2707|
> |**DRL-Echeverria [1]**|
> | 6 Ensemble Models - Greedy (Load Ckpt)|84 ± 27| 2347 ± 189 | 115 ± 32| 3105 ± 161|105 ± 7|4570 ± 218| 213 ± 10|7673 ± 356|
> | 6 Ensemble Models - Sampling (Load Ckpt)|157 ± 8| 2351 ± 139 | 217 ± 11| 3131 ± 165|427 ± 20|4657 ± 192| 603 ± 41| 7774 ± 271|
> | 4-6 Ensemble Models - Greedy (Retrained)|53 ± 3| 2426 ± 152 | 49 ± 3| 3147 ± 114|93 ± 6| 4758 ± 228|162 ± 13|7991 ± 455|
> | 4-6 Ensemble Models - Sampling (Retrained)|176 ± 8|2363 ± 160|222 ± 9|3103 ± 154|427 ± 20|4815 ± 273|470 ± 46|8012 ± 356   |
> |**DRL-Ho [3]**|
> | Greedy (Load Ckpt)| 4 ± 2| 3030 ± 69 | 6 ± 4| 4038 ±  87| 11 ± 6| 6045 ± 99| 22 ± 3| 10044 ± 115|
> | Sampling (Load Ckpt)| 174 ± 1| 2907 ± 46  | 168 ± 14| 3907 ± 56 | 385 ± 12| 5870 ± 71| 517 ± 10 |9848 ± 92|
> |Greedy (Retrained) |3± 0.4|3106 ± 77|9 ± 16|4041 ± 91|12 ± 4|6065 ± 91|26 ± 2|10073 ± 125|
> |Sampling (Retrained)|164 ± 5|3017 ± 57|190 ± 14|3990 ± 61|316 ± 45|6018 ± 65|565 ± 18|10032 ± 92|
> |**Default RHO**| 244 ± 21| 1558 ± 73|348 ± 26|2103 ± 78|545 ± 36|3136 ± 91|862 ± 42|5207 ± 114|
> |**L-RHO (450)**|126 ± 19|**1513 ± 70**|160 ± 23|**2015 ± 86**|259 ± 37|**3011 ± 106**|473 ± 52 |**4982 ± 132**|

---

> ### Author Response · Authors · 2024-11-22
> **Response to Reviewer PSPu (3/4)**
>
> ### **[Regarding “the more operations, the better performance”]**
>
> The reviewer commented that -
> > …, unfortunately it is commonly agreed in this area that the more operations, the better performance. For example, in [3], the performance is nearly the optimal for large operations.
>
> **We believe there may be a misunderstanding and kindly ask the reviewer to clarify this point.** Large-scale FJSP introduces significant complexity to problem-solving, as larger COPs lead to exponentially larger action spaces, making them more challenging than small-scale COPs. Moreover, we note that in [3], the majority of experiments focus on Job Shop Scheduling (without machine assignment) rather than FJSP stuided in this paper. For the more challenging FJSP, where machine assignment adds considerable complexity, they do not address large-scale scenarios, with their largest benchmark being only 200 operations (10, 20, 10)—significantly smaller than our setting of 600–2000 operations.
>
> **We further examined [3] in details, and find the performance gap in the paper is defined with respect to running OR-Tools for half a day (which is not optimal solution). When the problem scale increases, OR-Tools’ performance becomes significantly worse, making the “performance gap” smaller; however, this does not mean the performance is close to optimal as the comparison is not with respect to the optimal solution — as seen in the updated paper, we run OR-Tools for 10 hours in each setting, and we find OR-Tools’ performance significantly worse than RHO based methods.**
>
> ----------
>
> ### **[Performance on Benchmark Instances]**
> Thank you for the suggestion! We find most of the traditional benchmarks on small scale (< 500 operations). We thus benchmark the performance on Dauzere-Peres and Paulli, 1997, which is the largest traditional benchmarks available, containing 16 instances evenly divided to three different sizes (respectively, 196, 293, and 387 operations).  We exclude 4 instances from the dataset, which is simple and take < 2s for RHO methods to solve.  We further randomly sample with replacement $\times k$ operations for each job, with $k=3,6,9$ to mimic large scale settings with more operations: that is, if a job has 20 operations before, for $k=3$ we sample with replacement 60 operations within this set, where each operation maintains the same compatible machine and duration set as in the original dataset.
>
> We compare L-RHO with DRL-20K, the state-of-the-art deep learning method that we compare with in the submitted paper, and provide the performance comparison in the table below. For both methods, we directly transfer the model learned on 600 operations (10, 20, 30) setting to the real world instances. **The best known solution for the Dauzere original setting has an average makespan of 2120.**
>
> Our L-RHO outperforms DRL-20K in the real world dataset, achieving a close gap with respect to best known solution in the literature for the smallest non-augmented setting, and the performance advantage further holds for longer-horizon augmented settings. Furthermore, L-RHO significantly speeds up Default RHO in all settings, demonstrating the benefit of introducing learning to accelerate RHO. We further note there exists slight mismatch in the training distribution and the real world dataset, and it is an interesting future work is to train the model on a diverse set of distribution to further improve L-RHO’s transfer learning performance.
>
> || Dauzere Original| Size 196-387| Dauzere Aug 3.|Size 588-1161| Dauzere Aug. 6| Size 1176-2322  | Dauzere Aug. 9  | Size 1764-3483 |
> |-|-|-|-|-|-|-|-|-|
> ||Time (s)| Makespan| Time (s) |Makespan|Time (s)| Makespan| Time (s)| Makespan|
> |DRL-20K. - Greedy, Transfer |1.83 ± 0.1| 2493.1 ± 44.0 | 5.7 ± 0.5| 7279.0 ± 68.79  | 12.2 ± 1.0 | 14337.2 ± 218.1 | 21.6 ± 2.6      | 21217.7 ± 249.5 |
> | DRL-20K - Sample 10, Transfer | 8.10 ± 1.2| 2278.9 ± 18.7| 57.4 ± 9.6 | 6946.3 ± 82.5 | 234.7 ± 41.6   | 13935.3 ± 163.7         | 534.3 ±  95.2   | 20783.5 ± 225.8         |
> | DRL-20K - Sample 50, Transfer | 34.7 ± 6.1| 2253.3 ± 20.6| 321.2 ± 58.0 | 6887.0 ± 78.8    | 1341.4 ± 231.1 | 13884.2 ± 161.6         | 3014.3 ± 516.0| 20722.3 ± 215.6|
> | DRL-20K - Sample 1000, Transfer | 71.9 ± 12.5 | 2247.0 ± 19.9  | 670.2 ± 116.3  | 6857.6 ± 82.6 | 2679.1 ± 458.3 | 13859.1 ± 159.9   | 6067.6 ± 1038.4 | 20705.8 ± 217.0  |
> | Default RHO | 83.0 ± 33.2  | 2173.8 ± 67.8  | 289.7 ± 83.7   | 6504.5 ± 231.9 | 508.7 ± 150.3  | 13085.4 ± 427.7  | 793.5 ± 240.5   | 19688.7 ± 592.3 |
> | Warm Start RHO| 75.2 ± 24.5 | 2166.6 ± 67.9  | 241.2 ± 67.3   | 6490.0 ± 236.5 | 449.7 ± 141.1  | 13033.6 ± 431.8 | 683.6 ± 210.3   | 19589.3 ± 608.9  |
> | **L-RHO, Transfer** | 65.0 ± 22.5 | 2184.5 ± 65.8  | 199.1 ± 69.6   | 6498.8 ± 254.0  | 344.8 ± 133.9  | 13015.3 ± 503.6   | 492.1 ± 200.3   | 19592.8 ± 724.5|
> | **Best Known Solution**|   | **2120 .0 ± 78.27**|  | N/A due to augmentation |    | N/A due to augmentation |    | N/A due to augmentation |

---

> ### Author Response · Authors · 2024-11-22
> **Response to Reviewer PSPu (4/4)**
>
> ### **[Unclear Theoretical Analysis]**
> Sorry for the confusion. Additional discussions on the theoretical analysis were originally included in the appendix due to space constraint. We agree that the main paper should clearly illustrate the findings. Therefore, we have followed the suggestions and updated the main text to improve the clarity of the theoretical analysis. We also note that in the updated paper, the False Positive and Negative Errors, $n_{fp}$ and $n_{fn}$, are defined in the notation section.
>
> Specifically, we provide expanded interpretations of the analysis, including:
>
> - At the beginning of the section, we introduce the high level idea of comparing different RHO variants by analyzing their False Positive (FP) and Negative (FN) Errors. We also explain the effect of FP and FN on the solve time and objective.
> - Before Assumption 1, we explain the intuition of the linear decay assumption, which is used in Prop. 1.
> - At the end of the section, we interpret Prop. 1 and provide the following insights from the analysis: (i) the performance of the RHO heuristics (Random and First) is affected by the Oracle distribution $p^*_{fix}$, (ii) The linear decay’s strength further affects how much the First RHO can improve from Random RHO, and (iii) the theory further reveals the FP and FN rates that L-RHO needs to achieve to outperform the RHO baselines.
>
> ----------
> ### **[Look-Ahead Oracle (Q1)]**
> Thank you for the suggestion! We have provided more details in Appendix Table 4: see $O^*_{fix, r}$ under operation subset and Oracle under Methods.
>
> ----------
>
> ### **[Value of Weight W_pos (Q2)]**
> We set $w_{pos} = 0.5$ in our experiments, as noted in Table 11 of the Appendix. A lower $w_{pos}$ prioritizes avoiding False Positives, where the learned model incorrectly fixes operations with changing assignments. The lowered $w_{pos}$ can improve the objective but may increase solve time, as explained in our analysis.
>
> For the rest of your minor comments, we have carefully followed your suggestions and revised the submission accordingly. We hope our detailed responses, along with the extensive new results and new comparisons, address your concerns!

---

> > ### Author Response · Authors · 2024-11-29
> > **Looking Forward to Your Reply**
> >
> > Dear Reviewer PSPu,
> >
> > Thank you again for your valuable suggestions. We have conducted detailed experiments and provided thorough discussions to address your concerns. As the rebuttal period is nearing its conclusion, we wanted to follow up to ensure our responses have adequately addressed the reviewers' concerns. Please let us know if there are any remaining questions or areas where further clarification is needed. We sincerely appreciate the time and effort you have dedicated to reviewing our work. Thank you!
> >
> > Best,
> >
> > Authors

---

> ### Comment · Reviewer_PSPu · 2024-12-02
>
> I am glad that authors addressed most of my concerns. In addition, the performance reaches sota, after comparing some more sota works. Therefore, I am happy to raise the score up to 8.

---

> > ### Author Response · Authors · 2024-12-02
> > **Thank you for your strong support!**
> >
> > We sincerely thank the reviewer for acknowledging our rebuttal and recognizing that L-RHO achieves state-of-the-art performance. We deeply appreciate your constructive comments and strong support for our paper. Thank you!

---

### Official Review · Reviewer_ppet · 2024-11-03

**Soundness:** 4
**Presentation:** 2
**Contribution:** 3
**Rating:** 8
**Confidence:** 3

**Summary:**

Rolling horizon optimization (RHO) [1] is a widely used approach for tackling long-horizon production planning problems in stochastic environments. RHO solves these problems by breaking them into overlapping, shorter-horizon subproblems that roll forward over time. This overlap ensures continuity between consecutive subproblems.

This paper proposes a machine learning framework to identify decision variables that should be predetermined based on the prior horizons, allowing them to be treated as fixed for the current horizon. This helps to reduce the size of the optimization problem for each horizon.
Experimental results show the proposed approach is not only fast but also result in better solutions.


1: Sethi, Suresh, and Gerhard Sorger. "A theory of rolling horizon decision making." Annals of Operations Research 29.1 (1991): 387-415.

**Strengths:**

**Significance**: This paper studies an important problem relevant in many production planning operations. The proposed approach seem to surpass the state of the art, a significant contribution.

**Originality**: I am not entirely familiar with the relevant literature. I searched articles published in the last five years. This work seems to take a novel direction to tackle the RHO problem.

**Quality**: Quality of the paper is good as it covers both theoretical and empirical analysis.

**Clarity**: The overall objective and methodology of the paper are easy to understand at a high level. However, some of the introduced notations are difficult to follow, and the intuitive explanations provided are sometimes lacking
(See weakness).

**Weaknesses:**

- $m$ is used to denote a machine $m \in M$ as well as an assignment $m: O \rightarrow M$ (line 133 and 134).

- How can default RHO be suboptimal compared to proposed L-RHO? If I  understood correctly, default RHO solves the true problem till optimality, hence it might be computation-intensive but should have the best performance. Is it due to the fact that all algorithms have been run with a timelimit? If that's the case, can default RHO perfrom better than L-RHO if both are allowed to run for more amount of runtime?

-  It seems that the accuracy, precision, and recall of the classifier were not reported. I expect these metrics to be high for the model to demonstrate good performance. I would recommend to report this in the paper to provide a clearer picture of the ML model's performance.

**Questions:**

- Can you provide intuitive explanation of $q^\star$ (line 219)? Is it the assignment which has most number of overlapping assignments? What is the significance of this? I do not get the motivation for setting $y_O^\star = y_O^{q^\star}$​. Could you provide some insight/intuition into the reasoning behind this?
- What does $w_{pos}$ represent?
- Can you comment on what features turn out to be important in identifying decision variables that should be fixed?
- When increasing the congestion level (line 424), did you train a new model or use the one trained with low congestion?
- Have you explored the attention map? Is it possible draw a significant conclusion form that? For example, an operation is more likely to be fixed if a machine has been running for a long time!
- It is mentioned in the text that the operations are divided into subproblems of fixed duration T. How did you predetermine the value of T in your experiments? Should not it depend on the execution time of the operations?

---

> ### Author Response · Authors · 2024-11-22
> **Response to Reviewer ppet (1/2)**
>
> We appreciate the reviewer’s positive and valuable feedback! We are delighted that the reviewer recognizes our paper’s significant contributions and novel direction. We hope the following response addresses any remaining concerns on clarity.
>
> ----------
>
> ### **[Notation $m$ (W1)]**
> Sorry for the confusion! We update the first notation to $\bar{m} \in \mathcal{M}_{j, k}^{\bar{m}}$.
>
> ----------
>
> ### **[Timelimit in RHO (W2)]**
> Yes, it is due to the time limit that we impose on the RHO algorithm. With significantly longer (albeit prohibitively long) time limit, the default RHO can achieve a better objective. To address this comprehensively, we have followed the suggestion and included additional results as follows. As detailed in Appendix A5.4, two parameters control the time limit per subproblem: 1) total time limit, where solving terminates if the time exceeds $T$=60s; 2) early stop time, where solving terminates early if the objective does not improve within $T_{es}$=3s. In the table below, we evaluate the default RHO with increased $T$ and $T_{es}$. The results show that, as the time limit increases by over 10x, the default RHO achieves better objectives than our L-RHO, albeit with prohibitively long runtime. Our L-RHO delivers superior performance under practical time constraints, offering benefits especially in online cases.
>
> |   **Default RHO**   | 600 (10, 20, 30) |           | 800 (10, 20, 40) |           | 1200 (10, 20, 60) |            |
> | ---------------------------------------- | ---------------- | --------- | ---------------- | --------- | ----------------- | ---------- |
> | Time Limit $T$, Early Stop Time $T_{es}$ | Time (s)         | Makespan  | Time (s)         | Makespan  | Time (s)          | Makespan   |
> | 60, 3 (reported)                         | 244 ± 21         | 1558 ± 73 | 348 ± 26         | 2103 ± 78 | 545 ± 36          | 3136 ± 91  |
> | 60, 10                                   | 599± 55          | 1529 ± 58 | 728 ± 98         | 2044 ± 75 | 1109 ± 108        | 3002 ± 87  |
> | 60, 30                                   | 923 ± 71         | 1502 ± 66 | 1099 ± 130       | 2028 ± 72 | 1697 ± 175        | 2963 ± 78  |
> | 60, 60                                   | 819 ± 117        | 1494 ± 67 | 1096 ± 156       | 2018 ± 71 | 1714 ± 154        | 2968 ± 89  |
> | 120, 60                                  | 1476 ± 312       | 1497 ± 70 | 1811 ± 241       | 1997 ± 70 | 2809 ± 308        | 2932 ± 75  |
> | 180, 120                                 | 2142 ± 461       | 1490 ± 70 | 2509 ± 346       | 1994 ± 76 | 3932 ± 590        | 2924 ± 83  |
> | 360, 180                                 | 3631 ± 816       | 1476 ± 64 | 3673 ± 797       | 1987 ± 68 | 5663 ± 900        | 2924 ± 86  |
> ||
> | **L-RHO**                                    | 126 ± 19         | 1513 ± 70 | 160 ± 23         | 2015 ± 86 | 259 ± 37          | 3011 ± 106 |
>
> ----------
>
> ### **[Report More Metrics (W3)]**
> Thank you for the great suggestion! We have provided the accuracy, precision, and recall in Appendix A.6.2. We copy the results here for clarity.
>
> |   | Accuracy | True Positive Rate | True Negative Rate | Precision | Recall |
> | ----------------- | -------- | ------------------ | ------------------ | --------- | ------ |
> |       |          |                    | **Table 1**          |           |        |
> | 600 (10, 20, 30)  | 0.77     | 0.81               | 0.67               | 0.86      | 0.81   |
> | 800 (10, 20, 40)  | 0.76     | 0.79               | 0.67               | 0.86      | 0.79   |
> | 1200 (10, 20, 60) | 0.76     | 0.77               | 0.73               | 0.87      | 0.77   |
> |                   |          |                    | **Table 2**           |           |        |
> | 625 (25, 25, 25)  | 0.78     | 0.71               | 0.85               | 0.83      | 0.71   |
> | 1225 (35, 35, 35) | 0.81     | 0.72               | 0.87               | 0.79      | 0.72   |
> | 1600 (40, 40, 40) | 0.81     | 0.69               | 0.90               | 0.83      | 0.69   |

---

> ### Author Response · Authors · 2024-11-22
> **Response to Reviewer ppet (2/2)**
>
> ### **[Intuitive explanation of** $q^*$ **and** $y_O^* = y_O^{q^*}$ **(Q1)]**
> Thank you for the question! Yes, it is the solution with most number of overlapping assignments. The intuition behind this choice is, typically, the solve time of the subproblem reduces when more operations are fixed. As we would like the learning method to reduce the solve time of Default RHO, this label seems to be a natural choice. One caveat is that, the solution quality (the objective value) may be tempered if too many operations are fixed incorrectly. In our experiments, we find that this choice of $q^*$ has decent solution quality and substantially reducing the subproblem solve time, and hence it is chosen by us.
>
> ----------
>
> ### **[What is $w_{pos}$? (Q2)]**
> The weight $w_{pos}$ adjusts the balance between reducing False Positive and False Negative Rate in learning. Specifically, a lower $w_{pos}$ emphasizes avoiding predicting False Positives, i.e., cases where the learned model incorrectly fixes operations whose true assignments change. The lowered $w_{pos}$ can improve the objective but may increase the solve time, as pointed out in our analysis. We set $w_{pos} = 0.5$ in our experiments.
>
> ----------
>
> ### **[Importance of the Features (Q3)]**
> Good question! Summarizing and interpreting the black-box neural model into white-box decision rules understandable by humans is indeed exciting yet non-trivial. Below, we provide some discussions to uncover some potential insights.
>
> Our input features are divided into three groups (1) **Problem Characteristics**: Features for each operation that capture the problem's inherent characteristics, such as the operation’s order within the same job (the id), average durations across all potential assignments, release time to the system (for delay based objectives); (2) **Previous Solution Information**: Features for each overlapping operation, such as whether it is overlapping, its machine assignment, solution start time, and solution end time; (3) **Machine Features**: Features specific to each machine, such as the number of overlapping operations assigned to the machine in the previous solution, and the end time of the machine to process all those operations.
>
> We find that the second group of the features currently is the most important - the network cannot learn if we disable this subset, whereas disabling the reset of the features leads to roughly 1%-2% accuracy reduction. This makes sense as decisions of fixing or not should be made based on the quality of the previous solution. Our preliminary exploration also seems to indicate that the features in the group are similarly important — we tried to train a model with only one feature out of this set enabled (and the rest masked to 0). In this case, we find that no single feature alone can successfully learn meaningful information. Given the limited time for the rebuttal, we will further explore this and include additional discussions in the final version.
>
> ----------
>
> ### **[Explore the Attention Maps (Q4)]**
> We are really grateful for the reviewer’s question, which lead us to identify an implementation issue in the submitted paper that makes the weights of the attention layer converge to 0 (so only the residual connection is effective). Please refer to our general response (4/4) for more details. We have fixed the implementation issue and take great effort to compare the reported architecture and the architecture with attention. We refer the reviewer to Appendix A.6.3 for the detailed results and visualizations of the attention map.
>
> ----------
>
> ### **[Setup for Different Congestion Levels]**
> We train a new model for each congestion level.
>
> ----------
>
> ### **[How to choose value $T$]**
> We first note that the subproblem solve time T is different from and irrelevant to the scheduling start and end time for the FJSP.  The operations are divided into subproblems of fixed size (H operations per subproblem), and we solve each subproblem with a solve time limit up to T seconds. As mentioned in Appendix A.5.4, we perform a parameter grid search to find the best parameter value.

---

> > ### Author Response · Authors · 2024-11-29
> > **Looking Forward to Your Reply**
> >
> > Dear Reviewer ppet,
> >
> > Thank you again for your valuable suggestions. We have conducted detailed experiments and provided thorough discussions to address your concerns. As the rebuttal period is nearing its conclusion, we wanted to follow up to ensure our responses have adequately addressed the reviewers' concerns. Please let us know if there are any remaining questions or areas where further clarification is needed. We sincerely appreciate the time and effort you have dedicated to reviewing our work. Thank you!
> >
> > Best,
> >
> > Authors

---

> > > ### Comment · Reviewer_ppet · 2024-11-30
> > >
> > > Thank you for answering in detail. As I mentioned before, I find this work very much exciting, promising.
> > > I am curious whether there is a recognized benchmark for testing job-shop scheduling algorithms. Specifically, is the source of the experimental setup in Tables 1 and 2 a commonly used benchmark  in the field? Are these instances real-world scenarios or synthetic cases designed for testing purposes? For a particular instance, for instance $600(10, 20, 30)$, do you need training instances of 10 machines, 20 jobs and 30 operations?
> > >
> > > In my experience, job-shop scheduling in real-world applications tends to be highly nuanced, often requiring the inclusion of numerous application-specific constraints. These additional constraints can significantly impact the performance of algorithms that otherwise perform well in standard benchmarks.
> > > Could you comment on the generalizability of the proposed approach to real-world applications?
> > >
> > > Thank you.

---

> > > > ### Author Response · Authors · 2024-12-02
> > > > **Thank you for your support and further discussion (1/2)**
> > > >
> > > > **We deeply appreciate the reviewer’s support for our work! Thank you for acknowledging that our work is exciting and promising. We further respond to your outlined questions below.**
> > > >
> > > > In short, the test instances we used are commonly used benchmarks from the literature, including **both synthetic ones with diverse variations mimicking real-world complexities** (in the original paper) and **established benchmark ones derived from real-world scenarios** (added during rebuttal). Notably, L-RHO achieves state-of-the-art performance both when trained and tested on synthetic datasets with various variations and in zero-shot generalization to real-world instances.
> > > >
> > > > ----------
> > > >
> > > > ### **[Benchmark on Synthetic Instances]**
> > > > **We first utilize synthetic instances that are randomly sampled from underlying data distributions, following common synthetic instance generation schemes as in the literature [1, 2, 3, 4]**. Specifically, we follow the state-of-the-art DRL constructive method's paper [1] to define the instance distribution in Table 1 as follows:
> > > >
> > > > - Each instance contains $|M|$ machines, $|T|$ jobs, and $n$ operations per job, resulting in a total of $T\times n$ operations. For example, in FJSP 600 (10, 20, 30), the training and test instances include 10 machines, 20 jobs, and 30 operations per job, totaling 20x30 = 600 operations.
> > > > - For each operation $O_{j, k}$, the number of compatible machines $M_{j, k} \subseteq M$  is uniformly sampled, with the size $|M_{j, k}| \sim U(1, |M|)$; the process duration of operation $O_{j, k}$ by each machine $\bar{m}$ is uniformly sampled as $p_{j,k}^{\bar{m}} \sim U(1, 99)$.
> > > > - We note that [1] considers small-scale FJSP with $n \leq 10$ operations per jobs, while we consider larger-scale, long-horizon FJSPs with 30, 40, 60, and 100 operations per job.
> > > >
> > > > **In Table 2 and Fig. 3, we take a step further by extending these synthetic instances to incorporate more real-world complexities such as delay objectives, congestion, noisy observations, and machine breakdowns.** This aligns precisely with the reviewer’s comments to demonstrate L-RHO’s generality across diverse setups. For example, in Table 2 and Fig. 3 (left, middle), we study the delay objectives (start / end delay), where we further sample operations’ release times (start delay), and target end times (end delay) uniformly at random, with the additional constraint that operations should be scheduled after their release times. Furthermore, in Fig. 3 (middle), the noisy observation is simulated by randomly sampling a subset of operations, and further perturbing their process durations $p_{j,k}^{\bar{m}}$ with random observation noise. Fig. 3 (right), we simulate a series of machine breakdown events happening at non-overlapping random time intervals; for each breakdown time interval, we further sample a set of machines, which become unavailable during the corresponding breakdown interval. Due to the space limitation, we kindly refer the reviewer to Appendix A.5.1 for a detailed setup.
> > > >
> > > > **Across all the above settings, the results demonstrate state-of-the-art performance for the proposed L-RHO when trained and tested on synthetic datasets.**
> > > >
> > > >
> > > > References:
> > > >
> > > > [1] Flexible job shop scheduling via dual attention network-based reinforcement learning. (IEEE Transactions on Neural Networks and Learning Systems, 2023)
> > > >
> > > > [2] Flexible job-shop scheduling via graph neural network and deep reinforcement learning. (IEEE Transactions on Industrial Informatics, 2022)
> > > >
> > > > [3] Test instances for the flexible job shop scheduling problem with work centers. (2012)
> > > >
> > > > [4] Mathematical modeling and heuristic approaches to flexible job shop scheduling problems. (Journal of intelligent manufacturing, 2007)

---

> ### Author Response · Authors · 2024-12-02
> **Thank you for your support and further discussion (2/2)**
>
> ### **[Zero-Shot Generalization to Real-World Benchmarks]**
>
> **We also evaluated the zero-shot generalization of the model trained on synthetic data by testing it directly on real-world benchmarks [5] with varying problem sizes ($|M|, |T|, n$) and data distribution. This demonstrates L-RHO's ability to generalize by training on one FJSP setup and solving others.** As mentioned in our general response 2/4 and in the newly added Table 15 (Appendix A.6.1), our L-RHO model trained on synthetic data with FJSP 600 (10, 20, 30) setup achieves a 2.7% lower gap than DRL-20K compared to the best-known solution in zero-shot generalization setup. It also delivers near-optimal performance in smaller settings, maintains its advantage in larger ones without fine-tuning, and accelerates Default RHO by 24%-38%, demonstrating its practical impact and the benefits of learning to enhance RHO.
>
>
> |  | Dauzere Original | Size 196-387 | Dauzere Augment x3 | Size 588-1161 | Dauzere Augment x6 | Size 1176-2322  | Dauzere Augment x9 | Size 1764-3483 |
> | - | - | - | - | - | - | - | - | - |
> |  | Time (s)| Makespan | Time (s)  | Makespan | Time (s)  | Makespan  | Time (s)  | Makespan  |
> | DRL-20K. - Greedy, Transfer | 1.83 ± 0.1 | 2493.1 ± 44.0 | 5.7 ± 0.5 | 7279.0 ± 68.79 | 12.2 ± 1.0 | 14337.2 ± 218.1 | 21.6 ± 2.6 | 21217.7 ± 249.5  |
> | DRL-20K - Sample 10, Transfer   | 8.10 ± 1.2  | 2278.9 ± 18.7 | 57.4 ± 9.6         | 6946.3 ± 82.5 | 234.7 ± 41.6  | 13935.3 ± 163.7| 534.3 ±  95.2 | 20783.5 ± 225.8  |
> | DRL-20K - Sample 50, Transfer | 34.7 ± 6.1 | 2253.3 ± 20.6  | 321.2 ± 58.0 | 6887.0 ± 78.8  | 1341.4 ± 231.1  | 13884.2 ± 161.6  | 3014.3 ± 516.0   | 20722.3 ± 215.6  |
> | DRL-20K - Sample 1000, Transfer | 71.9 ± 12.5 | 2247.0 ± 19.9  | 670.2 ± 116.3      | 6857.6 ± 82.6  | 2679.1 ± 458.3 | 13859.1 ± 159.9 | 6067.6 ± 1038.4  | 20705.8 ± 217.0  |
> | Default RHO | 83.0 ± 33.2  | 2173.8 ± 67.8  | 289.7 ± 83.7 | 6504.5 ± 231.9  | 508.7 ± 150.3 | 13085.4 ± 427.7 | 793.5 ± 240.5  | 19688.7 ± 592.3  |
> | Warm Start RHO   | 75.2 ± 24.5  | 2166.6 ± 67.9  | 241.2 ± 67.3 | 6490.0 ± 236.5  | 449.7 ± 141.1  | 13033.6 ± 431.8 | 683.6 ± 210.3 | 19589.3 ± 608.9 |
> | **L-RHO, Transfer** | 65.0 ± 22.5 | 2184.5 ± 65.8  | 199.1 ± 69.6  | 6498.8 ± 254.0 | 344.8 ± 133.9  | 13015.3 ± 503.6  | 492.1 ± 200.3  | 19592.8 ± 724.5 |
> | **Best Known Solution**  | | **2120 .0± 78.27** || N/A due to augmentation | | N/A due to augmentation |  | N/A due to augmentation |
>
>
> Reference:
>
> [5] An integrated approach for modeling and solving the general multiprocessor job-shop scheduling problem using tabu search. (Annals of Operations Research, 1997)
>
> ----------
>
> ### **[Generalizability of L-RHO to Practical Applications]**
> We fully agree with the reviewer that FJSPs in real-world applications are highly nuanced. This recognition motivates our detailed study in Sec. 5.2, where we explore a broader range of FJSP variations, including different objectives (start/end delay), noisy observations, and machine breakdowns. **These scenarios introduce new constraints, such as release times in start/end delay variants that require operations to be scheduled only after their release, and machine unavailability during breakdown periods.** In these diverse settings, L-RHO consistently outperforms baseline methods, achieving significant speedups over Default RHO.
>
> We believe learning-based approaches have significant potential to enhance RHO performance under real-world constraints. While solvers like CP-SAT are capable of handling practical constraints, they could face inefficiencies with non-standard variants, leading to longer solve times or suboptimal solutions within a fixed time limit. In these cases, learning to fix variables is particularly valuable, as it reduces problem size and simplifies subproblems for the solver. **This advantage is evident in Table 2, where L-RHO achieves greater improvements in solve time and objectives for delay-based problems compared to makespan-based ones in Table 1, demonstrating its superior ability to accelerate RHO on more complex FJSP variants.**
>
> Additionally, our theoretical analysis in Sec. 6 provides valuable insights into when learning can effectively enhance RHO's performance, and it can be adopted to analyze the effectiveness of deploying L-RHO in practice. As this analysis applies broadly to general FJSP settings, we believe it can provide practitioners useful guidance on whether to leverage learning to address real-world scenarios.
>
> ----------
>
> **We sincerely thank you for your valuable feedback in strengthening and supporting our work. We will further refine the paper based on our discussions here. Please feel free to follow up with any additional questions. We look forward to your further reply!**

---

> > ### Comment · Reviewer_ppet · 2024-12-03
> >
> > Thank you for this explanation, which emphasizes  L-RHO's generalizability in real-world practical applications. Obviously, it might not be effective in every usecase; but the authors tried and tested in existing benchmark experiments from the literature and also on real-world benchmarks.
> > These experimental evaluations convinces me to  increase my score.
> >
> > All the best.

---

> > > ### Author Response · Authors · 2024-12-03
> > > **Thank you for your strong support!**
> > >
> > > We sincerely thank the reviewer for recognizing our contribution and the promising results. Your constructive feedback during the rebuttal process has truly helped us strengthen our work, and we deeply appreciate it. We will refine the paper to reflect our latest discussion with the reviewer. Thank you for your strong support!

---

### Official Review · Reviewer_qP8G · 2024-11-04

**Soundness:** 3
**Presentation:** 2
**Contribution:** 3
**Rating:** 5
**Confidence:** 2

**Summary:**

This paper proposes L-RHO, a rolling horizon optimization algorithm designed for the flexible job-shop scheduling problem (FJSP) that leverages learning to select overlapping variables in subproblems. As the first study to guide the RHO algorithm with a learning-based approach, L-RHO demonstrated performance improvements across diverse FJSP settings.

**Strengths:**

This paper is noteworthy as an initial study that guides the decomposition process of subproblems in the RHO algorithm using a learning-based neural network. It also stands out for conducting comprehensive experiments and analyses across diverse FJSP settings.

**Weaknesses:**

The specific details about making RHO learnable are mostly presented in the supplementary material rather than the main sections, making it challenging to fully grasp the proposed learning framework. Given the extensive content of the paper, focusing more on the learning aspects in the main text would improve clarity and comprehension. Additionally, it is somewhat disappointing that in Table 1, the performance of L-RHO is not markedly superior to other baseline comparisons.

**Questions:**

1. Could the learning guide proposed in this study potentially be adapted for use in other algorithms beyond RHO?
2. Would the ideas proposed in this paper be adaptable for subproblem decomposition in various types of long-horizon combinatorial optimization problems?

---

> ### Author Response · Authors · 2024-11-22
> **Response to Reviewer qP8G (1/2)**
>
> We appreciate the reviewer's valuable feedback! We are pleased that the reviewer finds our paper to be noteworthy with comprehensive experiments and analyses across diverse FJSP settings. We understand your main concern about performance significance and hope our response below addresses it clearly.
>
> ----------
>
> ### **[Details of L-RHO]**
> Following your suggestion, we have carefully reorganized our paper to include more details of L-RHO in the main text. Specifically, we have moved the architecture figure to the main paper (Figure 2 in the updated version) to complement the high-level concept illustrated in Figure 1, which explains how we aim to learn which operations retain the same assignments across subsequent RHO subproblems. Other critical details about the learning process—such as training data collection, input feature summaries, the loss function, and inference steps—have already be included in the main paper. The Appendix primarily serves as a resource for completeness, providing step-by-step RHO pseudocode, implementation details for FJSP and L-RHO on varies settings to facilitate algorithm re-implementation, as well as additional experiments and theoretical proofs. We believe the updated main paper covers all critical information. However, if you have specific suggestions on areas to emphasize further, we would be happy to incorporate them. Additionally, our code will be made publicly available, and can also be found in the supplementary materials.
>
> ----------
>
> ### **[Significance of L-RHO Results]**
>
> Our L-RHO achieves state-of-the-art performance, outperforming various traditional and learning-based baselines both with and without decomposition in various challenging settings. This is significant for the following reasons:
>
> 1. **L-RHO significantly accelerate the default RHO by 35-54% while enhancing its performance by 3-21%**, as demonstrated in Tables 1 and 2. This represent a major advancement for long-horizon FJSP and we are the first to showcase this finding.
> 2. **L-RHO consistently delivers state-of-the-art results across diverse scheduling variants.** This is evidenced by Tables 1 and 2 and Figure 3, across different problem sizes, data distributions, objectives, under online observation noise and machine disruption.
> 3. **L-RHO outperforms DRL baselines with less information and broader applicability.** While the DRL baselines rely on complete problem information and is unsuitable for online problems, L-RHO operates effectively within a limited RHO window, achieving much better objectives with comparable or even shorter solve times.
> 4. **L-RHO excels in its simplicity and effectiveness.** Beyond merely competing on performance benchmarks, where algorithms in the literature are becoming increasingly complex, L-RHO stands out for its remarkable simplicity and effectiveness, outperforming numerous complex baselines across a range of pratical FJSP variations.
> 5.**L-RHO can have real-world impact.** L-RHO achieves a 2.7% gap lower than the DRL baseline on real-world instances. While the number may look small, performance improvements that approach optimality are indeed challenging. As such, a 2.7% performance improvement is significant, and is typical for novel research in this area [1, 2]. In addition, such 2.7% improvement can translate in practice to a 2.7% revenue increase—an enormous benefit for companies managing billion-dollar operations.
>
> [1] Learning Topological Representations with Bidirectional Graph Attention Network for Solving Job Shop Scheduling Problem (UAI 2024)
>
> [2] Deep Reinforcement Learning Guided Improvement Heuristic for Job Shop Scheduling (ICLR 2024)
>
> *Note the above two papers only study Job Shop Scheduling (JSP, no machine assignments).*
>
> Moreover, we have added several new baselines, including two additional state-of-the-art DRL methods, in the revised paper and have tested our L-RHO on real-world benchmark instances. Kindly refer to **[Updated Table 1 with More Baselines and Comparison]**  and **[Performance of L-RHO on Real-World Benchmark Instances]** for more results and discussions the general response above. In summary, L-RHO achieves a 2.7% gap lower than DRL-20K, when both are compared to the best known solution. Notably, our L-RHO achieves near-optimal performance in the smallest setting, and maintains its advantage in larger settings without any fine-tuning. It also significantly accelerates Default RHO by 24%-38%, highlighting the benefits of learning to enhance RHO and its practical impact on real-world deployment.

---

> ### Author Response · Authors · 2024-11-22
> **Response to Reviewer qP8G (2/2)**
>
> ### **[Apply Learning to Guide Other Algorithms (Q1)]**
> Interesting question! Our learning method can be applied to guide other search-based algorithms, such as large neighborhood search (LNS). LNS begins with a complete solution and iteratively refines it by selecting and solving subproblems. In this context, the variables within a subproblem can be treated as “overlapping” variables. As the LNS iterations progress, an increasing number of these overlapping variables remain unchanged between consecutive iterations. Our learning method could be leveraged to identify and fix a subset of these variables, thereby accelerating the solution of LNS subproblems. Notably, learning-guided LNS for FJSP has not been explored before, making this an exciting and promising direction for future research.
>
> ----------
>
> ### **[Apply L-RHO to Other COPs (Q2)].**
> Thank you for the question regarding the applicability of L-RHO to other long-horizon COPs! This suggestion prompted us to revise our contributions statement; as stated there (General Response 1/4), L-RHO can be applied to any long-horizon COP for which rolling-horizon optimization can be applied. However, we note that the effectiveness of such an approach may vary from application to application and is an important direction of future research. As such, the theoretical analysis in our paper complements our FJSP-focused evaluation to provide application-agnostic insights into when such an approach would be advantageous.
>
> Within FJSP, we do demonstrate the effectiveness of L-RHO on a rich set of variants: we **investigate multiple scheduling objectives, including makespan, start delay, and combined start-end delay, while considering addressing more complex scheduling variants including observation noise and dynamic scenarios like machine breakdowns.** Our evaluation methodology is consistent with (or arguably richer than) learning-for-COP literature published in leading ML conferences, which for various reasons, including compute costs and benchmark availability, tend to focus on canonical problems within a COP class (e.g., routing [1-3] or scheduling [4-5]).
>
> Beyond FJSP, we provide a few demonstrative examples on how one can apply L-RHO to other COPs.  (1) In **Multi-item Inventory Management**, one needs to decide the quantity of inventory to hold over multiple periods. When solving this using a RHO framework, one can fix the quantity of a subset of items and only re-optimize the rest to improve the efficiency of the solving. (2) in **Vehicle Routing with Time Windows**, vehicles visit customers in increasing time order; as new customers come in, instead of re-optimizing the entire route, one can identify which segments of the route can stay the same and only re-optimize a partial segment. To apply L-RHO, the key is to identify fixing which variables in the overlapping subproblems, can be fixed, are amenable to learning, and can lead to a significant time improvement. Again, these extensions are valuable and interesting future directions.
>
> [1] Generalize Learned Heuristics to Solve Large-scale Vehicle Routing Problems in Real-time (ICLR 2023)
>
> [2] Neural Combinatorial Optimization with Heavy Decoder: Toward Large Scale Generalization (NeurIPS 2023)
>
> [3] Towards Omni-generalizable Neural Methods for Vehicle Routing Problems (ICML 2023)
>
> [4] Learning Topological Representations with Bidirectional Graph Attention Network for Solving Job Shop Scheduling Problem (UAI 2024)
>
> [5] Deep Reinforcement Learning Guided Improvement Heuristic for Job Shop Scheduling (ICLR 2024)

---

> > ### Comment · Reviewer_qP8G · 2024-11-29
> >
> > Thank you for your detailed response. I believe the attempt to reduce RHO subproblem sizes through learning is conceptually sound and logical. However, as the learning process appears to provide indirect assistance rather than being directly utilized in solving the problem, I find it somewhat limited in terms of novelty. After carefully reviewing your replies, including those addressing other reviewers’ comments, I have decided to maintain my original score.

---

> ### Author Response · Authors · 2024-11-29
> **Thank you for the reply; Clarifications on potential misunderstanding**
>
> We thank the reviewer for the opportunity to engage in further discussion and acknowledge the soundness of our L-RHO. We understand the key remaining concern is the perception that “the learning process provides indirect assistance rather than being directly utilized in solving the problem.” **However, we believe this may stem from a misunderstanding of our method, or, generally, the role of the ML component in novel work.**
>
>
> 1. Regarding indirect vs direct assistance in solving the problem, our understanding is that you may be referring to end-to-end ML methods as offering direct assistance. We take the perspective that end-to-end ML methods are only one type of ML method. **Our work falls into a broader class of methods that utilize ML in one way or another.** Here, we provide examples of prominent such works that provide “indirect assistance” that have made significant contribution to the field:
>     - **AlphaGo**, where learning augments Monte Carlo Tree Search (MCTS) rather than replacing it in solving Go [1].
>     - **Hyperparameter optimization**, using ML which assists optimization processes without directly solving them [2, 3].
>     - **The existing learning-guided combinatorial optimization literature**, which represent a significant portion of NCO works hybridizing ML with traditional solvers to leverage the strengths of both. Examples include **learning-guided local search** [4-6], **learning-guided ant colony algorithms** [6], **learning-guided dynamic/online VRPs** [7], **learning-guided branch and bound for Mixed Integer Linear Programming solvers** [8-9], and **learning-guided heuristic solvers like Neural LKH** [10]. In all above novel and impactful works, learning predicts useful information that enhances traditional methods rather than providing an end-to-end solution.
>     - Our L-RHO is a novel contribution within the above paradigm. To our knowledge, **we are the first-of-its-kind learning-guided optimization framework for large-scale, long-horizon COPs.**
>
>
> 2. **At the same time, we clarify that we *do* consider the learning in our approach to be directly used utilized in solving the problem.** Our neural network learns to predict a subset of variables to be fixed in both training and inference. This directly enhances CP-SAT solver by largely reducing the problem size, leading to significantly improved performance, accelerated runtime, and state-of-the-art performance across diverse data distributions and variants (see general response 2/4 [Significance of L-RHO Results]).
>
> Please let us know if this clarifies the novelty of our work and if you have further questions or concerns. We look forward to your reply.
>
> **References**
>
> [1] Mastering the game of Go with deep neural networks and tree search (Nature 2016).
>
> [2] How much data is sufficient to learn high-performing algorithms? Generalization guarantees for data-driven algorithm design (STOC 2021)
>
> [3] Learning to configure separators in branch-and-cut (NeurIPS 2023)
>
> [4] Learning to search feasible and infeasible regions of routing problems with flexible neural k-opt (NeurIPS 2023)
>
> [5] Local Search GFlowNet (ICLR 2024)
>
> [6] DeepACO: neural-enhanced ant systems for combinatorial optimization (NeurIPS 2023)
>
> [7] EURO Meets NeurIPS 2022 Vehicle Routing Competition (NeurIPS 2022)
>
> [8] Learning to Remove Cuts in Integer Linear Programming (ICML 2024)
>
> [9] Rethinking Branching on Exact Combinatorial Optimization Solver: The First Deep Symbolic Discovery Framework (ICLR 2024)
>
> [10] NeuroLKH: Combining Deep Learning Model with Lin-Kernighan-Helsgaun Heuristic for Solving the Traveling Salesman Problem (NeurIPS 2021)

---

> > ### Author Response · Authors · 2024-12-02
> >
> > Dear Reviewer qP8G,
> >
> > Thank you once again for your constructive comments and for acknowledging that our L-RHO is noteworthy, comprehensively evaluated, conceptually sound, and logical. Regarding your remaining point on the "novelty of indirect vs. direct assistance in solving the problem," we hope our response has clarified the distinct contributions of both lines of work—learning end-to-end solvers and learning-guided optimization—where our work significantly advances the latter.
> >
> > As the rebuttal period is approaching its ends, we would greatly appreciate it if you could share any remaining concerns or questions, and reconsider your assessment of our work’s novelty and contribution if our response clarifies the confusion. Thank you for the time and effort you have dedicated to reviewing our paper.
> >
> > Best regards,
> >
> > Authors

---

### Official Review · Reviewer_S42P · 2024-11-12

**Soundness:** 3
**Presentation:** 3
**Contribution:** 2
**Rating:** 6
**Confidence:** 3

**Summary:**

The paper proposes a learning-augmented approach for the Flexible Job Shop Scheduling Problem (FJSP) with a long horizon. The authors consider rolling horizon optimization (RHO) where overlapping ordered subproblems are solved sequentially. The idea is to predict which decisions should remain unchanged from one subproblem to the next in order to restrict the subproblem sizes. This may have two impacts: it can accelerate the solving and/or for a fixed time budget per subproblem, improve the quality of the solutions. More precisely, the paper proposes to learn a model which predicts which of the overlapping operations should keep their previous machine assignment. The model is trained in a supervised way where the ground truth is provided by performing RHO and extracting the operations with unchanged assignments. The approach is evaluated on several variants of the FSSP, with varying objectives, distributions and additionnal noise.

**Strengths:**

* The paper makes an original contribution by introducing learning in RHO for the FJSP
* The paper manages to solve much longer horizon problems than existing methods
* In-depth analysis of the performance of the proposed approach in various settings
* The paper is generally clear and well organized.

**Weaknesses:**

* The main weakness is the focus on the FJSP which limits its scope. The idea seems quite generic, so illustrating it on more long-horizon CO problems, e.g. other scheduling variants, would strengthen the contributions and broaden the scope.

* In the theoretical analysis (Sec 6), while the intuition that "the more irregular the operations to be fixed (O∗ fix), the more advantageous LRHO can be, because of the greater potential gain from prediction" is clear, I did not understand how proposition 1 was proving this point. I found the technical part difficult to follow.

* In the experiments, I found that the comparison to DRL-450 is not so informative. While the learning task of proposed L-RHO is a classification on the subproblems, so 450x[the number of supbroblems per instance] samples seem a reasonable training set size, the "DRL" baseline is  an end-to-end constructive method, and it makes sense that its training requires more data than 450 instances.

**Questions:**

* The paper considers several settings as first steps towards the online setting. Why not consider the purely online setting? What kind of adaptations would be needed?
* What's the impact of the time budget per subproblem on the final performance? I imagine if the budget is large enough, standard RHO would lead to the best solutions.

Remarks:
* CP-SAT being exact solver, it would be interesting to have the optimal value as a reference to assess the quality of the other methods. Clearly restricting it to 1800 sec is not eniugh to get high qulity solutions esp for |O|=2000 in Table 1.
* "Enhancing the efficiency and scalability of Combinatorial Optimization Problems (COPs) has been a central focus.." --> missing "solving" or "methods" or such.
L151: "Each sub-problem is limited to T seconds". We understand later that T is the solving time, but It would be nice to make it explicit already here.

---

> ### Author Response · Authors · 2024-11-22
> **Response to Reviewer S42P (1/2)**
>
> We appreciate the reviewer's positive and valuable feedback! We are pleased that the reviewer recognized our L-RHO’s originality, capability to handle longer horizons, and in-depth analysis in various FJSP settings. We hope the following response would clear the remaining concerns.
>
> ----------
>
> ### **[Applicability to other COPs (W1)]**
> Thank you for the suggestion to elaborate on the applicability of L-RHO to other long-horizon COPs! This suggestion prompted us to revise our contributions statement; as stated there (General Response 1/4), L-RHO can be applied to any long-horizon COP for which rolling-horizon optimization can be applied. However, we note that the effectiveness of such an approach may vary from application to application and is an important direction of future research. As such, the theoretical analysis in our paper complements our FJSP-focused evaluation to provide application-agnostic insights into when such an approach would be advantageous.
>
> Within FJSP, we do demonstrate the effectiveness of L-RHO on a rich set of variants: we **investigate multiple scheduling objectives, including makespan, start delay, and combined start-end delay, while considering addressing more complex scheduling variants including observation noise and dynamic scenarios like machine breakdowns**. Our evaluation methodology is consistent with (or arguably richer than) learning-for-COP literature published in leading ML conferences, which for various reasons, including compute costs and benchmark availability, tend to focus on canonical problems within a COP class (e.g., routing [1-3] or scheduling [4-5]).
>
>
> Beyond FJSP, we provide a few demonstrative examples on how one can apply L-RHO to other COPs.  (1) In **Multi-item Inventory Management**, one need to decide the quantity of inventory to hold over multiple periods. When solving this using a RHO framework, one can fix the quantity of a subset of items and only re-optimize the rest to improve the efficiency of the solving. (2) in **Vehicle Routing with Time Windows**, vehicles visit customers in increasing time order; as new customers come in, instead of re-optimizing the entire route, one can identify which segments of the route can stay the same and only re-optimize a partial segment. To apply L-RHO, the key is to identify fixing which variables in the overlapping subproblems, can be fixed, are amenable to learning, and can lead to a significant time improvement. Again, these extensions are valuable and interesting future directions.
>
> [1] Generalize Learned Heuristics to Solve Large-scale Vehicle Routing Problems in Real-time (ICLR 2023)
>
> [2] Neural Combinatorial Optimization with Heavy Decoder: Toward Large Scale Generalization (NeurIPS 2023)
>
> [3] Towards Omni-generalizable Neural Methods for Vehicle Routing Problems (ICML 2023)
>
> [4] Learning Topological Representations with Bidirectional Graph Attention Network for Solving Job Shop Scheduling Problem (UAI 2024)
>
> [5] Deep Reinforcement Learning Guided Improvement Heuristic for Job Shop Scheduling (ICLR 2024)
>
> ----------
>
> ### **[Clarity on Proposition 1 (W2)]**
> Sorry for the confusion! We have updated the theoretical analysis section to make it more clear. Specifically, as seen in Figure 4, middle (in the updated paper) and Proposition 1, the performance of the Random and First baselines depend on the underlying structure of the Oracle distribution. For example, when the problem structure is such that $n_{fix}^*$ is high (more overlapping operations should be fixed), randomly choosing a large number of operations to fix (Random with a high $\sigma_{R}$) results in good performance reflected by low FPR and FPN. Likewise, when the linear decay structure has a higher slope (operations with an earlier precedence order are more likely to be fixed, the First baseline further offers a performance benefit (as seen by the higher $m$ term, which reduces the FPR and FNR of First). When the problem structure is less regular (e.g. more flat slope for the linear decay), we cannot determine whether each operation should be fixed — in those cases, the First baseline’s performance reduces, and that’s when learning is more beneficial.
>
> ----------
>
> ### **[Remove DRL-450 Results (W3)]**
>
>  Thank you for the suggestion. We initially included the DRL-450 Results to highlight that our L-RHO achieves state-of-the-art performance with significantly fewer FJSP training instances than existing DRL works, which is a critical advantage in real-world scenarios where such instances are scarce. Based on your feedback, we have removed the DRL-450 results. Notably, the included baseline DRL-20K actually utilizes more samples than the suggested “450x[the number of subproblems per instance]” — for example, L-RHO for 600 operations contain only 20 subproblems per instance. **Moreover, we have added two new DRL baselines to further demonstrate that L-RHO achieves state-of-the-art performance.**

---

> ### Author Response · Authors · 2024-11-22
> **Response to Reviewer S42P (2/2)**
>
> ### **[Purely Online Setting (Q1)]**
> Thank you for your thoughtful comment. We do believe that our work is largely applicable to online FJSP. However, there is a technicality regarding **purely** online settings, where a single operation is revealed at a time. Because our method leverages the ability to solve shifted subproblems, it is advantageous for the method to observe a batch of new operations. We wrote “one of the first steps towards online setting” simply because we did not want to over-claim that our work is applicable in the limiting case (which we did not consider or evaluate). Upon further thought, we realized that it is often practical even in online settings to consider a batch-online setting; for example, in (online) ride-hailing, it is advantageous to the platform (e.g., Uber) to batch ride requests over 30 seconds rather than match each incoming request one at a time. In response to your comment, we have revised the language to say “Next, we explore FJSP variants, including observation noise and machine breakdowns, to evaluate the applicability of L-RHO to batch-online FJSP.”
>
> ----------
>
> ### **[RHO with Prolonged Runtime (Q2)]**
> Thanks for the Insightful question! Yes, with significantly longer (albeit prohibitively long) time limit, the default RHO can achieve a better objective. To address this comprehensively, we have followed the suggestion and included additional results as follows. As detailed in Appendix A5.4, two parameters control the time limit per subproblem: 1) total time limit, where solving terminates if the time exceeds $T$=60s; 2) early stop time, where solving terminates early if the objective does not improve within $T_{es}$=3s. In the table below, we evaluate the default RHO with increased $T$ and $T_{es}$. **The results show that, as the time limit increases by over 10x, the default RHO achieves better objectives than our L-RHO, albeit with prohibitively long runtime. Our L-RHO delivers superior performance under practical time constraints, offering benefits especially in online cases.**
>
> | **Default RHO** | 600 (10, 20, 30) |           | 800 (10, 20, 40) |           | 1200 (10, 20, 60) |            |
> | ---------------------------------------- | ---------------- | --------- | ---------------- | --------- | ----------------- | ---------- |
> | **Time Limit $T$, Early Stop Time $T_{es}$** | Time (s)         | Makespan  | Time (s)         | Makespan  | Time (s)          | Makespan   |
> | 60, 3 (reported)   | 244 ± 21         | 1558 ± 73 | 348 ± 26         | 2103 ± 78 | 545 ± 36          | 3136 ± 91  |
> | 60, 10   | 599± 55          | 1529 ± 58 | 728 ± 98         | 2044 ± 75 | 1109 ± 108        | 3002 ± 87  |
> | 60, 30   | 923 ± 71         | 1502 ± 66 | 1099 ± 130       | 2028 ± 72 | 1697 ± 175        | 2963 ± 78  |
> | 60, 60    | 819 ± 117        | 1494 ± 67 | 1096 ± 156       | 2018 ± 71 | 1714 ± 154        | 2968 ± 89  |
> | 120, 60     | 1476 ± 312       | 1497 ± 70 | 1811 ± 241       | 1997 ± 70 | 2809 ± 308        | 2932 ± 75  |
> | 180, 120     | 2142 ± 461       | 1490 ± 70 | 2509 ± 346       | 1994 ± 76 | 3932 ± 590        | 2924 ± 83  |
> | 360, 180     | 3631 ± 816       | 1476 ± 64 | 3673 ± 797       | 1987 ± 68 | 5663 ± 900        | 2924 ± 86  |
> | |
> | **L-RHO**     | 126 ± 19         | 1513 ± 70 | 160 ± 23         | 2015 ± 86 | 259 ± 37          | 3011 ± 106 |
>
> ----------
> ### **[Remarks]**
> Thank you for the remarks!  Please see our response: 1) In the updated Table 1, we report the CP-SAT performance with a time limit of 10 hours (36000 seconds). Unfortunately, CP-SAT still performs worse than RHO based methods under this generous time limit; this further illustrates the difficulty of the long horizon FJSP and the importance of decomposition based methods in solving these problems. 2) Thanks for the suggestion. We update the text to “solving COPs” and “solving for T seconds”.
>
> || 600 (10, 20, 30) |            | 800 (10, 20, 40) |            | 1200 (10, 20, 60) |              | 2000 (10, 20, 100) | Transfer     |
> | -- | -- | -- | -- | -- | -- | -- | -- | -- |
> |   | Time (s)         | Makespan   | Time (s)         | Makespan   | Time (s)          | Makespan     | Time (s)           | Makespan     |
> | Full FJSP   | 1800             | 2274 ± 147 | 1800             | 4017 ± 413 | 1800              | 10925 ± 1013 | 1800               | 39585 ± 2707 |
> |             | 36000 (10 hours) | 1583 ± 65  | 36000            | 2128 ± 75  | 36000             | 3206 ± 87    | 36000              | 18821 ± 1986 |
> | L-RHO (450) | 126 ± 19         | 1513 ± 70  | 160 ± 23         | 2015 ± 86  | 259 ± 37          | 3011 ± 106   | 473 ± 52           | 4982 ± 132   |

---

> > ### Author Response · Authors · 2024-11-29
> > **Looking Forward to Your Reply**
> >
> > Dear Reviewer S42P,
> >
> > Thank you again for your valuable suggestions. We have conducted detailed experiments and provided thorough discussions to address your concerns. As the rebuttal period is nearing its conclusion, we wanted to follow up to ensure our responses have adequately addressed the reviewers' concerns. Please let us know if there are any remaining questions or areas where further clarification is needed. We sincerely appreciate the time and effort you have dedicated to reviewing our work. Thank you!
> >
> > Best,
> >
> > Authors

---

> > ### Comment · Reviewer_S42P · 2024-12-02
> > **Reply to Authors Rebuttal**
> >
> > I thank the authors for their clarifications, additional experiments. Most of my concerns have been adequately addressed. I keep my positive score.
> >
> > Although I understand the authors arguments, in my opinion, the evaluation only on FJSP (with various objectives and settings) is still limiting for the scope of the paper. The authors mention a couple of other CO problems where the rolling horizon decomposition could be helpful. I believe that showing the value of the proposed L-RHO approach on these problems would strengthen the paper.
> >
> > A detail: I guess the authors write "Unfortunately, CP-SAT still outperforms RHO based methods under this generous time limit". I guess you mean "does not outperform", which we see in the table.

---

> > > ### Author Response · Authors · 2024-12-02
> > > **Thank you for your valuable feedback and continued support!**
> > >
> > > Thank you so much for your valuable feedback and continued support! We are glad that our rebuttal has adequately addressed most of the reviewer’s concern.
> > >
> > > Regarding the remaining point on the scope of the paper, we appreciate the reviewer’s valuable feedback. Given that our work extensively focuses on *a diverse set of* exemplar long-horizon FJSP variations, encompassing both *online* and *offline* setups, as well as providing both *theoretical analysis* and *empirical evaluations*, we believe this constitutes a significant initial step in demonstrating the benefits of learning-guided optimization for RHO. We will further discuss and highlight the impact of L-RHO on broader COPs in the revised paper.
> > >
> > > Regarding the detail — thank you for catching the typo. You are correct that we meant CP-SAT does not outperform RHO-based methods under the generous time limit. We have updated the response to fix this typo.

---

### Author Response · Authors · 2024-11-22
**General Response (1/4)**

We thank all the reviewers for their valuable comments! We are pleased that most reviewers found our L-RHO **original** (#S42P), **novel** (#ppet) and **noteworthy** (#qP8G). We also appreciate the positive feedback on L-RHO's ability to **handle longer horizons** (#S42P, #PSPu) and **diverse FJSP settings** (#qP8G), as well as its **state-of-the-art performance** (#ppet), **comprehensive experiments** (#qP8G), and **in-depth analysis** (#S42P, #ppet).

Reviewers suggested additional baselines, clarifying the significance of the performance, and refining some details; these suggestions have greatly strengthen our work, and we detail our responses below. We have conducted new expeirments and updated the paper to reflect our responses. We highlight key responses in this general response.


----------

### **[Restating our contributions and impact]**

Reviewer #PSPu identified our contribution as being centered on neural network design, which was not our main focus. To clarify this for future readers, we have revised the paper accordingly to better reflect the primary focus of our work: exploring how to integrate machine learning to solve challenging long-horizon Combinatorial Optimization Problems (COPs), which we then validate using FJSPs. Specifically,

1. **We contribute the first learning-guided rolling-horizon method for COPs, which we call L-RHO. By learning which overlapping solutions in consecutive RHO iterations do not need to be re-optimized, our method substantially reduces RHO subproblem sizes and thus accelerates their solve time.** Our experimental findings, validated using numerous long-horizon FJSPs, suggest that learning has a valuable role to play in algorithms for long-horizon COPs. Our method is generic and can be applied to any long-horizon COP for which a rolling-horizon approach can be applied.
2. **Validated on long-horizon FJSP, L-RHO effectively scales RHO, accelerating it by up to 54% while improving solution quality.**
3. **L-RHO further consistently outperforms various state-of-the-art learning and heuristics baselines (w/ or w/o decomposition) and achieves general effectiveness across various FJSP variations,** including varying sizes, objectives, observation noise, and machine breakdowns.
4. **We propose a principled probabilistic analysis to compare different RHO methods (heuristic and learning).** As the focus of NCO research has been primarily empirical, we believe the theoretical analysis is a valuable start for the community and can be an informative tool to the practitioner for designing whether to introduce learning to a RHO algorithm when solving long-horizon optimization problems.

---

> ### Author Response · Authors · 2024-11-22
> **General Response (2/4)**
>
> ### **[Performance of L-RHO on Real-World Benchmark Instances]**
>
>  In response to Reviewer #PSPu, we add a new comparison on the instances from Dauzere-Peres and Paulli (1997) when scaling them to long-horizon. As shown in the new Table 15 in Appendix A.6.1, L-RHO achieves a 2.7% lower gap than DRL-20K, when both are compared to the best known solution. Notably, L-RHO achieves near-optimal performance in the smallest setting, and maintains its advantage in larger settings without any fine-tuning. It also significantly accelerates Default RHO by 24%-38%, highlighting the benefits of learning to enhance RHO and its practical impact on real-world deployment.
>
> |  | Dauzere Original | Size 196-387   | Dauzere Augment x3 | Size 588-1161           | Dauzere Augment x6 | Size 1176-2322          | Dauzere Augment x9 | Size 1764-3483    |
> | - | - | - | - | - | - | - | - | - |
> |  | Time (s)   | Makespan       | Time (s)            | Makespan                | Time (s)              | Makespan                | Time (s)                | Makespan                |
> | DRL-20K. - Greedy, Transfer     | 1.83 ± 0.1       | 2493.1 ± 44.0  | 5.7 ± 0.5          | 7279.0 ± 68.79          | 12.2 ± 1.0         | 14337.2 ± 218.1         | 21.6 ± 2.6             | 21217.7 ± 249.5         |
> | DRL-20K - Sample 10, Transfer   | 8.10 ± 1.2       | 2278.9 ± 18.7  | 57.4 ± 9.6         | 6946.3 ± 82.5           | 234.7 ± 41.6       | 13935.3 ± 163.7         | 534.3 ±  95.2          | 20783.5 ± 225.8         |
> | DRL-20K - Sample 50, Transfer   | 34.7 ± 6.1       | 2253.3 ± 20.6  | 321.2 ± 58.0       | 6887.0 ± 78.8           | 1341.4 ± 231.1     | 13884.2 ± 161.6         | 3014.3 ± 516.0         | 20722.3 ± 215.6         |
> | DRL-20K - Sample 1000, Transfer | 71.9 ± 12.5      | 2247.0 ± 19.9  | 670.2 ± 116.3      | 6857.6 ± 82.6           | 2679.1 ± 458.3     | 13859.1 ± 159.9         | 6067.6 ± 1038.4        | 20705.8 ± 217.0         |
> | Default RHO                     | 83.0 ± 33.2      | 2173.8 ± 67.8  | 289.7 ± 83.7       | 6504.5 ± 231.9          | 508.7 ± 150.3      | 13085.4 ± 427.7         | 793.5 ± 240.5          | 19688.7 ± 592.3         |
> | Warm Start RHO                  | 75.2 ± 24.5      | 2166.6 ± 67.9  | 241.2 ± 67.3       | 6490.0 ± 236.5          | 449.7 ± 141.1      | 13033.6 ± 431.8         | 683.6 ± 210.3          | 19589.3 ± 608.9         |
> | **L-RHO, Transfer**                  | 65.0 ± 22.5      | 2184.5 ± 65.8  | 199.1 ± 69.6       | 6498.8 ± 254.0          | 344.8 ± 133.9      | 13015.3 ± 503.6         | 492.1 ± 200.3          | 19592.8 ± 724.5         |
> | **Best Known Solution**             |                  | **2120 .0± 78.27** |                    | N/A due to augmentation |                    | N/A due to augmentation |                        | N/A due to augmentation |
>
>
>
> ----------
>
> ### **[Significance of L-RHO Results]**
>
>  Our L-RHO achieves state-of-the-art performance, outperforming various traditional and learning-based baselines both with and without decomposition in various challenging settings. This is significant for the following reasons:
>
> 1. **L-RHO significantly accelerate the Default RHO by 35-54% while enhancing its performance by 3-21%**, as demonstrated in Tables 1 and 2.
> 2. **L-RHO consistently delivers state-of-the-art results across diverse scheduling variants.** This is evidenced by Tables 1 and 2 and Figure 3, across different problem sizes, data distributions, objectives, under online observation noise and machine disruption.
> 3. **L-RHO outperforms DRL baselines with less information and broader applicability.** While the DRL baselines rely on complete problem information and is unsuitable for online problems, L-RHO operates effectively within a limited RHO window, achieving better objectives with comparable or even shorter solve times and is well-suited for addressing online problems.
> 4. **L-RHO excels in its simplicity and effectiveness.** Beyond merely competing on performance benchmarks, where algorithms in the literature are becoming increasingly complex, L-RHO stands out for its remarkable simplicity and effectiveness, outperforming numerous complex baselines across a range of pratical FJSP variations.
> 5. **L-RHO can have real-world impact.** L-RHO achieves a 2.7% gap lower than the DRL baseline on real-world instances. While the number may look small, performance improvements that approach optimality are indeed challenging. As such, a 2.7% performance improvement is significant, and is typical for novel research in this area [3, 4]. In addition, such 2.7% improvement can translate in practice to a 2.7% revenue increase—an enormous benefit for companies managing billion-dollar operations.
>
> [3] Learning Topological Representations with Bidirectional Graph Attention Network for Solving Job Shop Scheduling Problem (UAI 2024)
>
> [4] Deep Reinforcement Learning Guided Improvement Heuristic for Job Shop Scheduling (ICLR 2024)

---

> ### Author Response · Authors · 2024-11-22
> **General Response (3/4)**
>
> ### **[Updated Table 1 with More Baselines and Comparison]**
>  In response to the valuable suggestions from Reviewer #ppet and #PSPu, we have updated Table 1 to include results for CP-SAT and default RHO with longer runtime, as well as two learning baselines recommended by Reviewer #PSPu (we copied the new results in the Table below):
>
> 1. **We add CP-SAT result when giving a generous time limit of 10 hours (36000s).** We find that CP-SAT is still subpar from the RHO based methods. This further illustrates the difficulty of solving long-horizon FJSP.
> 2. **We add another Default RHO result with an increasing time limit.** We find that, in this case, L-RHO still outperforms Default RHO in the objective with a 5 times less solve time.
> 3. **We add two FJSP learning baselines (DRL-Echeverria [1], DRL-Ho [2]) suggested by reviewer PSPu.** We find that the two learning baselines have significantly worse performance than the DRL baseline (Wang et al.) that we compared in the original paper.
>
> [1] Solving large flexible job shop scheduling instances by generating a diverse set of scheduling policies with deep reinforcement learning (arXiv 2023)
>
> [2] Residual scheduling: A new reinforcement learning approach to solving job shop scheduling problem (IEEE Access 2024)
>
>
> | | 600 (10, 20, 30) | | 800 (10, 20, 40) | | 1200 (10, 20, 60) || 2000 (10, 20, 100) | Transfer |
> | - | - | - | - | - | - | - | - | - |
> | | Time (s) | Makespan | Time (s) | Makespan | Time (s) | Makespan | Time (s) | Makespan |
> | CP-SAT (10 hours)  | 36000 | 1583 ± 65  | 36000 | 2128 ± 75  | 36000 | 3206 ± 87    | 36000 | 18821 ± 1986 |
> | CP-SAT | 1800 | 2274 ± 147 | 1800 | 4017 ± 413 | 1800 | 10925 ± 1013 | 1800 | 39585 ± 2707 |
> | DRL-Echeverria, Greedy | 84 ± 27 | 2347 ± 189 | 115 ± 32 | 3105 ± 161 | 105 ± 7 | 4570 ± 218 | 213 ± 10 | 7673 ± 356   |
> | DRL-Echeverria, Sampling | 157 ± 8 | 2351 ± 139 | 217 ± 11 | 3131 ± 165 | 427 ± 20 | 4657 ± 192 | 603 ± 41 | 7774 ± 271   |
> | DRL-Ho, Greedy | 4 ± 2 | 3030 ± 69  | 6 ± 4 | 4038 ±  87 | 11 ± 6 | 6045 ± 99 | 22 ± 3 | 10044 ± 115  |
> | DRL-Ho, Sampling | 174 ± 1 | 2907 ± 46  | 168 ± 14 | 3907 ± 56  | 385 ± 12  | 5870 ± 71    | 517 ± 10 | 9848 ± 92 |
> | DRL-20K, Greedy (Wang et al.) | 4 ± 0.02 | 1628 ± 72 | 6 ± 0.04 | 2128 ± 80 | 10 ± 0.1 | 3141 ± 97   | 16 ± 0.2 | 5184 ± 114  |
> | DRL-20K, Sampling 100 (Wang et al.) | 29 ±  0.3 | 1551 ± 60 | 47 ± 1 | 2048 ± 69 | 110 ± 3 | 3063 ± 81   | 302 ± 10 | 5082 ± 98   |
> | DRL-20K, Sampling 500 (Wang et al.) | 146 ± 5 | 1537 ± 61 | 261 ± 8 | 2031 ± 68 | 597 ± 16 | 3045 ± 79  | 1738 ± 16 | 5062 ± 98   |
> | Default RHO (Long) | 599 ± 55 | 1529 ± 58  | 728 ± 98 | 2044 ± 75  | 1109 ± 108 | 3002 ± 87 | 2871 ± 244 | 4994 ± 114   |
> | Default RHO | 244 ± 21 | 1558 ± 73  | 348 ± 26 | 2103 ± 78  | 545 ± 36  | 3136 ± 91    | 862 ± 42 | 5207 ± 114   |
> | L-RHO (450) | 126 ± 19 | **1513 ± 70**  | 160 ± 23         | **2015 ± 86**  | 259 ± 37 | **3011 ± 106**   | 473 ± 52 | **4982 ± 132**   |

---

> ### Author Response · Authors · 2024-11-22
> **General Response (4/4)**
>
> We would like to share an update regarding the model architecture. When we examine the attention map as suggested by Reviewer #ppet, we realized that our reported architecture’s attention map was not actually used (weights of the attention layer converges to 0 due to an implementation issue). That is, in effect, our reported experimental results reflect a simpler neural architecture design. We are grateful for the reviewer’s comment which led us to uncover this experimental issue. We believe this highlights a strength of our work, as it shows that integrating machine learning to Rolling Horizon Optimization can be highly effective for long-horizon FJSPs, even with a simple neural architecture.
>
> **We provide the following revisions and discussions:**
>
> - We have updated the architecture figure (Figure 2) in the main paper to reflect the architecture without the multi-head attention layer and rephrased the descriptions accordingly. To ensure experimental rigor and eliminate further issues, we carefully examined the network, reloaded the previous checkpoint into the architecture without the attention layer, and verified that the results remain consistent.
> - We conducted a detailed study analyzing the effects of incorporating the attention mechanism after resolving the implementation issue. Our findings show that adding the attention layer improves the True Negative Rate (TNR) but lowers the True Positive Rate (TPR), resulting in a comparable overall accuracy. Results below show that, on FJSP instances (600 operations), the architecture with attention achieves a better makespan but incurs longer solve times compared to the architecture without attention.
> - We note that such observed TNR-TPR tradeoff aligns with our theoretical analysis: as the simpler architecture without attention has a higher TPR but a lower TNR (i.e. it fixed more operations that should not have been fixed), the solve time is reduced with slight sacrifices in the objective. Conversely, the architecture with attention has a higher TNR but a lower TPR (i.e. it failed to operations that should have been fixed), which improves the objective but negatively impacts solve time.
> - **Due to the faster architecture without attention and similar objective, we revise the article to retain the simpler architecture (without attention), and include the detailed comparison with the architecture with the attention layer in Appendix A.6.3.**
>
> | 600 Ops (10, 20, 30)                                                                    | Accuracy | TPR  | TNR  | Precision | Recall | Time (s) | Makespan  |
> | --------------------------------------------------------------------------------------- | -------- | ---- | ---- | --------- | ------ | -------- | --------- |
> | No Attention (reported L-RHO result)                                                    | 0.77     | 0.81 | 0.67 | 0.86      | 0.81   | **126 ± 19** | 1513 ± 70 |
> | With Attention: Overlapping Operations (Query) & New Operations (Key)               | 0.76     | 0.77 | 0.74 | 0.88      | 0.77   | 170 ± 18 | **1504 ± 68** |
> | With Attention: Overlapping Operations (Query) & New Operations (Key)               | 0.75     | 0.75 | 0.75 | 0.88      | 0.75   | 179 ± 21 | 1507 ± 57 |
> | With Attention: Overlapping Operations (Query) & Overlapping & New Operations (Key) | 0.76     | 0.76 | 0.74 | 0.88      | 0.76   | 133 ± 14 | 1527 ± 68 |

---

### Meta-Review · Area_Chair_5gAn · 2024-12-20

**Metareview:**

Thank you for your submission to ICLR. This paper introduces L-RHO, a learning-guided rolling-horizon optimization (RHO) method for combinatorial optimization, in which overlapping solutions in subsequent RHO sub-problems do not require re-optimization, which reduces RHO sub-problem sizes and thus compute time.

This paper presents a thorough set of empirical results, and provides a number of strong additions during the rebuttal period. Results look promising, e.g., significantly reducing solve times, and improving solution quality. While there are still some small concerns about paper organization and clarity, most reviewers stated that their issues were addressed by the end of the rebuttal period and feel positively about this paper. I therefore recommend this paper for acceptance.

**Additional Comments On Reviewer Discussion:**

During the rebuttal period, the authors answered all questions thoroughly, and provided a number of additional empirical results, which showed strong performance of their method. Reviewers, on the whole, appreciated these updates, which led to an increase in scores.

---

### Decision · Program_Chairs · 2025-01-22

Accept (Poster)